# Rectified Factor Networks

**Djork-Arné Clevert, Andreas Mayr, Thomas Unterthiner and Sepp Hochreiter**
Institute of Bioinformatics, Johannes Kepler University, Linz, Austria
{okko,mayr,unterthiner,hochreit}@bioinf.jku.at

## Abstract

We propose rectified factor networks (RFNs) to efficiently construct very sparse, non-linear, high-dimensional representations of the input. RFN models identify rare and small events in the input, have a low interference between code units, have a small reconstruction error, and explain the data covariance structure. RFN learning is a generalized alternating minimization algorithm derived from the posterior regularization method which enforces non-negative and normalized posterior means. We proof convergence and correctness of the RFN learning algorithm.

On benchmarks, RFNs are compared to other unsupervised methods like autoencoders, RBMs, factor analysis, ICA, and PCA. In contrast to previous sparse coding methods, RFNs yield sparser codes, capture the data's covariance structure more precisely, and have a significantly smaller reconstruction error. We test RFNs as pretraining technique for deep networks on different vision datasets, where RFNs were superior to RBMs and autoencoders. On gene expression data from two pharmaceutical drug discovery studies, RFNs detected small and rare gene modules that revealed highly relevant new biological insights which were so far missed by other unsupervised methods.

RFN package for GPU/CPU is available at http://www.bioinf.jku.at/software/rfn.

## 1 Introduction

The success of deep learning is to a large part based on advanced and efficient input representations [1, 2, 3, 4]. These representations are sparse and hierarchical. Sparse representations of the input are in general obtained by rectified linear units (ReLU) [5, 6] and dropout [7]. The key advantage of sparse representations is that dependencies between coding units are easy to model and to interpret. Most importantly, distinct concepts are much less likely to interfere in sparse representations. Using sparse representations, similarities of samples often break down to co-occurrences of features in these samples. In bioinformatics sparse codes excelled in biclustering of gene expression data [8] and in finding DNA sharing patterns between humans and Neanderthals [9].

Representations learned by ReLUs are not only sparse but also *non-negative*. Non-negative representations do not code the degree of absence of events or objects in the input. As the vast majority of events is supposed to be absent, to code for their degree of absence would introduce a high level of random fluctuations. We also aim for *non-linear* input representations to stack models for constructing *hierarchical representations*. Finally, the representations are supposed to have a *large number of coding units* to allow coding of rare and small events in the input. Rare events are only observed in few samples like seldom side effects in drug design, rare genotypes in genetics, or small customer groups in e-commerce. Small events affect only few input components like pathways with few genes in biology, few relevant mutations in oncology, or a pattern of few products in e-commerce. In summary, our goal is to construct input representations that (1) are sparse, (2) are non-negative, (3) are non-linear, (4) use many code units, and (5) model structures in the input data (see next paragraph).

Current unsupervised deep learning approaches like autoencoders or restricted Boltzmann machines (RBMs) do encode all peculiarities in the data (including noise). Generative models can be design

to model specific structures in the data, but their codes cannot be enforced to be sparse and non-negative. The input representation of a generative model is its posterior's mean, median, or mode, which depends on the data. Therefore, sparseness and non-negativity cannot be guaranteed independent of the data. For example, generative models with rectified priors, like rectified factor analysis, have zero posterior probability for negative values, therefore their means are positive and not sparse [10, 11]. Sparse priors like Laplacian and Jeffrey's do not guarantee sparse posteriors (see experiments in Tab. 1). To address the data dependence of the code, we employ the *posterior regularization method* [12]. This method separates model characteristics from data dependent characteristics that are enforced by constraints on the model's posterior.

We aim at representations that are feasible for many code units and massive datasets, therefore the computational complexity of generating a code is essential in our approach. For non-Gaussian priors, the computation of the posterior mean of a new input requires either to numerically solve an integral or to iteratively update variational parameters [13]. In contrast, for Gaussian priors the posterior mean is the product between the input and a matrix that is independent of the input. Still the posterior regularization method leads to a quadratic (in the number of coding units) constrained optimization problem in each E-step (see Eq. (3) below). To speed up computation, we do not solve the quadratic problem but perform a gradient step. To allow for stochastic gradients and fast GPU implementations, also the M-step is a gradient step. These E-step and M-step modifications of the posterior regularization method result in a *generalized alternating minimization* (GAM) algorithm [12]. We will show that the GAM algorithm used for RFN learning (i) converges and (ii) is correct. Correctness means that the RFN codes are non-negative, sparse, have a low reconstruction error, and explain the covariance structure of the data.

## 2   Rectified Factor Network

Our goal is to construct representations of the input that (1) are sparse, (2) are non-negative, (3) are non-linear, (4) use many code units, and (5) model structures in the input. Structures in the input are identified by a generative model, where the model assumptions determine which input structures to explain by the model. We want to model the covariance structure of the input, therefore we choose maximum likelihood factor analysis as model. The constraints on the input representation are enforced by the *posterior regularization method* [12]. *Non-negative constraints* lead to sparse and non-linear codes, while *normalization constraints* scale the signal part of each hidden (code) unit. Normalizing constraints avoid that generative models explain away *rare and small signals* by noise. Explaining away becomes a serious problem for models with many coding units since their capacities are not utilized. Normalizing ensures that all hidden units are used but at the cost of coding also random and spurious signals. Spurious and true signals must be separated in a subsequent step either by supervised techniques, by evaluating coding units via additional data, or by domain experts.

A generative model with hidden units $\boldsymbol{h}$ and data $\boldsymbol{v}$ is defined by its prior $p(\boldsymbol{h})$ and its likelihood $p(\boldsymbol{v} \mid \boldsymbol{h})$. The full model distribution $p(\boldsymbol{h}, \boldsymbol{v}) = p(\boldsymbol{v} \mid \boldsymbol{h})p(\boldsymbol{h})$ can be expressed by the model's posterior $p(\boldsymbol{h} \mid \boldsymbol{v})$ and its evidence (marginal likelihood) $p(\boldsymbol{v})$: $p(\boldsymbol{h}, \boldsymbol{v}) = p(\boldsymbol{h} \mid \boldsymbol{v})p(\boldsymbol{v})$. The representation of input $\boldsymbol{v}$ is the posterior's mean, median, or mode. The posterior regularization method introduces a *variational distribution* $Q(\boldsymbol{h} \mid \boldsymbol{v}) \in \mathcal{Q}$ from a family $\mathcal{Q}$, which approximates the posterior $p(\boldsymbol{h} \mid \boldsymbol{v})$. We choose $\mathcal{Q}$ to constrain the posterior means to be non-negative and normalized. The full model distribution $p(\boldsymbol{h}, \boldsymbol{v})$ contains all model assumptions and, thereby, defines which structures of the data are modeled. $Q(\boldsymbol{h} \mid \boldsymbol{v})$ contains data dependent constraints on the posterior, therefore on the code.

For data $\{\boldsymbol{v}\} = \{\boldsymbol{v}_1, \dots, \boldsymbol{v}_n\}$, the posterior regularization method maximizes the objective $\mathcal{F}$ [12]:

$$\mathcal{F} = \frac{1}{n} \sum_{i=1}^{n} \log p(\boldsymbol{v}_i) - \frac{1}{n} \sum_{i=1}^{n} D_{\mathrm{KL}}(Q(\boldsymbol{h}_i \mid \boldsymbol{v}_i) \parallel p(\boldsymbol{h}_i \mid \boldsymbol{v}_i)) \tag{1}$$

$$= \frac{1}{n} \sum_{i=1}^{n} \int Q(\boldsymbol{h}_i \mid \boldsymbol{v}_i) \, \log p(\boldsymbol{v}_i \mid \boldsymbol{h}_i) \, d\boldsymbol{h}_i - \frac{1}{n} \sum_{i=1}^{n} D_{\mathrm{KL}}(Q(\boldsymbol{h}_i \mid \boldsymbol{v}_i) \parallel p(\boldsymbol{h}_i)) \,,$$

where $D_{\mathrm{KL}}$ is the Kullback-Leibler distance. Maximizing $\mathcal{F}$ achieves two goals simultaneously: (1) extracting desired structures and information from the data as imposed by the generative model and (2) ensuring desired code properties via $Q \in \mathcal{Q}$.

The factor analysis model $\boldsymbol{v} = \boldsymbol{W}\boldsymbol{h} + \boldsymbol{\epsilon}$ extracts the *covariance structure* of the data. The prior $\boldsymbol{h} \sim \mathcal{N}(\boldsymbol{0}, \boldsymbol{I})$ of the hidden units (factors) $\boldsymbol{h} \in \mathbb{R}^l$ and the noise $\boldsymbol{\epsilon} \sim \mathcal{N}(\boldsymbol{0}, \boldsymbol{\Psi})$ of visible units (observations) $\boldsymbol{v} \in \mathbb{R}^m$ are independent. The model parameters are the weight (loading) matrix $\boldsymbol{W} \in \mathbb{R}^{m \times l}$ and the noise covariance matrix $\boldsymbol{\Psi} \in \mathbb{R}^{m \times m}$. We assume diagonal $\boldsymbol{\Psi}$ to explain correlations between input components by the hidden units and not by correlated noise. The factor analysis model is depicted in Fig. 1. Given the mean-centered data $\{\boldsymbol{v}\} = \{\boldsymbol{v}_1, \ldots, \boldsymbol{v}_n\}$, the posterior $p(\boldsymbol{h}_i \mid \boldsymbol{v}_i)$ is Gaussian with mean vector $(\boldsymbol{\mu}_p)_i$ and covariance matrix $\boldsymbol{\Sigma}_p$:

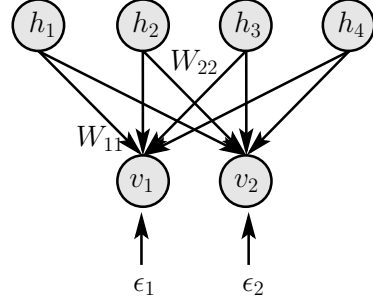

$$(\boldsymbol{\mu}_p)_i = \left(\boldsymbol{I} + \boldsymbol{W}^T \boldsymbol{\Psi}^{-1} \boldsymbol{W}\right)^{-1} \boldsymbol{W}^T \boldsymbol{\Psi}^{-1} \boldsymbol{v}_i ,$$

$$\boldsymbol{\Sigma}_p = \left(\boldsymbol{I} + \boldsymbol{W}^T \boldsymbol{\Psi}^{-1} \boldsymbol{W}\right)^{-1} . \qquad (2)$$

Figure 1: Factor analysis model: hidden units (factors) $\boldsymbol{h}$, visible units $\boldsymbol{v}$, weight matrix $\boldsymbol{W}$, noise $\boldsymbol{\epsilon}$.

A *rectified factor network* (RFN) consists of a single or stacked factor analysis model(s) with constraints on the posterior. To incorporate the posterior constraints into the factor analysis model, we use the posterior regularization method that maximizes the objective $\mathcal{F}$ given in Eq. (1) [12]. Like the expectation-maximization (EM) algorithm, the posterior regularization method alternates between an E-step and an M-step. Minimizing the first $D_{\mathrm{KL}}$ of Eq. (1) with respect to $Q$ leads to a constrained optimization problem. For Gaussian distributions, the solution with $(\boldsymbol{\mu}_p)_i$ and $\boldsymbol{\Sigma}_p$ from Eq. (2) is $Q(\boldsymbol{h}_i \mid \boldsymbol{v}_i) \sim \mathcal{N}(\boldsymbol{\mu}_i, \boldsymbol{\Sigma})$ with $\boldsymbol{\Sigma} = \boldsymbol{\Sigma}_p$ and the quadratic problem:

$$\min_{\boldsymbol{\mu}_i} \; \frac{1}{n} \sum_{i=1}^{n} (\boldsymbol{\mu}_i - (\boldsymbol{\mu}_p)_i)^T \boldsymbol{\Sigma}_p^{-1} (\boldsymbol{\mu}_i - (\boldsymbol{\mu}_p)_i) , \quad \text{s.t.} \quad \forall_i : \boldsymbol{\mu}_i \geq \boldsymbol{0} , \; \forall_j : \frac{1}{n} \sum_{i=1}^{n} \mu_{ij}^2 = 1 , \quad (3)$$

where "$\geq$" is component-wise. This is a constraint non-convex quadratic optimization problem in the number of hidden units which is too complex to be solved in each EM iteration. Therefore, we perform a step of the *gradient projection algorithm* [14, 15], which performs first a gradient step and then projects the result to the feasible set. We start by a step of the *projected Newton method*, then we try the *gradient projection algorithm*, thereafter the *scaled gradient projection algorithm* with reduced matrix [16] (see also [15]). If these methods fail to decrease the objective in Eq. (3), we use the *generalized reduced method* [17]. It solves each equality constraint for one variable and inserts it into the objective while ensuring convex constraints. Alternatively, we use Rosen's gradient projection method [18] or its improvement [19]. These methods guarantee a decrease of the E-step objective.

Since the projection P by Eq. (6) is very fast, the projected Newton and projected gradient update is very fast, too. A projected Newton step requires $O(nl)$ steps (see Eq. (7) and P defined in Theorem 1), a projected gradient step requires $O(\min\{nlm, nl^2\})$ steps, and a scaled gradient projection step requires $O(nl^3)$ steps. The RFN complexity per iteration is $O(n(m^2 + l^2))$ (see Alg. 1). In contrast, a quadratic program solver typically requires for the $(nl)$ variables (the means of the hidden units for all samples) $O(n^4 l^4)$ steps to find the minimum [20]. We exemplify these values on our benchmark datasets MNIST ($n = 50\text{k}, l = 1024, m = 784$) and CIFAR ($n = 50\text{k}, l = 2048, m = 1024$). The speedup with projected Newton or projected gradient in contrast to a quadratic solver is $O(n^3 l^2) = O(n^4 l^4)/O(nl^2)$, which gives **speedup ratios of $1.3 \cdot 10^{20}$ for MNIST and $5.2 \cdot 10^{20}$ for CIFAR.** These speedup ratios show that efficient E-step updates are essential for RFN learning. Furthermore, on our computers, RAM restrictions limited quadratic program solvers to problems with $nl \leq 20\text{k}$. Running times of RFNs with the Newton step and a quadratic program solver are given in the supplementary Section 15.

The M-step decreases the *expected reconstruction error*

$$\mathcal{E} = -\frac{1}{n} \sum_{i=1}^{n} \int_{\mathbb{R}^l} Q(\boldsymbol{h}_i \mid \boldsymbol{v}_i) \log\left(p(\boldsymbol{v}_i \mid \boldsymbol{h}_i)\right) d\boldsymbol{h}_i \qquad (4)$$

$$= \frac{1}{2}\Big(m \log(2\pi) + \log|\boldsymbol{\Psi}| + \mathrm{Tr}\left(\boldsymbol{\Psi}^{-1}\boldsymbol{C}\right) - 2\,\mathrm{Tr}\left(\boldsymbol{\Psi}^{-1}\boldsymbol{W}\boldsymbol{U}^T\right) + \mathrm{Tr}\left(\boldsymbol{W}^T\boldsymbol{\Psi}^{-1}\boldsymbol{W}\boldsymbol{S}\right)\Big) .$$

from Eq. (1) with respect to the model parameters $\boldsymbol{W}$ and $\boldsymbol{\Psi}$. Definitions of $\boldsymbol{C}$, $\boldsymbol{U}$ and $\boldsymbol{S}$ are given in Alg. 1. The M-step performs a gradient step in the Newton direction, since we want to

**Algorithm 1** Rectified Factor Network.

1: $\boldsymbol{C} = \frac{1}{n} \sum_{i=1}^n \boldsymbol{v}_i \boldsymbol{v}_i^T$
2: **while** STOP=false **do**
3:    ——**E-step1**——
4:    **for all** $1 \le i \le n$ **do**
5:      $(\boldsymbol{\mu}_p)_i = \left(\boldsymbol{I} + \boldsymbol{W}^T \boldsymbol{\Psi}^{-1} \boldsymbol{W}\right)^{-1} \boldsymbol{W}^T \boldsymbol{\Psi}^{-1} \boldsymbol{v}_i$
6:    **end for**
7:    $\boldsymbol{\Sigma} = \boldsymbol{\Sigma}_p = \left(\boldsymbol{I} + \boldsymbol{W}^T \boldsymbol{\Psi}^{-1} \boldsymbol{W}\right)^{-1}$
8:    ——**Constraint Posterior**——
9:    (1) projected Newton, (2) projected gradient, (3) scaled gradient projection, (4) generalized reduced method, (5) Rosen's gradient project.
10:    ——**E-step2**——
11:    $\boldsymbol{U} = \frac{1}{n} \sum_{i=1}^n \boldsymbol{v}_i \, \boldsymbol{\mu}_i^T$
12:    $\boldsymbol{S} = \frac{1}{n} \sum_{i=1}^n \boldsymbol{\mu}_i \, \boldsymbol{\mu}_i^T + \boldsymbol{\Sigma}$
13:    ——**M-step**——
14:    $\boldsymbol{E} = \boldsymbol{C} - \boldsymbol{U} \boldsymbol{W}^T - \boldsymbol{W} \boldsymbol{U} + \boldsymbol{W} \boldsymbol{S} \boldsymbol{W}^T$
15:    $\boldsymbol{W} = \boldsymbol{W} + \eta \left(\boldsymbol{U} \boldsymbol{S}^{-1} - \boldsymbol{W}\right)$
16:    **for all** $1 \le k \le m$ **do**
17:      $\Psi_{kk} = \Psi_{kk} + \eta \left(E_{kk} - \Psi_{kk}\right)$
18:    **end for**
19:    if stopping criterion is met: STOP=true
20: **end while**

**Complexity:** objective $\mathcal{F}$: $O(\min\{nlm, nl^2\} + l^3)$; E-step1: $O(\min\{m^2(m+l), l^2(m+l)\} + nlm)$; projected Newton: $O(nl)$; projected gradient: $O(\min\{nlm, nl^2\})$; scaled gradient projection: $O(nl^3)$; E-step2: $O(nl(m+l))$; M-step: $O(ml(m+l))$; overall complexity with projected Newton / gradient for $(l+m) < n$: $O(n(m^2 + l^2))$.

allow stochastic gradients, fast GPU implementation, and dropout regularization. The Newton step is derived in the supplementary which gives further details, too. Also in the E-step, RFN learning performs a gradient step using projected Newton or gradient projection methods. These projection methods require the Euclidean projection P of the posterior means $\{(\boldsymbol{\mu}_p)_i\}$ onto the *non-convex* feasible set:

$$\min_{\boldsymbol{\mu}_i} \frac{1}{n} \sum_{i=1}^n \left(\boldsymbol{\mu}_i - (\boldsymbol{\mu}_p)_i\right)^T \left(\boldsymbol{\mu}_i - (\boldsymbol{\mu}_p)_i\right) , \qquad \text{s.t.} \quad \boldsymbol{\mu}_i \ge \boldsymbol{0} , \frac{1}{n} \sum_{i=1}^n \mu_{ij}^2 = 1 . \quad (5)$$

The following Theorem 1 gives the Euclidean projection P as solution to Eq. (5).

**Theorem 1** (Euclidean Projection). *If at least one $(\mu_p)_{ij}$ is positive for $1 \le j \le l$, then the solution to optimization problem Eq. (5) is*

$$\mu_{ij} = [\mathrm{P}((\boldsymbol{\mu}_p)_i)]_j = \frac{\hat{\mu}_{ij}}{\sqrt{\frac{1}{n} \sum_{i=1}^n \hat{\mu}_{ij}^2}} , \quad \hat{\mu}_{ij} = \begin{cases} 0 & \text{for} \quad (\mu_p)_{ij} \le 0 \\ (\mu_p)_{ij} & \text{for} \quad (\mu_p)_{ij} > 0 \end{cases} . \quad (6)$$

*If all $(\mu_p)_{ij}$ are non-positive for $1 \le j \le l$, then the optimization problem Eq. (5) has the solution $\mu_{ij} = \sqrt{n}$ for $j = \arg\max_{\hat{j}}\{(\mu_p)_{i\hat{j}}\}$ and $\mu_{ij} = 0$ otherwise.*

*Proof.* See supplementary material. $\square$

Using the projection P defined in Eq. (6), the E-step updates for the posterior means $\boldsymbol{\mu}_i$ are:

$$\boldsymbol{\mu}_i^{\text{new}} = \mathrm{P}\left(\boldsymbol{\mu}_i^{\text{old}} + \gamma \left(\boldsymbol{d} - \boldsymbol{\mu}_i^{\text{old}}\right)\right) , \quad \boldsymbol{d} = \mathrm{P}\left(\boldsymbol{\mu}_i^{\text{old}} + \lambda \boldsymbol{H}^{-1} \boldsymbol{\Sigma}_p^{-1}((\boldsymbol{\mu}_p)_i - \boldsymbol{\mu}_i^{\text{old}})\right) \quad (7)$$

where we set for the projected Newton method $\boldsymbol{H}^{-1} = \boldsymbol{\Sigma}_p$ (thus $\boldsymbol{H}^{-1}\boldsymbol{\Sigma}_p^{-1} = \boldsymbol{I}$), and for the projected gradient method $\boldsymbol{H}^{-1} = \boldsymbol{I}$. For the scaled gradient projection algorithm with reduced matrix, the $\epsilon$-active set for $i$ consists of all $j$ with $\mu_{ij} \le \epsilon$. The reduced matrix $\boldsymbol{H}$ is the Hessian $\boldsymbol{\Sigma}_p^{-1}$ with $\epsilon$-active columns and rows $j$ fixed to unit vectors $\boldsymbol{e}_j$. The resulting algorithm is a posterior regularization method with a gradient based E- and M-step, leading to a *generalized alternating minimization* (GAM) algorithm [21]. The RFN learning algorithm is given in Alg. 1. Dropout regularization can be included before E-step2 by randomly setting code units $\mu_{ij}$ to zero with a predefined dropout rate (note that convergence results will no longer hold).

## 3 Convergence and Correctness of RFN Learning

**Convergence of RFN Learning.** Theorem 2 states that Alg. 1 converges to a maximum of $\mathcal{F}$.

**Theorem 2** (RFN Convergence). *The rectified factor network (RFN) learning algorithm given in Alg. 1 is a "generalized alternating minimization" (GAM) algorithm and converges to a solution that maximizes the objective $\mathcal{F}$.*

*Proof.* We present a sketch of the proof which is given in detail in the supplement. For convergence, we show that Alg. 1 is a GAM algorithm which convergences according to Proposition 5 in [21].

Alg. 1 ensures to decrease the M-step objective which is convex in $\boldsymbol{W}$ and $\boldsymbol{\Psi}^{-1}$. The update with $\eta = 1$ leads to the minimum of the objective. Convexity of the objective guarantees a decrease in the M-step for $0 < \eta \leq 1$ if not in a minimum. Alg. 1 ensures to decrease the E-step objective by using gradient projection methods. All other requirements for GAM convergence are also fulfilled. $\qquad\square$

Proposition 5 in [21] is based on Zangwill's generalized convergence theorem, thus updates of the RFN algorithm are viewed as point-to-set mappings [22]. Therefore, the numerical precision, the choice of the methods in the E-step, and GPU implementations are covered by the proof.

**Correctness of RFN Learning.** The goal of the RFN algorithm is to explain the data and its covariance structure. The *expected approximation error $\boldsymbol{E}$* is defined in line 14 of Alg. 1. Theorem 3 states that the RFN algorithm is correct, that is, it explains the data (low reconstruction error) and captures the covariance structure as good as possible.

**Theorem 3** (RFN Correctness). *The fixed point $\boldsymbol{W}$ of Alg. 1 minimizes $\operatorname{Tr}(\boldsymbol{\Psi})$ given $\boldsymbol{\mu}_i$ and $\boldsymbol{\Sigma}$ by ridge regression with*

$$\operatorname{Tr}(\boldsymbol{\Psi}) = \frac{1}{n} \sum_{i=1}^{n} \|\boldsymbol{\epsilon}_i\|_2^2 + \left\|\boldsymbol{W}\,\boldsymbol{\Sigma}^{1/2}\right\|_{\mathrm{F}}^2 , \tag{8}$$

*where $\boldsymbol{\epsilon}_i = \boldsymbol{v}_i - \boldsymbol{W}\,\boldsymbol{\mu}_i$. The model explains the data covariance matrix by*

$$\boldsymbol{C} = \boldsymbol{\Psi} + \boldsymbol{W}\,\boldsymbol{S}\,\boldsymbol{W}^T \tag{9}$$

*up to an error, which is quadratic in $\boldsymbol{\Psi}$ for $\boldsymbol{\Psi} \ll \boldsymbol{W}\boldsymbol{W}^T$. The reconstruction error $\frac{1}{n}\sum_{i=1}^{n}\|\boldsymbol{\epsilon}_i\|_2^2$ is quadratic in $\boldsymbol{\Psi}$ for $\boldsymbol{\Psi} \ll \boldsymbol{W}\boldsymbol{W}^T$.*

*Proof.* The fixed point equation for the $\boldsymbol{W}$ update is $\Delta\boldsymbol{W} = \boldsymbol{U}\boldsymbol{S}^{-1} - \boldsymbol{W} = \boldsymbol{0} \Rightarrow \boldsymbol{W} = \boldsymbol{U}\boldsymbol{S}^{-1}$. Using the definition of $\boldsymbol{U}$ and $\boldsymbol{S}$, we have $\boldsymbol{W} = \left(\frac{1}{n}\sum_{i=1}^{n} \boldsymbol{v}_i\,\boldsymbol{\mu}_i^T\right)\left(\frac{1}{n}\sum_{i=1}^{n} \boldsymbol{\mu}_i\,\boldsymbol{\mu}_i^T + \boldsymbol{\Sigma}\right)^{-1}$. $\boldsymbol{W}$ is the ridge regression solution of

$$\frac{1}{n} \sum_{i=1}^{n} \|\boldsymbol{v}_i - \boldsymbol{W}\,\boldsymbol{\mu}_i\|_2^2 + \left\|\boldsymbol{W}\,\boldsymbol{\Sigma}^{1/2}\right\|_{\mathrm{F}}^2 = \operatorname{Tr}\left(\frac{1}{n}\sum_{i=1}^{n} \boldsymbol{\epsilon}_i\,\boldsymbol{\epsilon}_i^T + \boldsymbol{W}\,\boldsymbol{\Sigma}\,\boldsymbol{W}^T\right) , \tag{10}$$

where $\operatorname{Tr}$ is the trace. After multiplying out all $\boldsymbol{\epsilon}_i\boldsymbol{\epsilon}_i^T$ in $1/n\sum_{i=1}^{n}\boldsymbol{\epsilon}_i\boldsymbol{\epsilon}_i^T$, we obtain:

$$\boldsymbol{E} = \frac{1}{n}\sum_{i=1}^{n} \boldsymbol{\epsilon}_i\,\boldsymbol{\epsilon}_i^T + \boldsymbol{W}\,\boldsymbol{\Sigma}\,\boldsymbol{W}^T . \tag{11}$$

For the fixed point of $\boldsymbol{\Psi}$, the update rule gives: $\operatorname{diag}(\boldsymbol{\Psi}) = \operatorname{diag}\left(\frac{1}{n}\sum_{i=1}^{n}\boldsymbol{\epsilon}_i\boldsymbol{\epsilon}_i^T + \boldsymbol{W}\boldsymbol{\Sigma}\boldsymbol{W}^T\right)$. Thus, $\boldsymbol{W}$ minimizes $\operatorname{Tr}(\boldsymbol{\Psi})$ given $\boldsymbol{\mu}_i$ and $\boldsymbol{\Sigma}$. Multiplying the Woodbury identity for $\left(\boldsymbol{W}\boldsymbol{W}^T + \boldsymbol{\Psi}\right)^{-1}$ from left and right by $\boldsymbol{\Psi}$ gives

$$\boldsymbol{W}\boldsymbol{\Sigma}\boldsymbol{W}^T = \boldsymbol{\Psi} - \boldsymbol{\Psi}\left(\boldsymbol{W}\,\boldsymbol{W}^T + \boldsymbol{\Psi}\right)^{-1}\boldsymbol{\Psi}. \tag{12}$$

Inserting this into the expression for $\operatorname{diag}(\boldsymbol{\Psi})$ and taking the trace gives

$$\operatorname{Tr}\left(\frac{1}{n}\sum_{i=1}^{n}\boldsymbol{\epsilon}_i\,\boldsymbol{\epsilon}_i^T\right) = \operatorname{Tr}\left(\boldsymbol{\Psi}\left(\boldsymbol{W}\boldsymbol{W}^T + \boldsymbol{\Psi}\right)^{-1}\boldsymbol{\Psi}\right) \leq \operatorname{Tr}\left(\left(\boldsymbol{W}\boldsymbol{W}^T + \boldsymbol{\Psi}\right)^{-1}\right)\operatorname{Tr}(\boldsymbol{\Psi})^2 . \tag{13}$$

Therefore, for $\boldsymbol{\Psi} \ll \boldsymbol{W}\boldsymbol{W}^T$ the error is quadratic in $\boldsymbol{\Psi}$. $\boldsymbol{W}\boldsymbol{U}^T = \boldsymbol{W}\boldsymbol{S}\boldsymbol{W}^T = \boldsymbol{U}\boldsymbol{W}^T$ follows from fixed point equation $\boldsymbol{U} = \boldsymbol{W}\boldsymbol{S}$. Using this and Eq. (12), Eq. (11) is

$$\frac{1}{n}\sum_{i=1}^{n}\boldsymbol{\epsilon}_i\,\boldsymbol{\epsilon}_i^T - \boldsymbol{\Psi}\left(\boldsymbol{W}\,\boldsymbol{W}^T + \boldsymbol{\Psi}\right)^{-1}\boldsymbol{\Psi} = \boldsymbol{C} - \boldsymbol{\Psi} - \boldsymbol{W}\,\boldsymbol{S}\,\boldsymbol{W}^T . \tag{14}$$

Using the trace norm (nuclear norm or Ky-Fan n-norm) on matrices, Eq. (13) states that the left hand side of Eq. (14) is quadratic in $\boldsymbol{\Psi}$ for $\boldsymbol{\Psi} \ll \boldsymbol{W}\boldsymbol{W}^T$. The trace norm of a positive semi-definite matrix is its trace and bounds the Frobenius norm [23]. Thus, for $\boldsymbol{\Psi} \ll \boldsymbol{W}\boldsymbol{W}^T$, the covariance is approximated up to a quadratic error in $\boldsymbol{\Psi}$ according to Eq. (9). The diagonal is exactly modeled. $\quad\square$

Since the minimization of the expected reconstruction error $\mathrm{Tr}\left(\boldsymbol{\Psi}\right)$ is based on $\boldsymbol{\mu}_i$, the quality of reconstruction depends on the correlation between $\boldsymbol{\mu}_i$ and $\boldsymbol{v}_i$. We ensure maximal information in $\boldsymbol{\mu}_i$ on $\boldsymbol{v}_i$ by the I-projection (the minimal Kullback-Leibler distance) of the posterior onto the family of rectified and normalized Gaussian distributions.

## 4 Experiments

**RFNs vs. Other Unsupervised Methods.** We assess the performance of rectified factor networks (RFNs) as unsupervised methods for data representation. We compare (1) **RFN**: rectified factor networks, (2) **RFNn**: RFNs without normalization, (3) **DAE**: denoising autoencoders with ReLUs, (4) **RBM**: restricted Boltzmann machines with Gaussian visible units, (5) **FAsp**: factor analysis with Jeffrey's prior ($p(z) \propto 1/z$) on the hidden units which is sparser than a Laplace prior, (6) **FAlap**: factor analysis with Laplace prior on the hidden units, (7) **ICA**: independent component analysis by FastICA [24], (8) **SFA**: sparse factor analysis with a Laplace prior on the parameters, (9) **FA**: standard factor analysis, (10) **PCA**: principal component analysis. The number of components are fixed to 50, 100 and 150 for each method. We generated nine different benchmark datasets (D1 to D9), where each dataset consists of 100 instances. Each instance has 100 samples and 100 features resulting in a $100{\times}100$ matrix. Into these matrices, biclusters are implanted [8]. A bicluster is a pattern of particular features which is found in particular samples like a pathway activated in some samples. An optimal representation will only code the biclusters that are present in a sample. The datasets have different noise levels and different bicluster sizes. Large biclusters have 20–30 samples and 20–30 features, while small biclusters 3–8 samples and 3–8 features. The pattern's signal strength in a particular sample was randomly chosen according to the Gaussian $\mathcal{N}\left(1,1\right)$. Finally, to each matrix, zero-mean Gaussian background noise was added with standard deviation 1, 5, or 10. The datasets are characterized by Dx=$(\sigma, n_1, n_2)$ with background noise $\sigma$, number of large biclusters $n_1$, and the number of small biclusters $n_2$: D1=(1,10,10), D2=(5,10,10), D3=(10,10,10), D4=(1,15,5), D5=(5,15,5), D6=(10,15,5), D7=(1,5,15), D8=(5,5,15), D9=(10,5,15).

We evaluated the methods according to the (1) *sparseness* of the components, the (2) input *reconstruction error* from the code, and the (3) *covariance reconstruction error* for generative models. For RFNs sparseness is the percentage of the components that are exactly 0, while for others methods it is the percentage of components with an absolute value smaller than 0.01. The reconstruction error is the sum of the squared errors across samples. The covariance reconstruction error is the Frobenius norm of the difference between model and data covariance. See supplement for more details on the data and for information on hyperparameter selection for the different methods. Tab. 1 gives averaged results for models with 50 (undercomplete), 100 (complete) and 150 (overcomplete) coding units. Results are the mean of 900 instances consisting of 100 instances for each dataset D1 to D9. In the supplement, we separately tabulate the results for D1 to D9 and confirm them with different noise levels. FAlap did not yield sparse codes since the variational parameter did not

Table 1: Comparison of RFN with other unsupervised methods, where the upper part contains methods that yielded sparse codes. Criteria: sparseness of the code (SP), reconstruction error (ER), difference between data and model covariance (CO). The panels give the results for models with 50, 100 and 150 coding units. Results are the mean of 900 instances, 100 instances for each dataset D1 to D9 (maximal value: 999). RFNs had the sparsest code, the lowest reconstruction error, and the lowest covariance approximation error of all methods that yielded sparse representations (SP>10%).

| | **undercomplete** 50 code units | | | **complete** 100 code units | | | **overcomplete** 150 code units | | |
|---|---|---|---|---|---|---|---|---|---|
| | SP | ER | CO | SP | ER | CO | SP | ER | CO |
| RFN | $75_{\pm0}$ | $249_{\pm3}$ | $108_{\pm3}$ | $81_{\pm1}$ | $68_{\pm9}$ | $26_{\pm6}$ | $85_{\pm1}$ | $17_{\pm6}$ | $7_{\pm6}$ |
| RFNn | $74_{\pm0}$ | $295_{\pm4}$ | $140_{\pm4}$ | $79_{\pm0}$ | $185_{\pm5}$ | $59_{\pm3}$ | $80_{\pm0}$ | $142_{\pm4}$ | $35_{\pm2}$ |
| DAE | $66_{\pm0}$ | $251_{\pm3}$ | — | $69_{\pm0}$ | $147_{\pm2}$ | — | $71_{\pm0}$ | $130_{\pm2}$ | — |
| RBM | $15_{\pm1}$ | $310_{\pm4}$ | — | $7_{\pm1}$ | $287_{\pm4}$ | — | $5_{\pm0}$ | $286_{\pm4}$ | — |
| FAsp | $40_{\pm1}$ | $999_{\pm63}$ | $999_{\pm99}$ | $63_{\pm0}$ | $999_{\pm65}$ | $999_{\pm99}$ | $80_{\pm0}$ | $999_{\pm65}$ | $999_{\pm99}$ |
| FAlap | $4_{\pm0}$ | $239_{\pm6}$ | $341_{\pm19}$ | $6_{\pm0}$ | $46_{\pm4}$ | $985_{\pm45}$ | $4_{\pm0}$ | $46_{\pm4}$ | $976_{\pm53}$ |
| ICA | $2_{\pm0}$ | $174_{\pm2}$ | — | $3_{\pm1}$ | $0_{\pm0}$ | — | $3_{\pm1}$ | $0_{\pm0}$ | — |
| SFA | $1_{\pm0}$ | $218_{\pm5}$ | $94_{\pm3}$ | $1_{\pm0}$ | $16_{\pm1}$ | $114_{\pm5}$ | $1_{\pm0}$ | $16_{\pm1}$ | $285_{\pm7}$ |
| FA | $1_{\pm0}$ | $218_{\pm4}$ | $90_{\pm3}$ | $1_{\pm0}$ | $16_{\pm1}$ | $83_{\pm4}$ | $1_{\pm0}$ | $16_{\pm1}$ | $263_{\pm6}$ |
| PCA | $0_{\pm0}$ | $174_{\pm2}$ | — | $2_{\pm0}$ | $0_{\pm0}$ | — | $2_{\pm0}$ | $0_{\pm0}$ | — |

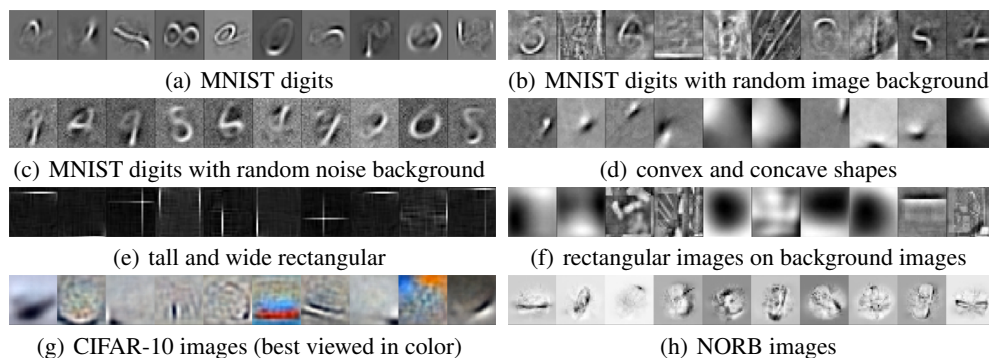

(a) MNIST digits

(b) MNIST digits with random image background

(c) MNIST digits with random noise background

(d) convex and concave shapes

(e) tall and wide rectangular

(f) rectangular images on background images

(g) CIFAR-10 images (best viewed in color)

(h) NORB images

Figure 2: Randomly selected filters trained on image datasets using an RFN with 1024 hidden units. RFNs learned stroke, local and global blob detectors. RFNs are robust to background noise (b,c,f).

push the absolute representations below the threshold of 0.01. The variational approximation to the Laplacian is a Gaussian [13]. *RFNs had the sparsest code, the lowest reconstruction error, and the lowest covariance approximation error of all methods yielding sparse representations (SP>10%).*

**RFN Pretraining for Deep Nets.** We assess the performance of rectified factor networks (RFNs) if used for pretraining of deep networks. Stacked RFNs are obtained by first training a single layer RFN and then passing on the resulting representation as input for training the next RFN. The deep network architectures use a RFN pretrained first layer (RFN-1) or stacks of 3 RFNs giving a 3-hidden layer network. The classification performance of deep networks with RFN pretrained layers was compared to (i) support vector machines, (ii) deep networks pretrained by stacking denoising autoencoders (SDAE), (iii) stacking regular autoencoders (SAE), (iv) restricted Boltzmann machines (RBM), and (v) stacking restricted Boltzmann machines (DBN).

The benchmark datasets and results are taken from previous publications [25, 26, 27, 28] and contain: (i) *MNIST* (original MNIST), (ii) *basic* (a smaller subset of MNIST for training), (iii) *bg-rand* (MNIST with random noise background), (iv) *bg-img* (MNIST with random image background), (v) *rect* (tall or wide rectangles), (vi) *rect-img* (tall or wide rectangular images with random background images), (vii) *convex* (convex or concave shapes), (viii) *CIFAR-10* (60k color images in 10 classes), and (ix) *NORB* (29,160 stereo image pairs of 5 categories). For each dataset its size of training, validation and test set is given in the second column of Tab. 2. As preprocessing we only performed median centering. Model selection is based on the validation set [26]. The RFNs hyper-parameters are (i) the number of units per layer from $\{1024, 2048, 4096\}$ and (ii) the dropout rate from $\{0.0, 0.25, 0.5, 0.75\}$. The learning rate was fixed to $\eta = 0.01$ (default value). For supervised fine-tuning with stochastic gradient descent, we selected the learning rate from $\{0.1, 0.01, 0.001\}$, the masking noise from $\{0.0, 0.25\}$, and the number of layers from $\{1, 3\}$. Fine-tuning was stopped based on the validation set, see [26]. Fig. 2 shows learned filters. Test error rates and the 95%

Table 2: Results of deep networks pretrained by RFNs and other models (taken from [25, 26, 27, 28]). The test error rate is reported together with the 95% confidence interval. The best performing method is given in bold, as well as those for which confidence intervals overlap. The first column gives the dataset, the second the size of training, validation and test set, the last column indicates the number of hidden layers of the selected deep network. In only one case RFN pretraining was significantly worse than the best method but still the second best. In six out of the nine experiments RFN pretraining performed best, where in four cases it was significantly the best.

| Dataset | | SVM | RBM | DBN | SAE | SDAE | RFN |
|---|---|---|---|---|---|---|---|
| MNIST | 50k-10k-10k | $\mathbf{1.40}_{\pm 0.23}$ | $\mathbf{1.21}_{\pm 0.21}$ | $\mathbf{1.24}_{\pm 0.22}$ | $\mathbf{1.40}_{\pm 0.23}$ | $\mathbf{1.28}_{\pm 0.22}$ | $\mathbf{1.27}_{\pm 0.22}$ (1) |
| basic | 10k-2k-50k | $3.03_{\pm 0.15}$ | $3.94_{\pm 0.17}$ | $3.11_{\pm 0.15}$ | $3.46_{\pm 0.16}$ | $\mathbf{2.84}_{\pm 0.15}$ | $\mathbf{2.66}_{\pm 0.14}$ (1) |
| bg-rand | 10k-2k-50k | $14.58_{\pm 0.31}$ | $9.80_{\pm 0.26}$ | $\mathbf{6.73}_{\pm 0.22}$ | $11.28_{\pm 0.28}$ | $10.30_{\pm 0.27}$ | $7.94_{\pm 0.24}$ (3) |
| bg-img | 10k-2k-50k | $22.61_{\pm 0.37}$ | $\mathbf{16.15}_{\pm 0.32}$ | $16.31_{\pm 0.32}$ | $23.00_{\pm 0.37}$ | $\mathbf{16.68}_{\pm 0.33}$ | $\mathbf{15.66}_{\pm 0.32}$ (1) |
| rect | 1k-0.2k-50k | $2.15_{\pm 0.13}$ | $4.71_{\pm 0.19}$ | $2.60_{\pm 0.14}$ | $2.41_{\pm 0.13}$ | $1.99_{\pm 0.12}$ | $\mathbf{0.63}_{\pm 0.06}$ (1) |
| rect-img | 10k-2k-50k | $24.04_{\pm 0.37}$ | $23.69_{\pm 0.37}$ | $22.50_{\pm 0.37}$ | $24.05_{\pm 0.37}$ | $21.59_{\pm 0.36}$ | $\mathbf{20.77}_{\pm 0.36}$ (1) |
| convex | 10k-2k-50k | $19.13_{\pm 0.34}$ | $19.92_{\pm 0.35}$ | $18.63_{\pm 0.34}$ | $18.41_{\pm 0.34}$ | $19.06_{\pm 0.34}$ | $\mathbf{16.41}_{\pm 0.32}$ (1) |
| NORB | 19k-5k-24k | $11.6_{\pm 0.40}$ | $8.31_{\pm 0.35}$ | - | $10.10_{\pm 0.38}$ | $9.50_{\pm 0.37}$ | $\mathbf{7.00}_{\pm 0.32}$ (1) |
| CIFAR | 40k-10k-10k | $62.7_{\pm 0.95}$ | $\mathbf{40.39}_{\pm 0.96}$ | $43.38_{\pm 0.97}$ | $43.25_{\pm 0.97}$ | - | $\mathbf{41.29}_{\pm 0.95}$ (1) |

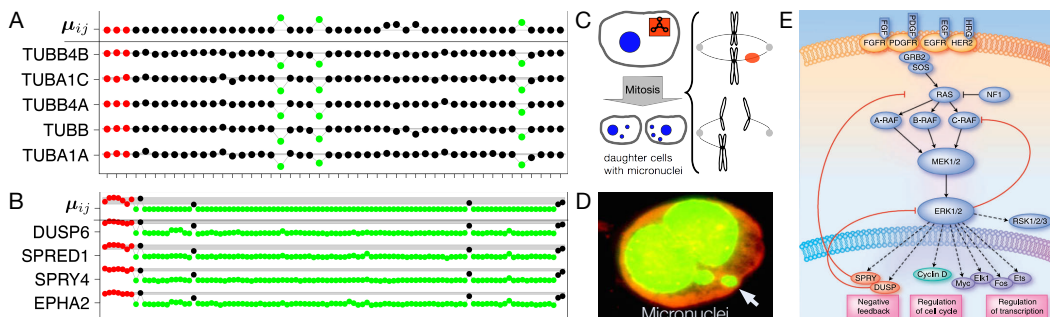

Figure 3: Examples of small and rare events identified by RFN in two drug design studies, which were missed by previous methods. Panel A and B: first row gives the coding unit, while the other rows display expression values of genes for controls (red), active drugs (green), and inactive drugs (black). Drugs (green) in panel A strongly downregulate the expression of tubulin genes which hints at a genotoxic effect by the formation of micronuclei (C). The micronuclei were confirmed by microscopic analysis (D). Drugs (green) in panel B show a transcriptional effect on genes with a negative feedback to the MAPK signaling pathway (E) and therefore are potential cancer drugs.

confidence interval (computed according to [26]) for deep network pretraining by RFNs and other methods are given in Tab. 2. Best results and those with overlapping confidence intervals are given in bold. RFNs were only once significantly worse than the best method but still the second best. In six out of the nine experiments RFNs performed best, where in four cases it was significantly the best. Supplementary Section 14 shows results of RFN pretraining for convolutional networks, where RFN pretraining decreased the test error rates to 7.63% for CIFAR-10 and to 29.75% for CIFAR-100.

**RFNs in Drug Discovery.** Using RFNs we analyzed gene expression datasets of two projects in the lead optimization phase of a big pharmaceutical company [29]. The first project aimed at finding novel antipsychotics that target PDE10A. The second project was an oncology study that focused on compounds inhibiting the FGF receptor. In both projects, the expression data was summarized by FARMS [30] and standardized. RFNs were trained with 500 hidden units, no masking noise, and a learning rate of $\eta = 0.01$. The identified transcriptional modules are shown in Fig. 3. Panels A and B illustrate that RFNs found rare and small events in the input. In panel A only a few drugs are genotoxic (rare event) by downregulating the expression of a small number of tubulin genes (small event). The genotoxic effect stems from the formation of micronuclei (panel C and D) since the mitotic spindle apparatus is impaired. Also in panel B, RFN identified a rare and small event which is a transcriptional module that has a negative feedback to the MAPK signaling pathway. Rare events are unexpectedly inactive drugs (black dots), which do not inhibit the FGF receptor. Both findings were not detected by other unsupervised methods, while they were highly relevant and supported decision-making in both projects [29].

## 5 Conclusion

We have introduced rectified factor networks (RFNs) for constructing very sparse and non-linear input representations with many coding units in a generative framework. Like factor analysis, RFN learning explains the data variance by its model parameters. The RFN learning algorithm is a posterior regularization method which enforces non-negative and normalized posterior means. We have shown that RFN learning is a generalized alternating minimization method which can be proved to converge and to be correct. RFNs had the sparsest code, the lowest reconstruction error, and the lowest covariance approximation error of all methods that yielded sparse representations (SP>10%). RFNs have shown that they improve performance if used for pretraining of deep networks. In two pharmaceutical drug discovery studies, RFNs detected small and rare gene modules that were so far missed by other unsupervised methods. These gene modules were highly relevant and supported the decision-making in both studies. RFNs are geared to large datasets, sparse coding, and many representational units, therefore they have high potential as unsupervised deep learning techniques.

**Acknowledgment.** The Tesla K40 used for this research was donated by the NVIDIA Corporation.

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
