[Supplementary Material]

# Rectified Factor Networks
# Supplement

**Djork-Arné Clevert, Andreas Mayr, Thomas Unterthiner and Sepp Hochreiter**
Institute of Bioinformatics, Johannes Kepler University, Linz, Austria
{okko,mayr,unterthiner,hochreit}@bioinf.jku.at

## Contents

# List of Theorems

# List of Algorithms

# 1 Introduction

This supplement contains additional information complementing the main manuscript and is structured as follows: First, the rectified factor network (RFN) learning algorithm with E- and M-step updates, weight decay and dropout regularization is given in Section 2. In Section 3, we proof that the (RFN) learning algorithm is a "generalized alternating minimization" (GAM) algorithm and converges to a solution that maximizes the RFN objective. The correctness of the RFN algorithm is proofed in Section 4. Section 5 describes the maximum likelihood factor analysis model and the model selection by the EM-algorithm. The RFN objective, which has to be maximized, is described in Section 6. Next, RFN's GAM algorithm via gradient descent both in the M-step and the E-step is reported in the Section 7. The following sections 8 and 9 describe the gradient-based M- and E-step, respectively. In Section 10, we describe how the RFNs sparseness can be controlled by a Gaussian prior. Additional information on the selected hyperparameters of the benchmark methods is given in Section 11. The sections 12 and 13 describe the data generation of the benchmark datasets and report the results for three different experimental settings, namely for extracting 50 (undercomplete), 100 (complete) or 150 (overcomplete) factors / hidden units. In Section 14 describes experiments, that we have done to assess the performance of RFN *first layer* pretraining on *CIFAR-10* and *CIFAR-100* for three deep convolutional network architectures: (i) the AlexNet Ciresan et al. [2012], Krizhevsky et al. [2012], (ii) Deeply Supervised Networks (DSN) Lee et al. [2014], and (iii) our 5-Convolution-Network-In-Network (5C-NIN). Finally, Section 15 provides running times for RFN's projected Newton step and for solving a quadratic program.

# 2 Rectified Factor Network (RFN) Algorithms

Algorithm 1 is the rectified factor network (RFN) learning algorithm. The RFN algorithm calls Algorithm 2 to project the posterior probability $p_i$ onto the family of rectified and normalized variational distributions $Q_i$. Algorithm 2 guarantees an improvement of the E-step objective $O = \frac{1}{n} \sum_{i=1}^{n} D_{\mathrm{KL}}(Q_i \parallel p_i)$. Projection Algorithm 2 relies on different projections, where a more complicated projection is tried if a simpler one failed to improve the E-step objective. If all following Newton-based gradient projection methods fail to decrease the E-step objective, then projection Algorithm 2 falls back to gradient projection methods. First the equality constraints are solved and inserted into the objective. Thereafter, the constraints are convex and gradient projection methods are applied. This approach is called "generalized reduced gradient method" Abadie and Carpentier [1969], which is our preferred alternative method. If this method fails, then Rosen's gradient projection method Rosen [1961] is used. Finally, the method of Haug and Arora Haug and Arora [1979] is used.

First we consider Newton-based projection methods, which are used by Algorithm 2. Algorithm 4 performs a simple projection, which is the projected Newton method with learning rate set to one. This projection is very fast and ideally suited to be performed on GPUs for RFNs with many coding units. Algorithm 3 is the fast and simple projection without normalization even simpler than Algorithm 4. Algorithm 5 generalizes Algorithm 4 by introducing step sizes $\lambda$ and $\gamma$. The step size $\lambda$ scales the gradient step, while $\gamma$ scales the difference between to old projection and the new projection. For both $\lambda$ and $\gamma$ annealing steps, that is, learning rate decay is used to find an appropriate update.

If these Newton-based update rules do not work, then Algorithm 6 is used. Algorithm 6 performs a scaled projection with a reduced Hessian matrix $\boldsymbol{H}$ instead of the full Hessian $\boldsymbol{\Sigma}_p^{-1}$. For computing $\boldsymbol{H}$ an $\epsilon$-active set is determined, which consists of all $j$ with $\mu_j \leq \epsilon$. The reduced matrix $\boldsymbol{H}$ is the Hessian $\boldsymbol{\Sigma}_p^{-1}$ with $\epsilon$-active columns and rows $j$ fixed to unit vector $\boldsymbol{e}_j$.

The RFN algorithm allows regularization of the parameters $\boldsymbol{W}$ and $\boldsymbol{\Psi}$ (off-diagonal elements) by weight decay. Priors on the parameters can be introduced. If the priors are convex functions, then convergence of the RFN algorithm is still ensured. The weight decay Algorithm 7 can optionally be used after the M-step of Algorithm 1. Coding units can be regularized by dropout. However dropout is not covered by the convergence proof for the RFN algorithm. The dropout Algorithm 8 is applied during the projection between rectifying and normalization. Methods like mini-batches or other stochastic gradient methods are not covered by the convergence proof for the RFN algorithm. However, in Gunawardana and Byrne [2005] it is shown how to generalize the GAM convergence

proof to mini-batches as it is shown for the incremental EM algorithm. Dropout and other stochastic gradient methods can be show to converge similar to mini-batches.

---

**Algorithm 1** Rectified Factor Network

---

**Input**
    for $1 \leq i \leq n$: $\boldsymbol{v}_i \in \mathbb{R}^m$,
    number of coding units $l$
**Hyper-Parameters**
    $\Psi_{\min}, W_{\max}, \eta_\Psi, \eta_W, \rho, \tau, 1 < \eta \leq 1$
**Initialization**
    $\boldsymbol{\Psi} = \tau \boldsymbol{I}$, $\boldsymbol{W}$ element-wise random in $[-\rho, \rho]$,
    $\boldsymbol{C} = \frac{1}{n} \sum_{k=1}^{n} \boldsymbol{v}_k \, \boldsymbol{v}_k^T$, STOP=false
**Main**
  **while** STOP=false **do**
      ——**E-step1**——
    **for all** $1 \leq i \leq n$ **do**
      $(\boldsymbol{\mu}_p)_i = \left(\boldsymbol{I} + \boldsymbol{W}^T \boldsymbol{\Psi}^{-1} \boldsymbol{W}\right)^{-1} \boldsymbol{W}^T \boldsymbol{\Psi}^{-1} \, \boldsymbol{v}_i$
    **end for**
    $\boldsymbol{\Sigma} = \left(\boldsymbol{I} + \boldsymbol{W}^T \boldsymbol{\Psi}^{-1} \boldsymbol{W}\right)^{-1}$
    ——**Projection**——
    perform projection of $(\boldsymbol{\mu}_p)_i$ onto the feasible set by Algorithm 2 giving $\boldsymbol{\mu}_i$
    ——**E-step2**——
    $\boldsymbol{U} = \frac{1}{n} \sum_{i=1}^{n} \boldsymbol{v}_i \, \boldsymbol{\mu}_i^T$

    $\boldsymbol{S} = \frac{1}{n} \sum_{i=1}^{n} \boldsymbol{\mu}_i \, \boldsymbol{\mu}_i^T + \boldsymbol{\Sigma}$
    ——**M-step**——
    $\eta_W = \eta_\Psi = \eta$
    $\boldsymbol{E} = \boldsymbol{C} - \boldsymbol{U} \, \boldsymbol{W}^T - \boldsymbol{W} \, \boldsymbol{U} + \boldsymbol{W} \, \boldsymbol{S} \, \boldsymbol{W}^T$
    ——*W update*——
    $\boldsymbol{W} = \boldsymbol{W} + \eta_W \left(\boldsymbol{U} \, \boldsymbol{S}^{-1} - \boldsymbol{W}\right)$
    ——*diagonal* $\boldsymbol{\Psi}$ *update*——
    **for all** $1 \leq k \leq m$ **do**
      $\Psi_{kk} = \Psi_{kk} + \eta_\Psi \left(E_{kk} - \Psi_{kk}\right)$
    **end for**
    ——*full* $\boldsymbol{\Psi}$ *update*——
    $\boldsymbol{\Psi} = \boldsymbol{\Psi} + \eta_\Psi \left(\boldsymbol{E} - \boldsymbol{\Psi}\right)$
    ——*bound parameters*——
    $\boldsymbol{W} = \text{median}\{-W_{\max}, \boldsymbol{W}, W_{\max}\}$
    $\boldsymbol{\Psi} = \text{median}\{\Psi_{\min}, \boldsymbol{\Psi}, \max\{\boldsymbol{C}\}\}$
    if stopping criterion is met: STOP=true
  **end while**

---

---

**Algorithm 2** Projection with E-Step Improvement

---

**Goal**
  obtain $\boldsymbol{\mu}_i^{\text{new}} = \boldsymbol{\mu}_i$ that decrease the E-step objective
**Input**
  $\boldsymbol{\Sigma}^{\text{new}} = \boldsymbol{\Sigma}_p, \boldsymbol{\Sigma}^{\text{old}} = \boldsymbol{\Sigma}_p^{\text{old}}$
  for $1 \leq i \leq n$: $(\boldsymbol{\mu}_p)_i, \boldsymbol{\mu}_i^{\text{old}}, p_i = \mathcal{N}((\boldsymbol{\mu}_p)_i, \boldsymbol{\Sigma}_p)$
  simple projection P (rectified or rectified & normalized),
  E-step objective: $O = \frac{1}{n} \sum_{i=1}^n D_{\text{KL}}(Q_i \parallel p_i)$
  $\gamma_{\min}, \lambda_{\min}, \rho_\gamma, \rho_\lambda, \epsilon$ (for $\epsilon$-active set)
**Main**
  ——Simple Projection——
  perform `Newton Projection` by Algorithm 4 or Algorithm 3
  ——Scaled Projection——
  **if** $0 \leq \Delta O$ **then**
     following loop for: (1) $\gamma$, (2) $\lambda$, or (3) $\gamma$ and $\lambda$ annealing
     $\gamma = \lambda = 1$
     **while** $0 \leq \Delta O$ and $\lambda > \lambda_{\min}$ and $\gamma > \gamma_{\min}$ **do**
        $\gamma = \rho_\gamma \gamma$ (skipped for $\lambda$ annealing)
        $\lambda = \rho_\lambda \lambda$ (skipped for $\gamma$ annealing)
        perform `Scaled Newton Projection` by Algorithm 5
     **end while**
  **end if**
  ——Scaled Projection With Reduced Matrix——
  **if** $0 \leq \Delta O$ **then**
     determine $\epsilon$-active set as all $j$ with $\mu_j \leq \epsilon$
     set $\boldsymbol{H}$ to $\boldsymbol{\Sigma}_p^{-1}$ with $\epsilon$-active columns and rows $j$ fixed to $\boldsymbol{e}_j$
     following loop for: (1) $\gamma$, (2) $\lambda$, or (3) $\gamma$ and $\lambda$ annealing
     $\gamma = \lambda = 1$
     **while** $0 \leq \Delta O$ and $\lambda > \lambda_{\min}$ and $\gamma > \gamma_{\min}$ **do**
        $\gamma = \rho_\gamma \gamma$ (skipped for $\lambda$ annealing)
        $\lambda = \rho_\lambda \lambda$ (skipped for $\gamma$ annealing)
        perform `Scaled Projection With Reduced Matrix` by Algorithm 6
     **end while**
  **end if**
  ——General Gradient Projection——
  **while** $0 \leq \Delta O$ **do**
     use generalized reduced gradient Abadie and Carpentier [1969] OR
     use Rosen's gradient projection Rosen [1961] OR
     use method of Haug and Arora Haug and Arora [1979]
  **end while**

---

---

**Algorithm 3** Simple Projection: Rectifying

---

**Goal**
  for $1 \leq i \leq n$: project $(\boldsymbol{\mu}_p)_i$ onto feasible set giving $\boldsymbol{\mu}_i$
**Input**
  $(\boldsymbol{\mu}_p)_i$
**Main**
  **for all** $1 \leq j \leq l$ **do**
     $\mu_{ij} = \max \left\{ 0, [(\boldsymbol{\mu}_p)_i]_j \right\}$
  **end for**

---

---
**Algorithm 4** Simple Projection: Rectifying and Normalization
---
**Goal**
  for $1 \leq i \leq n$: project $(\boldsymbol{\mu}_p)_i$ onto feasible set giving $\boldsymbol{\mu}_i$
**Input**
  for $1 \leq i \leq n$: $(\boldsymbol{\mu}_p)_i$
**Rectifier**
  **for all** $1 \leq i \leq n$ **do**
    **for all** $1 \leq j \leq l$ **do**
      $\hat{\mu}_{ij} = \max\left\{0, [(\boldsymbol{\mu}_p)_i]_j\right\}$
    **end for**
  **end for**
**Normalizer**
  **for all** $1 \leq i \leq n$ **do**
    **if** at least one $\hat{\mu}_{ij} > 0$ **then**
      **for all** $1 \leq j \leq l$ **do**
        $\mu_{ij} = \dfrac{\hat{\mu}_{ij}}{\sqrt{\frac{1}{n} \sum_{s=1}^{n} \hat{\mu}_{sj}^2}}$
      **end for**
    **else**
      **for all** $1 \leq j \leq l$ **do**
        $\mu_{ij} = \begin{cases} \sqrt{n} & \text{for} \quad j = \arg\max_{\hat{j}}\{[(\boldsymbol{\mu}_p)_i]_{\hat{j}}\} \\ 0 & \text{otherwise} \end{cases}$
      **end for**
    **end if**
  **end for**
---

---
**Algorithm 5** Scaled Newton Projection
---
**Goal**
  perform a scaled Newton step with subsequent projection
**Input**
  for $1 \leq i \leq n$: $(\boldsymbol{\mu}_p)_i$
  for $1 \leq i \leq n$: $\boldsymbol{\mu}_i^{\text{old}}$
  simple projection P (rectified or rectified & normalized),
  $\lambda$ (gradient step size), $\gamma$ (projection difference)
**Main**
  $\boldsymbol{d} = \text{P}\left(\boldsymbol{\mu}_i^{\text{old}} + \lambda\left((\boldsymbol{\mu}_p)_i - \boldsymbol{\mu}_i^{\text{old}}\right)\right)$
  $\boldsymbol{\mu}_i^{\text{new}} = \text{P}\left(\boldsymbol{\mu}_i^{\text{old}} + \gamma\left(\boldsymbol{d} - \boldsymbol{\mu}_i^{\text{old}}\right)\right)$
---

---
**Algorithm 6** Scaled Projection With Reduced Matrix
---
**Goal**
  perform a scaled projection step with reduced matrix
**Input**
  for $1 \leq i \leq n$: $(\boldsymbol{\mu}_p)_i$
  for $1 \leq i \leq n$: $\boldsymbol{\mu}_i^{\text{old}}$
  simple projection P (rectified or rectified & normalized),
  $\lambda, \gamma, \boldsymbol{H}, \boldsymbol{\Sigma}_p^{-1}$
**Main**
  $\boldsymbol{d} = \text{P}\left(\boldsymbol{\mu}_i^{\text{old}} + \lambda\, \boldsymbol{H}^{-1}\, \boldsymbol{\Sigma}_p^{-1}((\boldsymbol{\mu}_p)_i - \boldsymbol{\mu}_i^{\text{old}})\right)$
  $\boldsymbol{\mu}_i^{\text{new}} = \text{P}\left(\boldsymbol{\mu}_i^{\text{old}} + \gamma\left(\boldsymbol{d} - \boldsymbol{\mu}_i^{\text{old}}\right)\right)$
---

---

**Algorithm 7** Weight Decay

---

**Input**
  Parameters $\boldsymbol{W}$
  Weight decay factors $\gamma_G$ (Gaussian) and $\gamma_L$ (Laplacian)
**Gaussian**
  $\boldsymbol{W} \;=\; \boldsymbol{W} \;-\; \gamma_G\,\boldsymbol{W}$
**Laplacian**
  $\hat{\boldsymbol{W}} \;=\; \mathrm{median}\{-\gamma_L\,,\;\boldsymbol{W}\,,\;\gamma_L\}$
  $\boldsymbol{W} \;=\; \boldsymbol{W} \;-\; \hat{\boldsymbol{W}}$

---

---

**Algorithm 8** Dropout

---

**Input**
  for $1 \le i \le n$: $\boldsymbol{\mu}_i$
  dropout probability $d$
**Main**
  **for all** $1 \le i \le n$ **do**
    **for all** $1 \le j \le l$ **do**
      $\Pr(\delta = 0) \;=\; d$
      $\mu_{ij} \;=\; \delta\,\mu_{ij}$
    **end for**
  **end for**

---

## 3   Convergence Proof for the RFN Learning Algorithm

**Theorem 1** (RFN Convergence). *The rectified factor network (RFN) learning algorithm given in Algorithm 1 is a "generalized alternating minimization" (GAM) algorithm and converges to a solution that maximizes the objective $\mathcal{F}$.*

*Proof.* The factor analysis EM algorithm is given by Eq. (67) and Eq. (68) in Section 5. Algorithm 1 is the factor analysis EM algorithm with modified the E-step and the M-step. The E-step is modified by constraining the variational distribution $Q$ to non-negative means and by normalizing its means across the samples. The M-step is modified to a Newton direction gradient step.

Like EM factor analysis, Algorithm 1 aims at maximizing the negative *free energy* $\mathcal{F}$, which is

$$
\begin{aligned}
\mathcal{F} \;&=\; \frac{1}{n}\,\sum_{i=1}^{n} \log p(\boldsymbol{v}_i) \;-\; \frac{1}{n}\,\sum_{i=1}^{n} D_{\mathrm{KL}}(Q(\boldsymbol{h}_i) \,\|\, p(\boldsymbol{h}_i \mid \boldsymbol{v}_i)) \qquad\qquad (1)\\
&=\; \frac{1}{n}\,\sum_{i=1}^{n} \int Q(\boldsymbol{h}_i)\,\log p(\boldsymbol{v}_i)\,d\boldsymbol{h}_i \;-\; \frac{1}{n}\,\sum_{i=1}^{n} \int Q(\boldsymbol{h}_i)\,\log \frac{Q(\boldsymbol{h}_i)}{p(\boldsymbol{h}_i \mid \boldsymbol{v}_i)}\,d\boldsymbol{h}_i\\
&=\; -\,\frac{1}{n}\,\sum_{i=1}^{n} \int Q(\boldsymbol{h}_i)\,\log \frac{Q(\boldsymbol{h}_i)}{p(\boldsymbol{h}_i, \boldsymbol{v}_i)}\,d\boldsymbol{h}_i\\
&=\; -\,\frac{1}{n}\,\sum_{i=1}^{n} \int Q(\boldsymbol{h}_i)\,\log \frac{Q(\boldsymbol{h}_i)}{p(\boldsymbol{h}_i)}\,d\boldsymbol{h}_i \;+\; \frac{1}{n}\,\sum_{i=1}^{n} \int Q(\boldsymbol{h}_i)\,\log p(\boldsymbol{v}_i \mid \boldsymbol{h}_i)\,d\boldsymbol{h}_i\\
&=\; \frac{1}{n}\,\sum_{i=1}^{n} \int Q(\boldsymbol{h}_i)\,\log p(\boldsymbol{v}_i \mid \boldsymbol{h}_i)\,d\boldsymbol{h}_i \;-\; \frac{1}{n}\,\sum_{i=1}^{n} D_{\mathrm{KL}}(Q(\boldsymbol{h}_i) \,\|\, p(\boldsymbol{h}_i))\,.
\end{aligned}
$$

$D_{\mathrm{KL}}$ denotes the Kullback-Leibler (KL) divergence Kullback and Leibler [1951], which is larger than or equal to zero.

Algorithm 1 decreases $\frac{1}{n}\sum_{i=1}^{n} D_{\mathrm{KL}}(Q(\boldsymbol{h}_i) \,\|\, p(\boldsymbol{h}_i \mid \boldsymbol{v}_i))$ (the E-step objective) in its E-step under constraints for non-negative means and normalization. The constraint optimization problem from

Section 9.2 for the E-step is

$$\min_{Q(\boldsymbol{h}_i)} \quad \frac{1}{n} \sum_{i=1}^{n} D_{\mathrm{KL}}(Q(\boldsymbol{h}_i) \parallel p(\boldsymbol{h}_i \mid \boldsymbol{v}_i)) \tag{2}$$

$$\text{s.t.} \quad \forall_i : \boldsymbol{\mu}_i \geq \boldsymbol{0} \,,$$

$$\forall_j : \frac{1}{n} \sum_{i=1}^{n} \mu_{ij}^2 = 1 \,.$$

The M-step of Algorithm 1 aims at decreasing

$$\mathcal{E} = -\frac{1}{n} \sum_{i=1}^{n} \int_{\mathbb{R}^l} Q(\boldsymbol{h}_i) \, \log\left(p(\boldsymbol{v}_i \mid \boldsymbol{h}_i)\right) \, d\boldsymbol{h}_i \,. \tag{3}$$

Algorithm 1 performs one gradient descent step in the Newton direction to decrease $\mathcal{E}$, while EM factor analysis minimizes $\mathcal{E}$.

From the modification of the E-step and the M-step follows that Algorithm 1 is a *Generalized Alternating Minimization (GAM)* algorithm according to Gunawardana and Byrne [2005]. GAM is an EM algorithm that increases $\mathcal{F}$ in the E-step and increases $\mathcal{F}$ in the M-step (see also Section 7). The most important requirements for the convergence of the GAM algorithm according to Theorem 4 (Proposition 5 in Gunawardana and Byrne [2005]) are the increase of the objective $\mathcal{F}$ in both the E-step and the M-step. Therefore we first show these two decreases before showing that all requirements of convergence Theorem 4 are met.

**Algorithm 1 ensures to decrease the M-step objective.** The M-step objective $\mathcal{E}$ is convex in $\boldsymbol{W}$ and $\boldsymbol{\Psi}^{-1}$ according to Theorem 5 and Theorem 7. The update with $\eta_W = \eta_\Psi = \eta = 1$ leads to the minimum of $\mathcal{E}$ according to Theorem 5 and Theorem 7. The convexity of $\mathcal{E}$ guarantees that each update with $0 < \eta_W = \eta_\Psi = \eta \leq 1$ decreases the M-step objective $\mathcal{E}$, except the current $\boldsymbol{W}$ and $\boldsymbol{\Psi}^{-1}$ are already the minimizers.

**Algorithm 1 ensures to decrease the E-step objective.** The E-step decrease of Algorithm 1 is performed by Algorithm 2. According to Theorem 11 the scaled projection with reduced matrix ensures a decrease of the E-step objective for rectifying constraints (convex feasible set). According to Theorem 10 also gradient projection methods ensure a decrease of the E-step objective for rectifying constraints. For rectifying constraints and normalization, the feasible set is not convex because of the equality constraints. To optimize such problems, the generalized reduced gradient method Abadie and Carpentier [1969] solves each equality constraint for one variable and inserts it into the objective. For our problem Eq. (146) gives the solution and Eq. (147) the resulting convex constraints. Now scaled projection and gradient projection methods can be applied. For rectifying and normalizing constraints, also Rosen's Rosen [1961] and Haug & Arora's Haug and Arora [1979] gradient projection method ensures a decrease of the E-step objective since they can be applied to non-convex problems.

We show that the requirements as given in Section 7 for GAM convergence according to Theorem 4 (Proposition 5 in Gunawardana and Byrne [2005]) are fulfilled:

1. the learning rules, that is, the E-step and the M-step, are closed maps $\longrightarrow$ ensured by continuous and continuous differentiable maps,

2. the parameter set is compact $\longrightarrow$ ensured by bounding $\boldsymbol{\Psi}$ and $\boldsymbol{W}$,

3. the family of variational distributions is compact (often described by the feasible set of parameters of the variational distributions) $\longrightarrow$ ensured by continuous and continuous differentiable functions for the constraints and by the bounds on the variational parameters $\boldsymbol{\mu}$ and $\boldsymbol{\Sigma}$ determined by bounds on the parameters and the data,

4. the support of the density models does not depend on the parameter $\longrightarrow$ ensured by Gaussian models with full-rank covariance matrix,

5. the density models are continuous in the parameters $\longrightarrow$ ensured by Gaussian models

6. the E-step has a unique maximizer $\longrightarrow$ ensured by the convex, continuous, and continuous differentiable function that is minimized Dredze et al. [2008, 2012] together with compact feasible set for the variational parameters, the maximum may be local for non-convex feasible sets stemming from normalization,

7. the E-step increases the objective if not at the maximizer $\longrightarrow$ ensured as shown above,

8. the M-step has a unique maximizer (this is not required) $\longrightarrow$ ensured by minimizing a convex, continuous and continuous differentiable function in the model parameter and a convex feasible set, the maximum is a global maximum,

9. the M-step increases the objective if not at the maximizer $\longrightarrow$ ensured as shown above.

$\square$

Since this Proposition 5 in Gunawardana and Byrne [2005] is based on Zangwill's generalized convergence theorem, updates of the RFN algorithm are viewed as point-to-set mappings Zangwill [1969]. Therefore the numerical precision, the choice of the methods in the E-step, and GPU implementations are covered by the proof. That the M-step has a unique maximizer is not required to proof Theorem 1 by Theorem 4. However we obtain an alternative proof by exchanging the variational distribution $Q$ and the parameters $(\boldsymbol{W}, \boldsymbol{\Psi})$, that is, exchanging the E-step and the M-step. A theorem analog to Theorem 4 but with E-step and M-step conditions exchanged can be derived from Zangwill's generalized convergence theorem Zangwill [1969].

The resulting model from the GAM procedure is at a local maximum of the objective given the model family and the family of variational distributions. *The solution minimizes the KL-distance between the family of full variational distributions and full model family.* "Full" means that both the observed and the hidden variables are taken into account, where for the variational distributions the probability of the observations is set to 1. The *desired family* is defined as the set of all probability distributions that assign probability one to the observation. In our case the family of variational distributions is not the desired family since some distributions are excluded by the constraints. Therefore the solution of the GAM optimization does not guarantee stationary points in likelihood Gunawardana and Byrne [2005]. This means that we do not maximize the likelihood but minimize

$$- \mathcal{F} \approx D_{\mathrm{KL}}(Q(\boldsymbol{h}, \boldsymbol{v}) \parallel p(\boldsymbol{h}, \boldsymbol{v})) + c \tag{4}$$

according to Eq. (73), where $c$ is a constant independent of $Q$ and independent of the model parameters.

## 4 Correctness Proofs for the RFN Learning Algorithms

The RFN algorithm is correct if it has a low reconstruction error and explains the data covariance matrix by its parameters like factor analysis. We show in Theorem 2 and Theorem 3 that the RFN algorithm

1. minimizes the reconstruction error given $\boldsymbol{\mu}_i$ and $\boldsymbol{\Sigma}$ (the error is quadratic in $\boldsymbol{\Psi}$);

2. explains the covariance matrix by its parameters $\boldsymbol{W}$ and $\boldsymbol{\Psi}$ plus an estimate of the second moment of the coding units $\boldsymbol{S}$.

Since the minimization of the reconstruction error is based on $\boldsymbol{\mu}_i$, the quality of reconstruction and covariance explanation depends on the correlation between $\boldsymbol{\mu}_i$ and $\boldsymbol{v}_i$. The larger the correlation between $\boldsymbol{\mu}_i$ and $\boldsymbol{v}_i$, the lower the reconstruction error and the better the explanation of the data covariance. We ensure maximal information in $\boldsymbol{\mu}_i$ on $\boldsymbol{v}_i$ by the I-projection (the minimal Kullback-Leibler distance) of the posterior onto the family of rectified and normalized Gaussian distributions.

The reconstruction error for given mean values $\boldsymbol{\mu}_i$ is

$$\frac{1}{n} \sum_{i=1}^{n} \|\boldsymbol{\epsilon}_i\|_2^2 \, , \tag{5}$$

where

$$\boldsymbol{\epsilon}_i \;=\; \boldsymbol{v}_i \;-\; \boldsymbol{W}\,\boldsymbol{\mu}_i \,. \tag{6}$$

The reconstruction error for using the whole variational distribution $Q(\boldsymbol{h}_i)$ instead of its means is $\boldsymbol{\Psi}$. Below we will derive Eq. (17), which is

$$\boldsymbol{\Psi} \;=\; \mathrm{diag}\left(\frac{1}{n}\sum_{i=1}^{n}\boldsymbol{\epsilon}_i\,\boldsymbol{\epsilon}_i^T \;+\; \boldsymbol{W}\,\boldsymbol{\Sigma}\,\boldsymbol{W}^T\right)\,. \tag{7}$$

Therefore $\boldsymbol{\Psi}$ is the reconstruction error for given mean values plus the variance $\boldsymbol{W}\boldsymbol{\Sigma}\boldsymbol{W}^T$ introduced by the hidden variables.

## 4.1 Diagonal Noise Covariance Update

**Theorem 2** (RFN Correctness: Diagonal Noise Covariance Update). *The fixed point $\boldsymbol{W}$ minimizes* $\mathrm{Tr}\left(\boldsymbol{\Psi}\right)$ *given $\boldsymbol{\mu}_i$ and $\boldsymbol{\Sigma}$ by ridge regression with*

$$\mathrm{Tr}\left(\boldsymbol{\Psi}\right) \;=\; \frac{1}{n}\sum_{i=1}^{n}\|\boldsymbol{\epsilon}_i\|_2^2 \;+\; \left\|\boldsymbol{W}\,\boldsymbol{\Sigma}^{1/2}\right\|_{\mathrm{F}}^2 \,, \tag{8}$$

*where we used the error*

$$\boldsymbol{\epsilon}_i \;=\; \boldsymbol{v}_i \;-\; \boldsymbol{W}\,\boldsymbol{\mu}_i \tag{9}$$

*The model explains the data covariance matrix by*

$$\boldsymbol{C} \;=\; \boldsymbol{\Psi} \;+\; \boldsymbol{W}\,\boldsymbol{S}\,\boldsymbol{W}^T \tag{10}$$

*up to an error, which is quadratic in $\boldsymbol{\Psi}$ for $\boldsymbol{\Psi} \ll \boldsymbol{W}\boldsymbol{W}^T$. The reconstruction error*

$$\frac{1}{n}\sum_{i=1}^{n}\|\boldsymbol{\epsilon}_i\|_2^2 \tag{11}$$

*is quadratic in $\boldsymbol{\Psi}$ for $\boldsymbol{\Psi} \ll \boldsymbol{W}\boldsymbol{W}^T$.*

*Proof.* The fixed point equation for the $\boldsymbol{W}$ update is

$$\Delta\boldsymbol{W} \;=\; \boldsymbol{U}\,\boldsymbol{S}^{-1} \;-\; \boldsymbol{W} \;=\; \boldsymbol{0} \;\Rightarrow\; \boldsymbol{W} \;=\; \boldsymbol{U}\,\boldsymbol{S}^{-1}\,. \tag{12}$$

Using the definition of $\boldsymbol{U}$ and $\boldsymbol{S}$, the fixed point equation Eq. (12) gives

$$\boldsymbol{W} \;=\; \left(\frac{1}{n}\sum_{i=1}^{n}\boldsymbol{v}_i\,\boldsymbol{\mu}_i^T\right)\left(\frac{1}{n}\sum_{i=1}^{n}\boldsymbol{\mu}_i\,\boldsymbol{\mu}_i^T \;+\; \boldsymbol{\Sigma}\right)^{-1} \tag{13}$$

Therefore $\boldsymbol{W}$ is a *ridge regression* estimate, also called *generalized Tikhonov regularization* estimate, which minimizes

$$\frac{1}{n}\sum_{i=1}^{n}\|\boldsymbol{v}_i \;-\; \boldsymbol{W}\,\boldsymbol{\mu}_i\|_2^2 \;+\; \left\|\boldsymbol{W}\,\boldsymbol{\Sigma}^{1/2}\right\|_{\mathrm{F}}^2 \tag{14}$$

$$= \; \frac{1}{n}\sum_{i=1}^{n}\|\boldsymbol{\epsilon}_i\|_2^2 \;+\; \left\|\boldsymbol{W}\,\boldsymbol{\Sigma}^{1/2}\right\|_{\mathrm{F}}^2$$

$$= \; \frac{1}{n}\sum_{i=1}^{n}\boldsymbol{\epsilon}_i^T\,\boldsymbol{\epsilon}_i \;+\; \mathrm{Tr}\left(\boldsymbol{W}\,\boldsymbol{\Sigma}^{1/2}\,\boldsymbol{\Sigma}^{1/2}\boldsymbol{W}^T\right)$$

$$= \; \mathrm{Tr}\left(\frac{1}{n}\sum_{i=1}^{n}\boldsymbol{\epsilon}_i\,\boldsymbol{\epsilon}_i^T \;+\; \boldsymbol{W}\,\boldsymbol{\Sigma}\,\boldsymbol{W}^T\right)\,,$$

where we used the reconstruction error

$$\boldsymbol{\epsilon}_i \;=\; \boldsymbol{v}_i \;-\; \boldsymbol{W}\,\boldsymbol{\mu}_i\,. \tag{15}$$

We obtain with this definition of the error

$$\frac{1}{n} \sum_{i=1}^{n} \boldsymbol{\epsilon}_i \, \boldsymbol{\epsilon}_i^T \; + \; \boldsymbol{W} \, \boldsymbol{\Sigma} \, \boldsymbol{W}^T \tag{16}$$

$$= \; \frac{1}{n} \sum_{i=1}^{n} \boldsymbol{v}_i \, \boldsymbol{v}_i^T \; - \; \frac{1}{n} \sum_{i=1}^{n} \boldsymbol{v}_i \, \boldsymbol{\mu}_i^T \, \boldsymbol{W}^T \; - \; \frac{1}{n} \sum_{i=1}^{n} \boldsymbol{W} \, \boldsymbol{\mu}_i \, \boldsymbol{v}_i^T$$

$$+ \; \frac{1}{n} \sum_{i=1}^{n} \boldsymbol{W} \, \boldsymbol{\mu}_i \, \boldsymbol{\mu}_i^T \, \boldsymbol{W}^T \; + \; \boldsymbol{W} \, \boldsymbol{\Sigma} \, \boldsymbol{W}^T$$

$$= \; \boldsymbol{C} \; - \; \boldsymbol{U} \, \boldsymbol{W}^T \; - \; \boldsymbol{W} \, \boldsymbol{U}^T \; + \; \boldsymbol{W} \, \boldsymbol{S} \, \boldsymbol{W}^T \, .$$

Therefore from the fixed point equation for $\boldsymbol{\Psi}$ with the diagonal update rule follows

$$\boldsymbol{\Psi} \; = \; \mathrm{diag} \left( \frac{1}{n} \sum_{i=1}^{n} \boldsymbol{\epsilon}_i \, \boldsymbol{\epsilon}_i^T \; + \; \boldsymbol{W} \, \boldsymbol{\Sigma} \, \boldsymbol{W}^T \right) , \tag{17}$$

where "diag" projects a matrix to a diagonal matrix. From this follows that

$$\mathrm{Tr} \left( \boldsymbol{\Psi} \right) \; = \; \mathrm{Tr} \left( \frac{1}{n} \sum_{i=1}^{n} \boldsymbol{\epsilon}_i \, \boldsymbol{\epsilon}_i^T \; + \; \boldsymbol{W} \, \boldsymbol{\Sigma} \, \boldsymbol{W}^T \right) . \tag{18}$$

Consequently, the fixed point $\boldsymbol{W}$ minimizes $\mathrm{Tr} \left( \boldsymbol{\Psi} \right)$ given $\boldsymbol{\mu}_i$ and $\boldsymbol{\Sigma}$.

After convergence of the algorithm $\boldsymbol{\Sigma} \; = \; \left( \boldsymbol{I} + \boldsymbol{W}^T \boldsymbol{\Psi}^{-1} \boldsymbol{W} \right)^{-1}$ holds. The Woodbury identity (matrix inversion lemma) states

$$\left( \boldsymbol{W} \, \boldsymbol{W}^T \; + \; \boldsymbol{\Psi} \right)^{-1} \; = \; \boldsymbol{\Psi}^{-1} \; - \; \boldsymbol{\Psi}^{-1} \boldsymbol{W} \left( \boldsymbol{I} \; + \; \boldsymbol{W}^T \boldsymbol{\Psi}^{-1} \boldsymbol{W} \right)^{-1} \boldsymbol{W}^T \boldsymbol{\Psi}^{-1} \tag{19}$$

from which follows by multiplying the equation from right and left by $\boldsymbol{\Psi}$ that

$$\boldsymbol{W} \, \boldsymbol{\Sigma} \, \boldsymbol{W}^T \; = \; \boldsymbol{W} \left( \boldsymbol{I} \; + \; \boldsymbol{W}^T \boldsymbol{\Psi}^{-1} \boldsymbol{W} \right)^{-1} \boldsymbol{W}^T \tag{20}$$

$$= \; \boldsymbol{\Psi} \; - \; \boldsymbol{\Psi} \left( \boldsymbol{W} \, \boldsymbol{W}^T \; + \; \boldsymbol{\Psi} \right)^{-1} \boldsymbol{\Psi}$$

Inserting this equation Eq. (20) into Eq. (17) gives

$$\boldsymbol{\Psi} \; = \; \mathrm{diag} \left( \frac{1}{n} \sum_{i=1}^{n} \boldsymbol{\epsilon}_i \, \boldsymbol{\epsilon}_i^T \; + \; \boldsymbol{\Psi} \; - \; \boldsymbol{\Psi} \left( \boldsymbol{W} \, \boldsymbol{W}^T \; + \; \boldsymbol{\Psi} \right)^{-1} \boldsymbol{\Psi} \right) \tag{21}$$

$$= \; \boldsymbol{\Psi} \; + \; \mathrm{diag} \left( \frac{1}{n} \sum_{i=1}^{n} \boldsymbol{\epsilon}_i \, \boldsymbol{\epsilon}_i^T \; - \; \boldsymbol{\Psi} \left( \boldsymbol{W} \, \boldsymbol{W}^T \; + \; \boldsymbol{\Psi} \right)^{-1} \boldsymbol{\Psi} \right) .$$

Therefore we have

$$\mathrm{diag} \left( \frac{1}{n} \sum_{i=1}^{n} \boldsymbol{\epsilon}_i \, \boldsymbol{\epsilon}_i^T \; - \; \boldsymbol{\Psi} \left( \boldsymbol{W} \, \boldsymbol{W}^T \; + \; \boldsymbol{\Psi} \right)^{-1} \boldsymbol{\Psi} \right) \; = \; \boldsymbol{0} \, . \tag{22}$$

It follows that

$$\mathrm{Tr} \left( \frac{1}{n} \sum_{i=1}^{n} \boldsymbol{\epsilon}_i \, \boldsymbol{\epsilon}_i^T \right) \; = \; \mathrm{Tr} \left( \boldsymbol{\Psi} \left( \boldsymbol{W} \, \boldsymbol{W}^T \; + \; \boldsymbol{\Psi} \right)^{-1} \boldsymbol{\Psi} \right)$$

$$\leq \; \mathrm{Tr} \left( \left( \boldsymbol{W} \, \boldsymbol{W}^T \; + \; \boldsymbol{\Psi} \right)^{-1} \right) \mathrm{Tr} \left( \boldsymbol{\Psi} \right)^2 \, . \tag{23}$$

The inequality uses the fact that for positive definite matrices $\boldsymbol{A}$ and $\boldsymbol{B}$ inequality $\mathrm{Tr}(\boldsymbol{AB}) \leq \mathrm{Tr}(\boldsymbol{A})\mathrm{Tr}(\boldsymbol{B})$ holds Patel and Toda [1979]. Thus, for $\boldsymbol{\Psi} \ll \boldsymbol{W}\boldsymbol{W}^T$ the error $\mathrm{Tr} \left( \frac{1}{n} \sum_{i=1}^{n} \boldsymbol{\epsilon}_i \boldsymbol{\epsilon}_i^T \right) = \frac{1}{n} \sum_{i=1}^{n} \boldsymbol{\epsilon}_i^T \boldsymbol{\epsilon}_i$ is quadratic in $\boldsymbol{\Psi}$.

Multiplying the fixed point equation Eq. (12) by $\boldsymbol{S}$ gives $\boldsymbol{U} = \boldsymbol{W}\boldsymbol{S}$. Therefore we have:

$$\boldsymbol{W} \, \boldsymbol{U}^T \; = \; \boldsymbol{W} \, \boldsymbol{S} \, \boldsymbol{W}^T \; = \; \boldsymbol{U} \, \boldsymbol{W}^T \, . \tag{24}$$

Inserting Eq. (20) into the first line of Eq. (16) and Eq. (24) for simplifying the last line of Eq. (16) gives

$$\frac{1}{n} \sum_{i=1}^{n} \boldsymbol{\epsilon}_i \, \boldsymbol{\epsilon}_i^T \; - \; \boldsymbol{\Psi} \, \left( \boldsymbol{W} \, \boldsymbol{W}^T \; + \; \boldsymbol{\Psi} \right)^{-1} \, \boldsymbol{\Psi} \; = \; \boldsymbol{C} \; - \; \boldsymbol{\Psi} \; - \; \boldsymbol{W} \, \boldsymbol{S} \, \boldsymbol{W}^T \, . \tag{25}$$

Using the trace norm (nuclear norm or Ky-Fan n-norm) on matrices, Eq. (23) states that the left hand side is quadratic in $\boldsymbol{\Psi}$ for $\boldsymbol{\Psi} \ll \boldsymbol{W}\boldsymbol{W}^T$. The trace norm of a positive semi-definite matrix is its trace and bounds the Frobenius norm Srebro [2004]. Furthermore, Eq. (22) states that the left hand side of this equation has zero diagonal entries. Therfore it follows that

$$\boldsymbol{C} \; = \; \boldsymbol{\Psi} \; + \; \boldsymbol{W} \, \boldsymbol{S} \, \boldsymbol{W}^T \tag{26}$$

holds except an error, which is quadratic in $\boldsymbol{\Psi}$ for $\boldsymbol{\Psi} \ll \boldsymbol{W}\boldsymbol{W}^T$. The diagonal is exactly modeled according to Eq. (22). □

Therefore the model corresponding to the fixed point explains the empirical matrix of second moments $\boldsymbol{C}$ by a noise part $\boldsymbol{\Psi}$ and a signal part $\boldsymbol{W}\boldsymbol{S}\boldsymbol{W}^T$. Like factor analysis the data variance is explained by the model via the parameters $\boldsymbol{\Psi}$ (noise) and $\boldsymbol{W}$ (signal).

### 4.2 Full Noise Covariance Update

**Theorem 3** (RFN Correctness: Full Noise Covariance Update). *The fixed point $\boldsymbol{W}$ minimizes* $\mathrm{Tr}\left(\boldsymbol{\Psi}\right)$ *given $\boldsymbol{\mu}_i$ and $\boldsymbol{\Sigma}$ by ridge regression with*

$$\mathrm{Tr}\left(\boldsymbol{\Psi}\right) \; = \; \frac{1}{n} \sum_{i=1}^{n} \|\boldsymbol{\epsilon}_i\|_2^2 \; + \; \left\| \boldsymbol{W} \, \boldsymbol{\Sigma}^{1/2} \right\|_{\mathrm{F}}^2 \, , \tag{27}$$

*where we used the error*

$$\boldsymbol{\epsilon}_i \; = \; \boldsymbol{v}_i \; - \; \boldsymbol{W} \, \boldsymbol{\mu}_i \tag{28}$$

*The model explains the data covariance matrix by*

$$\boldsymbol{C} \; = \; \boldsymbol{\Psi} \; + \; \boldsymbol{W} \, \boldsymbol{S} \, \boldsymbol{W}^T \, . \tag{29}$$

*The reconstruction error*

$$\frac{1}{n} \sum_{i=1}^{n} \|\boldsymbol{\epsilon}_i\|_2^2 \tag{30}$$

*is quadratic in $\boldsymbol{\Psi}$ for $\boldsymbol{\Psi} \ll \boldsymbol{W}\boldsymbol{W}^T$.*

*Proof.* The first part follows from previous Theorem 2. The fixed point equation for the $\boldsymbol{\Psi}$ update is

$$\boldsymbol{\Psi} \; = \; \boldsymbol{C} \; - \; \boldsymbol{U} \, \boldsymbol{W}^T \; - \; \boldsymbol{W} \, \boldsymbol{U}^T \; + \; \boldsymbol{W} \, \boldsymbol{S} \, \boldsymbol{W}^T \, , \tag{31}$$

using Eq. (24) this leads to

$$\boldsymbol{C} \; = \; \boldsymbol{\Psi} \; + \; \boldsymbol{W} \, \boldsymbol{S} \, \boldsymbol{W}^T \, . \tag{32}$$

From Eq. (16) follows for the fixed point of $\boldsymbol{\Psi}$ with the full update rule:

$$\boldsymbol{\Psi} \; = \; \frac{1}{n} \sum_{i=1}^{n} \boldsymbol{\epsilon}_i \, \boldsymbol{\epsilon}_i^T \; + \; \boldsymbol{W} \, \boldsymbol{\Sigma} \, \boldsymbol{W}^T \, . \tag{33}$$

Inserting Eq. (20) into Eq. (33) gives

$$\boldsymbol{\Psi} \; = \; \frac{1}{n} \sum_{i=1}^{n} \boldsymbol{\epsilon}_i \, \boldsymbol{\epsilon}_i^T \; + \; \boldsymbol{\Psi} \; - \; \boldsymbol{\Psi} \, \left( \boldsymbol{W} \, \boldsymbol{W}^T \; + \; \boldsymbol{\Psi} \right)^{-1} \, \boldsymbol{\Psi} \, , \tag{34}$$

from which follows

$$\frac{1}{n} \sum_{i=1}^{n} \boldsymbol{\epsilon}_i \, \boldsymbol{\epsilon}_i^T \; = \; \boldsymbol{\Psi} \, \left( \boldsymbol{W} \, \boldsymbol{W}^T \; + \; \boldsymbol{\Psi} \right)^{-1} \, \boldsymbol{\Psi} \, . \tag{35}$$

Thus, the error $\mathrm{Tr}\left(\frac{1}{n} \sum_{i=1}^{n} \boldsymbol{\epsilon}_i \boldsymbol{\epsilon}_i^T\right) = \frac{1}{n} \sum_{i=1}^{n} \boldsymbol{\epsilon}_i^T \boldsymbol{\epsilon}_i$ is quadratic in $\boldsymbol{\Psi}$, for $\boldsymbol{\Psi} \ll \boldsymbol{W}\boldsymbol{W}^T$. □

# 5  Maximum Likelihood Factor Analysis

We are given the data $\{\boldsymbol{v}\} = \{\boldsymbol{v}_1, \ldots, \boldsymbol{v}_n\}$ which is assumed to be centered. Centering can be done by subtracting the mean $\boldsymbol{\mu}$ from the data. The model is

$$\boldsymbol{v} \;=\; \boldsymbol{W}\boldsymbol{h} \;+\; \boldsymbol{\epsilon}\,, \tag{36}$$

where

$$\boldsymbol{h} \sim \mathcal{N}\left(\boldsymbol{0}, \boldsymbol{I}\right) \quad\text{and}\quad \boldsymbol{\epsilon} \sim \mathcal{N}\left(\boldsymbol{0}, \boldsymbol{\Psi}\right)\,. \tag{37}$$

The model includes the *observations* $\boldsymbol{v} \in \mathbb{R}^m$, the *noise* $\boldsymbol{\epsilon} \in \mathbb{R}^m$, the *factors* $\boldsymbol{h} \in \mathbb{R}^l$, the *factor loading matrix* $\boldsymbol{W} \in \mathbb{R}^{m \times l}$, and the *noise covariance matrix* $\boldsymbol{\Psi} \in \mathbb{R}^{m \times m}$. Typically we assume that $\boldsymbol{\Psi}$ is a diagonal matrix to explain data covariance by signal and not by noise. The data variance is explained through a signal part $\boldsymbol{W}\boldsymbol{h}$ and through a noise part $\boldsymbol{\epsilon}$. The parameters of the model are $\boldsymbol{W}$ and $\boldsymbol{\Psi}$. From the model assumption it follows that if $\boldsymbol{h}$ is given, then only the noise $\boldsymbol{\epsilon}$ is a random variable and we have

$$\boldsymbol{v} \mid \boldsymbol{h} \sim \mathcal{N}\left(\boldsymbol{W}\boldsymbol{h}, \boldsymbol{\Psi}\right)\,. \tag{38}$$

We want to derive the *likelihood* of the data under the model, that is, the likelihood that the model has produced the data. Let E denote the expectation of the data including the prior distribution of the factors and the noise distribution. We obtain for the first two moments and the variance:

$$\mathrm{E}(\boldsymbol{v}) \;=\; \mathrm{E}(\boldsymbol{W}\boldsymbol{h} \;+\; \boldsymbol{\epsilon}) \;=\; \boldsymbol{W}\mathrm{E}(\boldsymbol{h}) \;+\; \mathrm{E}(\boldsymbol{\epsilon}) \;=\; \boldsymbol{0}\,, \tag{39}$$

$$\begin{aligned}
\mathrm{E}\left(\boldsymbol{v}\,\boldsymbol{v}^T\right) \;&=\; \mathrm{E}\left((\boldsymbol{W}\boldsymbol{h} \;+\; \boldsymbol{\epsilon})(\boldsymbol{W}\boldsymbol{h} \;+\; \boldsymbol{\epsilon})^T\right) \;= \\
&\quad\; \boldsymbol{W}\mathrm{E}\left(\boldsymbol{h}\,\boldsymbol{h}^T\right)\boldsymbol{W}^T \;+\; \boldsymbol{W}\mathrm{E}(\boldsymbol{h})\,\mathrm{E}\left(\boldsymbol{\epsilon}^T\right) \\
&\quad\; +\; \mathrm{E}(\boldsymbol{\epsilon})\,\mathrm{E}\left(\boldsymbol{h}^T\right)\boldsymbol{W}^T \;+\; \mathrm{E}\left(\boldsymbol{\epsilon}\,\boldsymbol{\epsilon}^T\right) \;= \\
&\quad\; \boldsymbol{W}\,\boldsymbol{W}^T \;+\; \boldsymbol{\Psi}
\end{aligned}$$

$$\mathrm{var}(\boldsymbol{v}) \;=\; \mathrm{E}\left(\boldsymbol{v}\,\boldsymbol{v}^T\right) \;-\; (\mathrm{E}(\boldsymbol{v}))^2 \;=\; \boldsymbol{W}\,\boldsymbol{W}^T \;+\; \boldsymbol{\Psi}\,. \tag{40}$$

The observations are Gaussian distributed since their distribution is the product of two Gaussian densities divided by a normalizing constant. Therefore, the marginal distribution for $\boldsymbol{v}$ is

$$\boldsymbol{v} \sim \mathcal{N}\left(\boldsymbol{0}\,,\; \boldsymbol{W}\boldsymbol{W}^T \;+\; \boldsymbol{\Psi}\right)\,. \tag{41}$$

The log-likelihood $\log \prod_{i=1}^{n} p(\boldsymbol{v}_i)$ of the data $\{\boldsymbol{v}\}$ under the model $(\boldsymbol{W}, \boldsymbol{\Psi})$ is

$$\begin{aligned}
\log \prod_{i=1}^{n} p(\boldsymbol{v}_i) \;=\;& \log \prod_{i=1}^{n} (2\pi)^{-m/2} \left|\boldsymbol{W}\boldsymbol{W}^T \;+\; \boldsymbol{\Psi}\right|^{-1/2} \tag{42} \\
& \exp\left(-\frac{1}{2}\left(\boldsymbol{v}_i^T \left(\boldsymbol{W}\boldsymbol{W}^T \;+\; \boldsymbol{\Psi}\right)^{-1} \boldsymbol{v}_i\right)\right) \\
=\;& -\frac{n\,m}{2}\,\log\left(2\pi\right) \;-\; \frac{n}{2}\,\log\left|\boldsymbol{W}\boldsymbol{W}^T \;+\; \boldsymbol{\Psi}\right| \\
& -\frac{1}{2}\sum_{i=1}^{n} \boldsymbol{v}_i^T \left(\boldsymbol{W}\boldsymbol{W}^T \;+\; \boldsymbol{\Psi}\right)^{-1} \boldsymbol{v}_i\,,
\end{aligned}$$

where $|.|$ denotes the absolute value of the determinant of a matrix.

To maximize the likelihood is difficult since a closed form for the maximum does not exists. Therefore, typically the expectation maximization (EM) algorithm is used to maximize the likelihood. For the EM algorithm a variational distribution $Q$ is required which estimates the factors given the observations.

We consider a single data vector $\boldsymbol{v}_i$. The posterior is also Gaussian with mean $(\boldsymbol{\mu}_p)_i$ and covariance matrix $\boldsymbol{\Sigma}_p$:

$$\begin{aligned}
\boldsymbol{h}_i \mid \boldsymbol{v}_i \;&\sim\; \mathcal{N}\left((\boldsymbol{\mu}_p)_i, \boldsymbol{\Sigma}_p\right) \tag{43} \\
(\boldsymbol{\mu}_p)_i \;&=\; \boldsymbol{W}^T \left(\boldsymbol{W}\,\boldsymbol{W}^T \;+\; \boldsymbol{\Psi}\right)^{-1} \boldsymbol{v}_i \\
\boldsymbol{\Sigma}_p \;&=\; \boldsymbol{I} \;-\; \boldsymbol{W}^T \left(\boldsymbol{W}\,\boldsymbol{W}^T \;+\; \boldsymbol{\Psi}\right)^{-1} \boldsymbol{W}\,,
\end{aligned}$$

where we used the fact that

$$\boldsymbol{a} \sim \mathcal{N}(\boldsymbol{\mu}_a, \Sigma_{aa}) , \ \boldsymbol{u} \sim \mathcal{N}(\boldsymbol{\mu}_u, \Sigma_{uu}) , \tag{44}$$

$$\Sigma_{ua} = \mathrm{Cov}(\boldsymbol{u}, \boldsymbol{a}) \text{ and } \Sigma_{au} = \mathrm{Cov}(\boldsymbol{a}, \boldsymbol{u}) :$$

$$\boldsymbol{a} \mid \boldsymbol{u} \sim \mathcal{N}\left(\boldsymbol{\mu}_a + \Sigma_{au}\Sigma_{uu}^{-1}(\boldsymbol{u} - \boldsymbol{\mu}_u) , \ \Sigma_{aa} - \Sigma_{au}\Sigma_{uu}^{-1}\Sigma_{ua}\right)$$

and

$$\mathrm{E}(\boldsymbol{h}\boldsymbol{v}) = \boldsymbol{W}\,\mathrm{E}(\boldsymbol{h}\,\boldsymbol{h}^T) = \boldsymbol{W} . \tag{45}$$

The EM algorithm sets $Q$ to the posterior distribution for data vector $\boldsymbol{v}_i$:

$$Q_i(\boldsymbol{h}_i) = p(\boldsymbol{h}_i \mid \boldsymbol{v}_i; \boldsymbol{W}, \boldsymbol{\Psi}) = \mathcal{N}((\boldsymbol{\mu}_p)_i, \boldsymbol{\Sigma}_p) , \tag{46}$$

therefore we obtain for standared EM

$$\boldsymbol{\mu}_i = (\boldsymbol{\mu}_q)_i = (\boldsymbol{\mu}_p)_i \tag{47}$$

$$\boldsymbol{\Sigma} = \boldsymbol{\Sigma}_q = \boldsymbol{\Sigma}_p . \tag{48}$$

The matrix inversion lemma (Woodbury identiy) can be used to compute $\boldsymbol{\mu}_i$ and $\boldsymbol{\Sigma}$:

$$\left(\boldsymbol{W}\,\boldsymbol{W}^T + \boldsymbol{\Psi}\right)^{-1} = \boldsymbol{\Psi}^{-1} - \boldsymbol{\Psi}^{-1}\boldsymbol{W}\left(\boldsymbol{I} + \boldsymbol{W}^T\boldsymbol{\Psi}^{-1}\boldsymbol{W}\right)^{-1}\boldsymbol{W}^T\boldsymbol{\Psi}^{-1} . \tag{49}$$

Using this identity, the mean and the covariance matrix can be computed as:

$$\boldsymbol{\mu}_i = \boldsymbol{W}^T\left(\boldsymbol{W}\,\boldsymbol{W}^T + \boldsymbol{\Psi}\right)^{-1}\boldsymbol{v}_i = \left(\boldsymbol{I} + \boldsymbol{W}^T\boldsymbol{\Psi}^{-1}\boldsymbol{W}\right)^{-1}\boldsymbol{W}^T\boldsymbol{\Psi}^{-1}\boldsymbol{v}_i , \tag{50}$$

$$\boldsymbol{\Sigma} = \boldsymbol{I} - \boldsymbol{W}^T\left(\boldsymbol{W}\,\boldsymbol{W}^T + \boldsymbol{\Psi}\right)^{-1}\boldsymbol{W} = \left(\boldsymbol{I} + \boldsymbol{W}^T\boldsymbol{\Psi}^{-1}\boldsymbol{W}\right)^{-1} .$$

The EM algorithm maximizes a lower bound $\mathcal{F}$ on the log-likelihood:

$$\mathcal{F} = \log p(\boldsymbol{v}_i) - D_{\mathrm{KL}}(Q(\boldsymbol{h}_i) \parallel p(\boldsymbol{h}_i \mid \boldsymbol{v}_i)) \tag{51}$$

$$= \int Q(\boldsymbol{h}_i)\,\log p(\boldsymbol{v}_i)\,d\boldsymbol{h}_i - \int Q(\boldsymbol{h}_i)\,\log\frac{Q(\boldsymbol{h}_i)}{p(\boldsymbol{h}_i \mid \boldsymbol{v}_i)}\,d\boldsymbol{h}_i$$

$$= -\int Q(\boldsymbol{h}_i)\,\log\frac{Q(\boldsymbol{h}_i)}{p(\boldsymbol{h}_i, \boldsymbol{v}_i)}\,d\boldsymbol{h}_i$$

$$= -\int Q(\boldsymbol{h}_i)\,\log\frac{Q(\boldsymbol{h}_i)}{p(\boldsymbol{h}_i)}\,d\boldsymbol{h}_i + \int Q(\boldsymbol{h}_i)\,\log p(\boldsymbol{v}_i \mid \boldsymbol{h}_i)\,d\boldsymbol{h}_i$$

$$= \int Q(\boldsymbol{h}_i)\,\log p(\boldsymbol{v}_i \mid \boldsymbol{h}_i)\,d\boldsymbol{h}_i - D_{\mathrm{KL}}(Q(\boldsymbol{h}_i) \parallel p(\boldsymbol{h}_i)) .$$

$D_{\mathrm{KL}}$ denotes the Kullback-Leibler (KL) divergence Kullback and Leibler [1951] which is larger than zero.

$\mathcal{F}$ is the EM objective which has to be maximized in order to maximize the likelihood. The **E-step** maximizes $\mathcal{F}$ with respect to the variational distribution $Q$, therefore the E-step minimizes $D_{\mathrm{KL}}(Q(\boldsymbol{h}_i) \parallel p(\boldsymbol{h}_i \mid \boldsymbol{v}_i))$. After the standard unconstrained E-step, the variational distribution is equal to the posterior, i.e. $Q(\boldsymbol{h}_i) = p(\boldsymbol{h}_i \mid \boldsymbol{v}_i)$. Therefore the KL divergence

$$D_{\mathrm{KL}}(Q(\boldsymbol{h}_i) \parallel p(\boldsymbol{h}_i \mid \boldsymbol{v}_i)) = 0 \tag{52}$$

is zero, thus $\mathcal{F}$ is equal to the log-likelihood $\log p(\boldsymbol{v}_i)$ ($\mathcal{F} = \log p(\boldsymbol{v}_i)$). The **M-step** maximizes $\mathcal{F}$ with respect to the parameters $(\boldsymbol{W}, \boldsymbol{\Psi})$, therefore the M-step maximizes $\int Q(\boldsymbol{h}_i)\log p(\boldsymbol{v}_i \mid \boldsymbol{h}_i)d\boldsymbol{h}_i$.

We next consider again all $n$ samples $\{\boldsymbol{v}\} = \{\boldsymbol{v}_1, \ldots, \boldsymbol{v}_n\}$. The *expected reconstruction error $\mathcal{E}$* for these $n$ data samples is

$$\mathcal{E} = -\frac{1}{n}\sum_{i=1}^{n}\int_{\mathbb{R}^l} Q(\boldsymbol{h}_i)\,\log\left(p(\boldsymbol{v}_i \mid \boldsymbol{h}_i)\right)\,d\boldsymbol{h}_i = \frac{1}{n}\sum_{i=1}^{n}\mathrm{E}_Q\left(\log\left(p(\boldsymbol{v}_i \mid \boldsymbol{h}_i)\right)\right) \tag{53}$$

and objective to maximize becomes

$$\mathcal{F} = -\mathcal{E} - \frac{1}{n}\sum_{i=1}^{n}D_{\mathrm{KL}}(Q(\boldsymbol{h}_i) \parallel p(\boldsymbol{h}_i)) . \tag{54}$$

The M-step requires to minimize $\mathcal{E}$:

$$
\mathcal{E} = \frac{m}{2} \log(2\pi) + \frac{1}{2} \log |\boldsymbol{\Psi}| + \tag{55}
$$

$$
\frac{1}{2\,n} \sum_{i=1}^{n} \mathrm{E}_Q \left( (\boldsymbol{v}_i - \boldsymbol{W}\boldsymbol{h}_i)^T \boldsymbol{\Psi}^{-1} (\boldsymbol{v}_i - \boldsymbol{W}\boldsymbol{h}_i) \right)
$$

$$
= \frac{m}{2} \log(2\pi) + \frac{1}{2} \log |\boldsymbol{\Psi}| + \tag{56}
$$

$$
\frac{1}{2\,n} \sum_{i=1}^{n} \mathrm{E}_Q \left( \boldsymbol{v}_i^T \boldsymbol{\Psi}^{-1} \boldsymbol{v}_i - 2\,\boldsymbol{v}_i^T \boldsymbol{\Psi}^{-1} \boldsymbol{W}\boldsymbol{h}_i + \boldsymbol{h}_i^T \boldsymbol{W}^T \boldsymbol{\Psi}^{-1} \boldsymbol{W}\boldsymbol{h}_i \right)
$$

$$
= \frac{m}{2} \log(2\pi) + \frac{1}{2} \log |\boldsymbol{\Psi}| + \frac{1}{2\,n} \sum_{i=1}^{n} \boldsymbol{v}_i^T \boldsymbol{\Psi}^{-1} \boldsymbol{v}_i \tag{57}
$$

$$
- \mathrm{Tr}\left( \boldsymbol{\Psi}^{-1} \boldsymbol{W} \sum_{i=1}^{n} \mathrm{E}_Q\left( \boldsymbol{h}_i \right) \boldsymbol{v}_i^T \right) + \frac{1}{2} \mathrm{Tr}\left( \boldsymbol{W}^T \boldsymbol{\Psi}^{-1} \boldsymbol{W} \sum_{i=1}^{n} \mathrm{E}_Q\left( \boldsymbol{h}_i \boldsymbol{h}_i^T \right) \right)
$$

$$
= \frac{m}{2} \log(2\pi) + \frac{1}{2} \log |\boldsymbol{\Psi}| + \frac{1}{2} \mathrm{Tr}\left( \boldsymbol{\Psi}^{-1} \frac{1}{n} \sum_{i=1}^{n} \boldsymbol{v}_i \boldsymbol{v}_i^T \right) \tag{58}
$$

$$
- \mathrm{Tr}\left( \boldsymbol{\Psi}^{-1} \boldsymbol{W} \frac{1}{n} \sum_{i=1}^{n} \boldsymbol{\mu}_i \boldsymbol{v}_i^T \right) + \frac{1}{2} \mathrm{Tr}\left( \boldsymbol{W}^T \boldsymbol{\Psi}^{-1} \boldsymbol{W} \frac{1}{n} \sum_{i=1}^{n} \left( \boldsymbol{\Sigma} + \boldsymbol{\mu}_i \boldsymbol{\mu}_i^T \right) \right)
$$

$$
= \frac{1}{2} \left( m \log(2\pi) + \log |\boldsymbol{\Psi}| + \mathrm{Tr}\left( \boldsymbol{\Psi}^{-1} \boldsymbol{C} \right) \right. \tag{59}
$$

$$
\left. - 2 \mathrm{Tr}\left( \boldsymbol{\Psi}^{-1} \boldsymbol{W} \boldsymbol{U}^T \right) + \mathrm{Tr}\left( \boldsymbol{W}^T \boldsymbol{\Psi}^{-1} \boldsymbol{W} \boldsymbol{S} \right) \right) ,
$$

where Tr gives the trace of a matrix.

The derivatives with respect to the parameters are set to zero for the optimal parameters:

$$
\nabla_{\boldsymbol{W}} \mathcal{E} = -\frac{1}{2\,n} \sum_{i=1}^{n} \boldsymbol{\Psi}^{-1} \boldsymbol{W} \, \mathrm{E}_Q\left( \boldsymbol{h}_i \, \boldsymbol{h}_i^T \right) + \frac{1}{2\,n} \sum_{i=1}^{n} \boldsymbol{\Psi}^{-1} \, \boldsymbol{v}_i \, \mathrm{E}_Q^T\left( \boldsymbol{h}_i \right) = \boldsymbol{0} \tag{60}
$$

and

$$
\nabla_{\boldsymbol{\Psi}} \mathcal{E} = -\frac{1}{2} \boldsymbol{\Psi}^{-1} + \tag{61}
$$

$$
\frac{1}{2\,n} \sum_{i=1}^{n} \mathrm{E}_Q \left( \boldsymbol{\Psi}^{-1} \left( \boldsymbol{v}_i - \boldsymbol{W}\boldsymbol{h}_i \right) \left( \boldsymbol{v}_i - \boldsymbol{W}\boldsymbol{h}_i \right)^T \boldsymbol{\Psi}^{-1} \right) = \boldsymbol{0} .
$$

Solving above equations gives:

$$
\boldsymbol{W}^{\mathrm{new}} = \left( \frac{1}{n} \sum_{i=1}^{n} \boldsymbol{v}_i \, \mathrm{E}_{\boldsymbol{h}_i | \boldsymbol{v}_i}^T \left( \boldsymbol{h}_i \right) \right) \left( \frac{1}{n} \sum_{i=1}^{n} \mathrm{E}_Q\left( \boldsymbol{h}_i \, \boldsymbol{h}_i^T \right) \right)^{-1} \tag{62}
$$

and

$$
\boldsymbol{\Psi}^{\mathrm{new}} = \frac{1}{n} \sum_{i=1}^{n} \mathrm{E}_Q \left( \left( \boldsymbol{v}_i - \boldsymbol{W}^{\mathrm{new}} \boldsymbol{h}_i \right) \left( \boldsymbol{v}_i - \boldsymbol{W}^{\mathrm{new}} \boldsymbol{h}_i \right)^T \right) = \tag{63}
$$

$$
\frac{1}{n} \sum_{i=1}^{n} \boldsymbol{v}_i \, \boldsymbol{v}_i^T - \frac{1}{n} \sum_{i=1}^{n} \boldsymbol{v}_i \mathrm{E}_Q^T\left( \boldsymbol{h}_i \right) \left( \boldsymbol{W}^{\mathrm{new}} \right)^T -
$$

$$
\frac{1}{n} \sum_{i=1}^{n} \boldsymbol{W}^{\mathrm{new}} \mathrm{E}_Q\left( \boldsymbol{h}_i \right) \boldsymbol{v}_i^T + \boldsymbol{W}^{\mathrm{new}} \frac{1}{n} \sum_{i=1}^{n} \mathrm{E}_Q\left( \boldsymbol{h}_i \, \boldsymbol{h}_i^T \right) \left( \boldsymbol{W}^{\mathrm{new}} \right)^T .
$$

We obtain the following EM updates:

<div>
**E-step:** (64)

$$\boldsymbol{\mu}_i = \left(\boldsymbol{I} + \boldsymbol{W}^T \boldsymbol{\Psi}^{-1} \boldsymbol{W}\right)^{-1} \boldsymbol{W}^T \boldsymbol{\Psi}^{-1} \boldsymbol{v}_i ,$$

$$\boldsymbol{\Sigma} = \left(\boldsymbol{I} + \boldsymbol{W}^T \boldsymbol{\Psi}^{-1} \boldsymbol{W}\right)^{-1} ,$$

$$\mathrm{E}_Q\left(\boldsymbol{h}_i\right) = \boldsymbol{\mu}_i$$

$$\mathrm{E}_Q\left(\boldsymbol{h}_i\,\boldsymbol{h}_i^T\right) = \boldsymbol{\mu}_i\,\boldsymbol{\mu}_i^T + \boldsymbol{\Sigma}$$
</div>

**M-step:** (65)

$$\boldsymbol{W}^{\mathrm{new}} = \left(\frac{1}{n}\sum_{i=1}^{n}\boldsymbol{v}_i\,\mathrm{E}_{\boldsymbol{h}_i|\boldsymbol{v}_i}^T\left(\boldsymbol{h}_i\right)\right)\left(\frac{1}{n}\sum_{i=1}^{n}\mathrm{E}_Q\left(\boldsymbol{h}_i\,\boldsymbol{h}_i^T\right)\right)^{-1}$$

$$\boldsymbol{\Psi}^{\mathrm{new}} = \frac{1}{n}\sum_{i=1}^{n}\boldsymbol{v}_i\,\boldsymbol{v}_i^T - \frac{1}{n}\sum_{i=1}^{n}\boldsymbol{v}_i\mathrm{E}_Q^T\left(\boldsymbol{h}_i\right)(\boldsymbol{W}^{\mathrm{new}})^T - \quad (66)$$

$$\frac{1}{n}\sum_{i=1}^{n}\boldsymbol{W}^{\mathrm{new}}\mathrm{E}_Q\left(\boldsymbol{h}_i\right)\boldsymbol{v}_i^T + \boldsymbol{W}^{\mathrm{new}}\frac{1}{n}\sum_{i=1}^{n}\mathrm{E}_Q\left(\boldsymbol{h}_i\,\boldsymbol{h}_i^T\right)(\boldsymbol{W}^{\mathrm{new}})^T .$$

The EM algorithms can be reformulated as:

**E-step:** (67)

$$\boldsymbol{\mu}_i = \left(\boldsymbol{I} + \boldsymbol{W}^T \boldsymbol{\Psi}^{-1} \boldsymbol{W}\right)^{-1} \boldsymbol{W}^T \boldsymbol{\Psi}^{-1} \boldsymbol{v}_i ,$$

$$\boldsymbol{\Sigma} = \left(\boldsymbol{I} + \boldsymbol{W}^T \boldsymbol{\Psi}^{-1} \boldsymbol{W}\right)^{-1} ,$$

$$\mathrm{E}_Q\left(\boldsymbol{h}_i\right) = \boldsymbol{\mu}_i$$

$$\mathrm{E}_Q\left(\boldsymbol{h}_i\,\boldsymbol{h}_i^T\right) = \boldsymbol{\mu}_i\,\boldsymbol{\mu}_i^T + \boldsymbol{\Sigma}$$

**M-step:** (68)

$$\boldsymbol{C} = \frac{1}{n}\sum_{i=1}^{n}\boldsymbol{v}_i\,\boldsymbol{v}_i^T$$

$$\boldsymbol{U} = \frac{1}{n}\sum_{i=1}^{n}\boldsymbol{v}_i\,\mathrm{E}_Q^T\left(\boldsymbol{h}_i\right) \quad (69)$$

$$\boldsymbol{S} = \frac{1}{n}\sum_{i=1}^{n}\mathrm{E}_Q\left(\boldsymbol{h}_i\,\boldsymbol{h}_i^T\right) \quad (70)$$

$$\boldsymbol{W}^{\mathrm{new}} = \boldsymbol{U}\,\boldsymbol{S}^{-1} \quad (71)$$

$$\boldsymbol{\Psi}^{\mathrm{new}} = \boldsymbol{C} - \boldsymbol{U}\boldsymbol{W}^T - \boldsymbol{W}\boldsymbol{U}^T + \boldsymbol{W}\boldsymbol{S}\boldsymbol{W}^T . \quad (72)$$

# 6  The RFN Objective

*Our goal is to find a sparse, non-negative representation of the input which extracts structure from the input.* A sparse, non-negative representation is desired to code only events or objects that have caused the input. We assume that only few events or objects caused the input, therefore, we aim at sparseness. Furthermore, we do not want to code the degree of absence of events or objects. As the vast majority of events and objects is supposed to be absent, to code for their degree of absence would introduce a high level of random fluctuations.

We aim at extracting structures from the input, therefore generative models are use as they explicitly model input structures. For example factor analysis models the covariance structure of the data. However a generative model cannot enforce sparse, non-negative representation of the input.

The input representation of a generative model is the posterior's mean, median, or mode. Generative models with rectified priors (zero probability for negative values) lead to rectified posteriors. However these posteriors do not have sparse means (they must be positive), that is, they do not yield sparse codes Frey and Hinton [1999]. For example, rectified factor analysis, which rectifies Gaussian priors and selects models using a variational Bayesian learning procedure, does not yield posteriors with sparse means Harva and Kaban [2005, 2007]. A generative model with hidden units $\boldsymbol{h}$ and data $\boldsymbol{v}$ is defined by its prior $p(\boldsymbol{h})$ and its likelihood $p(\boldsymbol{v} \mid \boldsymbol{h})$. The posterior $p(\boldsymbol{h} \mid \boldsymbol{v})$ supplies the input representation of a model by the posterior's mean, median, or mode. However, the posterior depends on the data $\boldsymbol{v}$, therefore sparseness and non-negativity of its means cannot be guaranteed independent of the data. Problem at coding the input by generative models is the data-dependency of the posterior means.

Therefore we use the *posterior regularization method* (*posterior constraint method*) Ganchev et al. [2010], Graca et al. [2009, 2007]. The posterior regularization framework separates model characteristics from data dependent characteristics like the likelihood or posterior constraints. Posterior regularization incorporates data-dependent characteristics as constraints on model posteriors given the observed data, which are difficult to encode via model parameters by Bayesian priors.

A generative model with prior $p(\boldsymbol{h})$ and likelihood $p(\boldsymbol{v} \mid \boldsymbol{h})$ has the full model distribution $p(\boldsymbol{h}, \boldsymbol{v}) = p(\boldsymbol{v} \mid \boldsymbol{h})p(\boldsymbol{h})$. It can be written as $p(\boldsymbol{h}, \boldsymbol{v}) = p(\boldsymbol{h} \mid \boldsymbol{v})p(\boldsymbol{v})$, where $p(\boldsymbol{h} \mid \boldsymbol{v})$ is the model posterior of the hidden variables and $p(\boldsymbol{v})$ is the evidence, that is, the likelihood of the data to be produced by the model. The model family and its parametrization determines which structures are extracted from the data. Typically the model parameters enter the likelihood $p(\boldsymbol{v} \mid \boldsymbol{h})$ and are adjusted to the observed data. For the posterior regularization method, a family $\mathcal{Q}$ of allowed posterior distributions is introduced. $\mathcal{Q}$ is defined by the expectations of constraint features. In our case the posterior means have to be non-negative. Distributions $Q \in \mathcal{Q}$ are called *variational distributions* (see later for using this term). The full variational distribution is $Q(\boldsymbol{h}, \boldsymbol{v}) = Q(\boldsymbol{h} \mid \boldsymbol{v})p_v(\boldsymbol{v})$ with $Q(\boldsymbol{h} \mid \boldsymbol{v}) \in \mathcal{Q}$. The distribution $p_v(\boldsymbol{v})$ is the unknown distribution of observations as determined by the world or the data generation process. This distribution is approximated by samples drawn from the world, namely the training samples. $p(\boldsymbol{h}, \boldsymbol{v})$ *contains all model assumptions like the structures used to model the data, while* $Q(\boldsymbol{h}, \boldsymbol{v})$ *contains all data dependent characteristics including data dependent constraints on the posterior.*

The goal is to achieve $Q(\boldsymbol{h}, \boldsymbol{v}) = p(\boldsymbol{h}, \boldsymbol{v})$, to obtain (1) a desired structure that is extracted from the data and (2) desired code properties. However in general it is to achieve this identity, therefore we want to minimize the distance between these distributions. We use the Kullback-Leibler (KL) divergence Kullback and Leibler [1951] $D_{\mathrm{KL}}$ to measure the distance between these distributions. Therefore our objective is $D_{\mathrm{KL}}(Q(\boldsymbol{h}, \boldsymbol{v}) \parallel p(\boldsymbol{h}, \boldsymbol{v}))$. Minimizing this KL divergence (1) extracts the desired structure from the data by increasing the likelihood, that is, $p_v(\boldsymbol{v}) \approx p(\boldsymbol{v})$, and (2) enforces desired code properties by $Q(\boldsymbol{h} \mid \boldsymbol{v}) \approx p(\boldsymbol{h} \mid \boldsymbol{v})$. Thus, the code derived from $Q(\boldsymbol{h} \mid \boldsymbol{v})$ has the desired properties and t extracts the desired input data structures.

We now approximate the KL divergence by approximating the expectation over $p_v(\boldsymbol{v})$ by the empirical mean of samples $\{\boldsymbol{v}\} = \{\boldsymbol{v}_1, \ldots, \boldsymbol{v}_n\}$ drawn from $p_v(\boldsymbol{v})$:

$$
\begin{aligned}
D_{\mathrm{KL}}(Q(\boldsymbol{h}, \boldsymbol{v}) \parallel p(\boldsymbol{h}, \boldsymbol{v})) &= \int Q(\boldsymbol{h}, \boldsymbol{v}) \, \log \frac{Q(\boldsymbol{h}, \boldsymbol{v})}{p(\boldsymbol{h}, \boldsymbol{v})} \, d\boldsymbol{h} \, d\boldsymbol{v} \qquad (73) \\
&= \int_V p_v(\boldsymbol{v}) \int_H Q(\boldsymbol{h} \mid \boldsymbol{v}) \, \log \frac{Q(\boldsymbol{h}, \boldsymbol{v})}{p(\boldsymbol{h}, \boldsymbol{v})} \, d\boldsymbol{h} \, d\boldsymbol{v} \\
&\approx \frac{1}{n} \sum_{i=1}^{n} \int_H Q(\boldsymbol{h} \mid \boldsymbol{v}_i) \, \log \frac{Q(\boldsymbol{h}, \boldsymbol{v}_i)}{p(\boldsymbol{h}, \boldsymbol{v}_i)} \, d\boldsymbol{h} \\
&= \frac{1}{n} \sum_{i=1}^{n} \int_H Q(\boldsymbol{h} \mid \boldsymbol{v}_i) \, \log \frac{Q(\boldsymbol{h} \mid \boldsymbol{v}_i)}{p(\boldsymbol{h}, \boldsymbol{v}_i)} \, d\boldsymbol{h} \; + \; \frac{1}{n} \sum_{i=1}^{n} \log p_v(\boldsymbol{v}_i) \,.
\end{aligned}
$$

The last term $\frac{1}{n} \sum_{i=1}^{n} \log p_v(\boldsymbol{v}_i)$ neither depends on $Q$ nor on the model, therefore we will neglect it. In the following, we often abbreviate $Q(\boldsymbol{h} \mid \boldsymbol{v}_i)$ by $Q(\boldsymbol{h}_i)$ or write $Q(\boldsymbol{h}_i \mid \boldsymbol{v}_i)$, since the hidden variable is based on the observation $\boldsymbol{v}_i$. Similarly we often write $p(\boldsymbol{h}_i, \boldsymbol{v}_i)$ instead of $p(\boldsymbol{h}, \boldsymbol{v}_i)$ and even more often $p(\boldsymbol{h}_i \mid \boldsymbol{v}_i)$ instead of $p(\boldsymbol{h} \mid \boldsymbol{v}_i)$.

We obtain the objective $\mathcal{F}$ (to be maximized) of the *posterior constraint method* Ganchev et al. [2010], Graca et al. [2009, 2007]:

$$\mathcal{F} = \frac{1}{n} \sum_{i=1}^{n} \log p(\boldsymbol{v}_i) - \frac{1}{n} \sum_{i=1}^{n} D_{\mathrm{KL}}(Q(\boldsymbol{h}_i) \parallel p(\boldsymbol{h}_i \mid \boldsymbol{v}_i)) \tag{74}$$

$$= \frac{1}{n} \sum_{i=1}^{n} \int Q(\boldsymbol{h}_i) \, \log p(\boldsymbol{v}_i) \, d\boldsymbol{h}_i - \frac{1}{n} \sum_{i=1}^{n} \int Q(\boldsymbol{h}_i) \, \log \frac{Q(\boldsymbol{h}_i)}{p(\boldsymbol{h}_i \mid \boldsymbol{v}_i)} \, d\boldsymbol{h}_i$$

$$= -\frac{1}{n} \sum_{i=1}^{n} \int Q(\boldsymbol{h}_i) \, \log \frac{Q(\boldsymbol{h}_i)}{p(\boldsymbol{h}_i, \boldsymbol{v}_i)} \, d\boldsymbol{h}_i$$

$$nonumber = -\frac{1}{n} \sum_{i=1}^{n} \int Q(\boldsymbol{h}_i) \, \log \frac{Q(\boldsymbol{h}_i)}{p(\boldsymbol{h}_i)} \, d\boldsymbol{h}_i + \frac{1}{n} \sum_{i=1}^{n} \int Q(\boldsymbol{h}_i) \, \log p(\boldsymbol{v}_i \mid \boldsymbol{h}_i) \, d\boldsymbol{h}_i \tag{75}$$

$$= \frac{1}{n} \sum_{i=1}^{n} \int Q(\boldsymbol{h}_i) \, \log p(\boldsymbol{v}_i \mid \boldsymbol{h}_i) \, d\boldsymbol{h}_i - \frac{1}{n} \sum_{i=1}^{n} D_{\mathrm{KL}}(Q(\boldsymbol{h}_i) \parallel p(\boldsymbol{h}_i)) \, .$$

The first line is the negative objective of the posterior constraint method while the third line is the negative Eq. (73) without the term $\frac{1}{n} \sum_{i=1}^{n} \log p_v(\boldsymbol{v}_i)$.

**$\mathcal{F}$ is the objective in our framework which has to be maximized.** Maximizing $\mathcal{F}$ (1) increases the model likelihood $\frac{1}{n} \sum_{i=1}^{n} \log p(\boldsymbol{v}_i)$, (2) finds a proper input representation by small $D_{\mathrm{KL}}(Q(\boldsymbol{h}_i) \parallel p(\boldsymbol{h}_i \mid \boldsymbol{v}_i))$. Thus, the data representation (1) extracts structures from the data as imposed by the generative model while (2) ensuring desired code properties via $Q \in \mathcal{Q}$.

In the variational framework, $Q$ is the variational distribution and $\mathcal{F}$ is called the negative *free energy* Neal and Hinton [1998]. This physical term is used since variational methods were introduced for quantum physics by Richard Feynman Feynman [1972]. The hidden variables can be considered as the fictive causes or explanations of environmental fluctuations Friston [2012].

If $p(\boldsymbol{h} \mid \boldsymbol{v}) \in \mathcal{Q}$, then $Q(\boldsymbol{h} \mid \boldsymbol{v}) = p(\boldsymbol{h} \mid \boldsymbol{v})$ and we obtain the classical EM algorithm. The EM algorithm maximizes the lower bound $\mathcal{F}$ on the $\log$-likelihood as seen at the first line of Eq. (74) and ensures in its E-step $Q(\boldsymbol{h} \mid \boldsymbol{v}) = p(\boldsymbol{h} \mid \boldsymbol{v})$.

## 7 Generalized Alternating Minimization

Instead of the EM algorithm we use the *Generalized Alternating Minimization (GAM)* algorithm Gunawardana and Byrne [2005] to allow for gradient descent both in the M-step and the E-step. The representation of an input by a generative model is the vector of the mean values of the posterior, that is, the most likely hidden variables that produced the observed data. We have to modify the E-step to enforce variational distributions which lead to sparse codes via zero values of the components of its mean vector. Sparse codes, that is, many components of the mean vector are zero, are obtained by enforcing non-negative means. This rectification is analog to rectified linear units for neural networks, which have enabled sparse codes for neural networks. Therefore the variational distributions are restricted to stem from a family with non-negative constraints on the means. To impose constraints on the posterior is known as the *posterior constraint method* Ganchev et al. [2010], Graca et al. [2009, 2007]. The posterior constraint method maximizes the objective both in the E-step and the M-step. The posterior constraint method is computationally infeasible for our approach, since we assume a large number of hidden units. For models with many hidden units, the maximization in the E-step would take too much time. The posterior constraint method does not support fast implementations on GPUs and stochastic gradients, which we want to allow in order to use mini-batches and dropout regularization.

Therefore we perform only one gradient descent step both in the E-step and in the M-step. Unfortunately, the convergence proofs of the EM algorithm are no longer valid. However we show that our algorithm is a generalized alternating minimization (GAM) method. Gunawardana and Byrne showed that the GAM converges Gunawardana and Byrne [2005] (see also Wu [1983]).

The following GAM convergence Theorem 4 is Proposition 5 in Gunawardana and Byrne [2005] and proves the convergence of the GAM algorithm to a solution that minimizes $-\mathcal{F}$.

**Theorem 4** (GAM Convergence Theorem). *Let the point-to-set map* FB *the composition* B ∘ F *of point-to-set maps* F : $\mathcal{D} \times \boldsymbol{\Theta} \to \mathcal{D} \times \boldsymbol{\Theta}$ *and* B : $\mathcal{D} \times \boldsymbol{\Theta} \to \mathcal{D} \times \boldsymbol{\Theta}$. *Suppose that the point-to-set maps* F *and* B *are defined so that*

    *(1)* F *and* B *are closed on* $\mathcal{D}' \times \boldsymbol{\Theta}$

    *(2)* $F(\mathcal{D}' \times \boldsymbol{\Theta}) \subseteq \mathcal{D} \times \boldsymbol{\Theta}$ *and* $B(\mathcal{D}' \times \boldsymbol{\Theta}) \subseteq \mathcal{D} \times \boldsymbol{\Theta}$

*Suppose also that* F *is such that all* $(Q'_X, \boldsymbol{\theta}') \in F(Q_X, \boldsymbol{\theta})$ *have* $\boldsymbol{\theta}' = \boldsymbol{\theta}$ *and satisfy*

$$(GAM.F): \qquad D_{\mathrm{KL}}(Q'_X \parallel p_{X;\boldsymbol{\theta}}) \ \leq \ D_{\mathrm{KL}}(Q_X \parallel p_{X;\boldsymbol{\theta}})$$

*with equality only if*

$$(EQ.F): \qquad Q_X \ = \ \arg \min_{Q''_X \in \mathcal{D}} D_{\mathrm{KL}}(Q''_X \parallel p_{X;\boldsymbol{\theta}}) \,,$$

*with* $Q_X$ *being the unique minimizer. Suppose also that the point-to-set map* B *is such that all* $(Q'_X, \boldsymbol{\theta}') \in B(Q_X, \boldsymbol{\theta})$ *have* $Q'_X = Q_X$ *and satisfy*

$$(GAM.B): \qquad D_{\mathrm{KL}}(Q_X \parallel p_{X;\boldsymbol{\theta}'}) \ \leq \ D_{\mathrm{KL}}(Q_X \parallel p_{X;\boldsymbol{\theta}})$$

*with equality only if*

$$(EQ.B): \qquad \boldsymbol{\theta} \ \in \ \arg \min_{\boldsymbol{\xi} \in \boldsymbol{\Theta}} D_{\mathrm{KL}}(Q_X \parallel p_{X;\boldsymbol{\xi}}) \,.$$

*Then,*

    *(1) the point-to-set map* FB *is closed on* $\mathcal{D}' \times \boldsymbol{\Theta}$

    *(2)* $FB(\mathcal{D}' \times \boldsymbol{\Theta}) \subseteq \mathcal{D} \times \boldsymbol{\Theta}$

*and* FB *satisfies the GAM and EQ conditions of the GAM convergence theorem, that is, Theorem 3 in Gunawardana and Byrne [2005].*

*Proof.* See Proposition 5 in Gunawardana and Byrne [2005]. □

The point-to-set mappings allow extended E-step and M-steps without unique iterates. Therefore, Theorem 4 holds for different implementations, different hardware, different precisions of the algorithm under consideration.

For a GAM method to converge, we have to ensure that the objective increases in both the E-step and the M-step. $Q$ is from a constrained family of variational distributions, while the posterior and the full distribution (observation and hidden units) are both derived from a model family. The model family is a parametrized family. For our models (i) the support of the density models does not depend on the parameter and (ii) the density models are continuous in their parameters. GAM convergence requires both (i) and (ii). Furthermore, both the E-step and the M-step must have unique maximizers and they increase the objective if they are not at a maximum point.

The learning rules, that is, the E-step and the M-step are closed maps as they are continuous functions. The objective for the E-step is strict convex in all its parameters for the variational distributions, simultaneously Dredze et al. [2008, 2012]. It is quadratic for the mean vectors on which constraints are imposed. The objective for the M-step is convex in both parameters $\boldsymbol{W}$ and $\boldsymbol{\Psi}^{-1}$ (we sometimes estimate $\boldsymbol{\Psi}$ instead of $\boldsymbol{\Psi}^{-1}$). The objective is quadratic in the loading matrix $\boldsymbol{W}$. For rectifying only, we guarantee unique global maximizers by convex and compact sets for both the family of desired distributions and the set of possible parameters. For this convex optimization problem with one *global* maximum. For rectifying and normalizing, the family of desired distributions is not convex due to equality constraints introduced by the normalization. However we can guarantee *local* unique maximizers.

Summary of the requirements for GAM convergence Theorem 4:

    1. the learning rules, that is, the E-step and the M-step, are closed maps,

    2. the parameter set is compact,

3. the family of variational distributions is compact (often described by the feasible set of parameters of the variational distributions),

4. the support of the density models does not depend on the parameter,

5. the density models are continuous in the parameters,

6. the E-step has a unique maximizer,

7. the E-step increases the objective if not at the maximizer,

8. the M-step has a unique maximizer (not required by Theorem 4),

9. the M-step increases the objective if not at the maximizer.

The resulting model from the GAM procedure is at a local maximum of the objective given the model family and the family of variational distributions. *The solution minimizes the KL-distance between the family of full variational distributions and full model family.* "Full" means that both the observed and the hidden variables are taken into account, where for the variational distributions the probability of the observations is set to 1. The *desired family* is defined as the set of all probability distributions that assign probability one to the observation. In our case the family of variational distributions is not the desired family since some distributions are excluded by the constraints. Therefore the solution of the GAM optimization does not guarantee stationary points in likelihood Gunawardana and Byrne [2005]. This means that we do not maximize the likelihood but minimize the KL-distance between variational distributions and model.

## 8 Gradient-based M-step

### 8.1 Gradient Ascent

The gradients in the M-step are:

$$\nabla_{\boldsymbol{W}} \mathcal{E} \;=\; \frac{1}{2\,n} \sum_{i=1}^{n} \boldsymbol{\Psi}^{-1}\, \boldsymbol{v}_i\, \mathrm{E}_Q^T\left(\boldsymbol{h}_i\right) \;-\; \frac{1}{2\,n} \sum_{i=1}^{n} \boldsymbol{\Psi}^{-1}\, \boldsymbol{W}\, \mathrm{E}_Q\left(\boldsymbol{h}_i\, \boldsymbol{h}_i^T\right)$$

and

$$\nabla_{\boldsymbol{\Psi}} \mathcal{E} \;=\; -\frac{1}{2} \boldsymbol{\Psi}^{-1} \;+\; \frac{1}{2\,n} \sum_{i=1}^{n} \mathrm{E}_Q\left(\boldsymbol{\Psi}^{-1}\left(\boldsymbol{v}_i - \boldsymbol{W}\boldsymbol{h}_i\right)\left(\boldsymbol{v}_i - \boldsymbol{W}\boldsymbol{h}_i\right)^T \boldsymbol{\Psi}^{-1}\right). \tag{76}$$

Alternatively, we can estimate $\boldsymbol{\Psi}^{-1}$ which leads to the derivatives:

$$\nabla_{\boldsymbol{\Psi}^{-1}} \mathcal{E} \;=\; \frac{1}{2} \boldsymbol{\Psi} \;-\; \frac{1}{2\,n} \sum_{i=1}^{n} \mathrm{E}_Q\left(\left(\boldsymbol{v}_i - \boldsymbol{W}\boldsymbol{h}_i\right)\left(\boldsymbol{v}_i - \boldsymbol{W}\boldsymbol{h}_i\right)^T\right). \tag{77}$$

Scaling the gradients leads to:

$$2\,\nabla_{\boldsymbol{W}} \mathcal{E} \;=\; \boldsymbol{\Psi}^{-1} \frac{1}{n} \sum_{i=1}^{n} \boldsymbol{v}_i\, \mathrm{E}_Q^T\left(\boldsymbol{h}_i\right) \;-\; \boldsymbol{\Psi}^{-1}\, \boldsymbol{W} \frac{1}{n} \sum_{i=1}^{n} \mathrm{E}_Q\left(\boldsymbol{h}_i\, \boldsymbol{h}_i^T\right) \tag{78}$$

and

$$2\,\nabla_{\boldsymbol{\Psi}} \mathcal{E} \;= \tag{79}$$

$$-\,\boldsymbol{\Psi}^{-1} + \boldsymbol{\Psi}^{-1}\left(\frac{1}{n} \sum_{i=1}^{n} \boldsymbol{v}_i\, \boldsymbol{v}_i^T - \frac{1}{n} \sum_{i=1}^{n} \boldsymbol{v}_i\, \mathrm{E}_Q^T\left(\boldsymbol{h}_i\right) \boldsymbol{W}^T\right.$$

$$\left. -\, \frac{1}{n} \sum_{i=1}^{n} \boldsymbol{W}\, \mathrm{E}_Q\left(\boldsymbol{h}_i\right) \boldsymbol{v}_i^T \;+\; \boldsymbol{W} \frac{1}{n} \sum_{i=1}^{n} \mathrm{E}_Q\left(\boldsymbol{h}_i\, \boldsymbol{h}_i^T\right) \boldsymbol{W}^T\right) \boldsymbol{\Psi}^{-1}.$$

or

$$2\,\nabla_{\boldsymbol{\Psi}^{-1}}\mathcal{E}\; = \tag{80}$$

$$\boldsymbol{\Psi}\; -\; \left(\frac{1}{n}\sum_{i=1}^{n}\boldsymbol{v}_i\,\boldsymbol{v}_i^T\; -\; \frac{1}{n}\sum_{i=1}^{n}\boldsymbol{v}_i\,\mathrm{E}_Q^T\left(\boldsymbol{h}_i\right)\,\boldsymbol{W}^T\right.$$

$$\left.-\; \frac{1}{n}\sum_{i=1}^{n}\boldsymbol{W}\,\mathrm{E}_Q\left(\boldsymbol{h}_i\right)\,\boldsymbol{v}_i^T\; +\; \boldsymbol{W}\,\frac{1}{n}\sum_{i=1}^{n}\mathrm{E}_Q\left(\boldsymbol{h}_i\,\boldsymbol{h}_i^T\right)\boldsymbol{W}^T\right)\,.$$

Only the sums

$$\boldsymbol{U}\; =\; \frac{1}{n}\sum_{i=1}^{n}\boldsymbol{v}_i\mathrm{E}_Q^T\left(\boldsymbol{h}_i\right) \tag{81}$$

and

$$\boldsymbol{S}\; =\; \frac{1}{n}\sum_{i=1}^{n}\mathrm{E}_Q\left(\boldsymbol{h}_i\,\boldsymbol{h}_i^T\right) \tag{82}$$

must be computed for both gradients.

$$\boldsymbol{C}\; =\; \frac{1}{n}\sum_{i=1}^{n}\boldsymbol{v}_i\,\boldsymbol{v}_i^T \tag{83}$$

is the estimated covariance matrix (matrix of second moments for zero mean).

**The generalized EM algorithm update rules are:**

$$\boldsymbol{C}\; =\; \frac{1}{n}\sum_{i=1}^{n}\boldsymbol{v}_i\,\boldsymbol{v}_i^T \tag{84}$$

**E-step:**

$$\boldsymbol{\mu}_i\; =\; \boldsymbol{W}^T\left(\boldsymbol{W}\,\boldsymbol{W}^T\; +\; \boldsymbol{\Psi}\right)^{-1}\boldsymbol{v}_i\; =\; \left(\boldsymbol{I}\; +\; \boldsymbol{W}^T\boldsymbol{\Psi}^{-1}\boldsymbol{W}\right)^{-1}\boldsymbol{W}^T\boldsymbol{\Psi}^{-1}\,\boldsymbol{v}_i\,,$$

$$\boldsymbol{\Sigma}\; =\; \boldsymbol{I}\; -\; \boldsymbol{W}^T\left(\boldsymbol{W}\,\boldsymbol{W}^T\; +\; \boldsymbol{\Psi}\right)^{-1}\boldsymbol{W}\; =\; \left(\boldsymbol{I}\; +\; \boldsymbol{W}^T\boldsymbol{\Psi}^{-1}\boldsymbol{W}\right)^{-1}\,,$$

$$\mathrm{E}_Q\left(\boldsymbol{h}_i\right)\; =\; \boldsymbol{\mu}_i$$

$$\mathrm{E}_Q\left(\boldsymbol{h}_i\,\boldsymbol{h}_i^T\right)\; =\; \boldsymbol{\mu}_i\,\boldsymbol{\mu}_i^T\; +\; \boldsymbol{\Sigma}$$

$$\boldsymbol{U}\; =\; \frac{1}{n}\sum_{i=1}^{n}\boldsymbol{v}_i\mathrm{E}_Q^T\left(\boldsymbol{h}_i\right)$$

$$\boldsymbol{S}\; =\; \frac{1}{n}\sum_{i=1}^{n}\mathrm{E}_Q\left(\boldsymbol{h}_i\,\boldsymbol{h}_i^T\right)$$

**M-step:** $\tag{85}$

$$\Delta\boldsymbol{W}\; =\; \boldsymbol{\Psi}^{-1}\,\boldsymbol{U}\; -\; \boldsymbol{\Psi}^{-1}\,\boldsymbol{W}\,\boldsymbol{S}$$

$$\Delta\boldsymbol{\Psi}\; =\; -\,\boldsymbol{\Psi}^{-1}\; +\; \boldsymbol{\Psi}^{-1}\left(\boldsymbol{C}\; -\; \boldsymbol{U}\,\boldsymbol{W}^T\; -\; \boldsymbol{W}\,\boldsymbol{U}\; +\; \boldsymbol{W}\,\boldsymbol{S}\,\boldsymbol{W}^T\right)\boldsymbol{\Psi}^{-1}\,.$$

## 8.2 Newton Update

Instead of gradient ascent, we now consider a Newton update step. The Newton update for finding the roots of $\frac{\partial f}{\partial \boldsymbol{v}}$ is

$$\boldsymbol{v}_{n+1}\; =\; \boldsymbol{v}_n\; -\; \eta\,\boldsymbol{H}^{-1}\,\nabla_{\boldsymbol{v}}f(\boldsymbol{v}_n)\,, \tag{86}$$

where $\eta$ is a small step size and $\boldsymbol{H}$ is the Hessian of $f$ with respect to $\boldsymbol{v}$ evaluated at $\boldsymbol{v}_n$. We denote the update direction by

$$\Delta\boldsymbol{v}\; =\; -\,\boldsymbol{H}^{-1}\,\nabla_{\boldsymbol{v}}f(\boldsymbol{v}_n)\,. \tag{87}$$

### 8.2.1 Newton Update of the Loading Matrix

**Theorem 5** (Newton Update for Loading Matrix)**.** *The M-step objective $\mathcal{E}$ is quadratic in $\boldsymbol{W}$, thus convex in $\boldsymbol{W}$. The Newton update direction for $\boldsymbol{W}$ in the M-step is*

$$\Delta \boldsymbol{W} \ = \ \boldsymbol{U} \, \boldsymbol{S}^{-1} \ - \ \boldsymbol{W} \ . \tag{88}$$

*Proof.* The M-step objective is the *expected reconstruction error $\mathcal{E}$*, which is according to Eq. (55)

$$\mathcal{E} \ = \ - \frac{1}{n} \sum_{i=1}^{n} \int_{\mathbb{R}^l} Q(\boldsymbol{h}_i) \, \log\left(p(\boldsymbol{v}_i \mid \boldsymbol{h}_i)\right) \, d\boldsymbol{h}_i \ = \ \frac{1}{2}\Big( m \, \log\left(2\pi\right) \ + \ \log |\boldsymbol{\Psi}| \tag{89}$$

$$+ \ \text{Tr}\left(\boldsymbol{\Psi}^{-1}\boldsymbol{C}\right) \ - \ 2\,\text{Tr}\left(\boldsymbol{\Psi}^{-1}\boldsymbol{W}\boldsymbol{U}^T\right) \ + \ \text{Tr}\left(\boldsymbol{W}^T\boldsymbol{\Psi}^{-1}\boldsymbol{W}\boldsymbol{S}\right) \Big) \, ,$$

where Tr gives the trace of a matrix. This is a quadratic function in $\boldsymbol{W}$, as stated in the theorem.

The Hessian $\boldsymbol{H}_{\boldsymbol{W}}$ of $(2\mathcal{E})$ with respect to $\boldsymbol{W}$ as a vector is:

$$\boldsymbol{H}_{\boldsymbol{W}} \ = \ \frac{\partial \text{vec}\left(2 \, \nabla_{\boldsymbol{W}} \mathcal{E}\right)}{\partial \text{vec}(\boldsymbol{W})^T} \ = \ \frac{\partial \text{vec}\left(-\, \boldsymbol{\Psi}^{-1}\, \boldsymbol{U} \ + \ \boldsymbol{\Psi}^{-1}\, \boldsymbol{W}\, \boldsymbol{S}\right)}{\partial \text{vec}(\boldsymbol{W})^T} \tag{90}$$

$$= \ \boldsymbol{S} \ \otimes \ \boldsymbol{\Psi}^{-1} \, ,$$

where $\otimes$ is the Kronecker product of matrices. $\boldsymbol{H}_{\boldsymbol{W}}$ is positive definite, thus the problem is convex in $\boldsymbol{W}$. The inverse of $\boldsymbol{H}_{\boldsymbol{W}}$ is

$$\boldsymbol{H}_{\boldsymbol{W}}^{-1} \ = \ \boldsymbol{S}^{-1} \ \otimes \ \boldsymbol{\Psi} \, . \tag{91}$$

For the product of the inverse Hessian with the gradient we have:

$$\boldsymbol{H}_{\boldsymbol{W}}^{-1} \, \text{vec}\left(-\, \boldsymbol{\Psi}^{-1}\, \boldsymbol{U} \ + \ \boldsymbol{\Psi}^{-1}\, \boldsymbol{W}\, \boldsymbol{S}\right) \ = \ \text{vec}\left(\boldsymbol{\Psi} \left(-\, \boldsymbol{\Psi}^{-1}\, \boldsymbol{U} \ + \ \boldsymbol{\Psi}^{-1}\, \boldsymbol{W}\, \boldsymbol{S}\right) \boldsymbol{S}^{-1}\right) \tag{92}$$
$$= \ \text{vec}\left(-\, \boldsymbol{U}\, \boldsymbol{S}^{-1} \ + \ \boldsymbol{W}\right) \, .$$

If we apply a Newton update, then the update direction for $\boldsymbol{W}$ in the M-step is

$$\Delta \boldsymbol{W} \ = \ \boldsymbol{U} \, \boldsymbol{S}^{-1} \ - \ \boldsymbol{W} \ . \tag{93}$$

$\square$

This is the exact EM update if the step-size $\eta$ is 1. Since the objective is a quadratic function in $\boldsymbol{W}$, one Newton update would lead to the exact solution.

### 8.2.2 Newton Update of the Noise Covariance

We define the expected approximation error by

$$\boldsymbol{E} \ = \ \boldsymbol{C} \ - \ \boldsymbol{U}\,\boldsymbol{W}^T \ - \ \boldsymbol{W}\,\boldsymbol{U} \ + \ \boldsymbol{W}\,\boldsymbol{S}\,\boldsymbol{W}^T \tag{94}$$

$$= \ \frac{1}{n} \sum_{i=1}^{n} \text{E}_Q\left((\boldsymbol{v}_i \ - \ \boldsymbol{W}\boldsymbol{h}_i) \, (\boldsymbol{v}_i \ - \ \boldsymbol{W}\boldsymbol{h}_i)^T\right) \, .$$

**$\boldsymbol{\Psi}$ as parameter.**

**Theorem 6** (Newton Update for Noise Covariance)**.** *The Newton update direction for $\boldsymbol{\Psi}$ as parameter in the M-step is*

$$\Delta \boldsymbol{\Psi} \ = \ \boldsymbol{E} \ - \ \boldsymbol{\Psi} \ . \tag{95}$$

*An update with $\Delta \boldsymbol{\Psi}$ ($\eta = 1$) leads to the minimum of the M-step objective $\mathcal{E}$.*

*Proof.* The M-step objective is the *expected reconstruction error $\mathcal{E}$*, which is according to Eq. (55)

$$\mathcal{E} \ = \ - \frac{1}{n} \sum_{i=1}^{n} \int_{\mathbb{R}^l} Q(\boldsymbol{h}_i) \, \log\left(p(\boldsymbol{v}_i \mid \boldsymbol{h}_i)\right) \, d\boldsymbol{h}_i \ = \ \frac{1}{2}\Big( m \, \log\left(2\pi\right) \ + \ \log |\boldsymbol{\Psi}| \tag{96}$$

$$+ \ \text{Tr}\left(\boldsymbol{\Psi}^{-1}\boldsymbol{C}\right) \ - \ 2\,\text{Tr}\left(\boldsymbol{\Psi}^{-1}\boldsymbol{W}\boldsymbol{U}^T\right) \ + \ \text{Tr}\left(\boldsymbol{W}^T\boldsymbol{\Psi}^{-1}\boldsymbol{W}\boldsymbol{S}\right) \Big) \, ,$$

where Tr gives the trace of a matrix.

Since

$$2 \, \nabla_{\boldsymbol{\Psi}} \mathcal{E} \; = \; \boldsymbol{\Psi}^{-1} \; - \; \boldsymbol{\Psi}^{-1} \boldsymbol{E} \boldsymbol{\Psi}^{-1} \,, \tag{97}$$

is

$$\boldsymbol{\Psi} \; = \; \boldsymbol{E} \tag{98}$$

the minimum of $\mathcal{E}$ with respect to $\boldsymbol{\Psi}$. Therefore an update with $\Delta \boldsymbol{\Psi} = \boldsymbol{E} - \boldsymbol{\Psi}$ leads to the minimum.

The Hessian $\boldsymbol{H}_{\boldsymbol{\Psi}}$ of $(2\mathcal{E})$ with respect to $\boldsymbol{\Psi}$ as a vector is:

$$\begin{aligned} \boldsymbol{H}_{\boldsymbol{\Psi}} \; &= \; \frac{\partial \mathrm{vec} \left( 2 \, \nabla_{\boldsymbol{\Psi}} \mathcal{E} \right)}{\partial \mathrm{vec}(\boldsymbol{\Psi})^T} \; = \; \frac{\partial \mathrm{vec} \left( \boldsymbol{\Psi}^{-1} \; - \; \boldsymbol{\Psi}^{-1} \boldsymbol{E} \boldsymbol{\Psi}^{-1} \right)}{\partial \mathrm{vec}(\boldsymbol{\Psi})^T} \\ &= \; - \, \boldsymbol{\Psi}^{-1} \; \otimes \; \boldsymbol{\Psi}^{-1} \; + \; \boldsymbol{\Psi}^{-1} \; \otimes \; \left( \boldsymbol{\Psi}^{-1} \boldsymbol{E} \boldsymbol{\Psi}^{-1} \right) \; + \; \left( \boldsymbol{\Psi}^{-1} \boldsymbol{E} \boldsymbol{\Psi}^{-1} \right) \; \otimes \; \boldsymbol{\Psi}^{-1} \,. \end{aligned} \tag{99}$$

The expected approximation error $\boldsymbol{E}$ is a sample estimate for $\boldsymbol{\Psi}$, therefore we have $\boldsymbol{\Psi} \approx \boldsymbol{E}$. The Hessian may not be positive definite for some values of $\boldsymbol{E}$, like for small values of $\boldsymbol{E}$. In order to guarantee a positive definite Hessian, more precisely an approximation to it, for minmization, we set

$$\boldsymbol{E} \; = \; \boldsymbol{\Psi} \tag{100}$$

and obtain

$$\boldsymbol{H}_{\boldsymbol{\Psi}} \; = \; \boldsymbol{\Psi}^{-1} \; \otimes \; \boldsymbol{\Psi}^{-1} \,. \tag{101}$$

We derive an approximate Newton update that is very close to the Newton update.

The inverse of the approximated $\boldsymbol{H}_{\boldsymbol{\Psi}}$ is

$$\boldsymbol{H}_{\boldsymbol{\Psi}}^{-1} \; = \; \boldsymbol{\Psi} \; \otimes \; \boldsymbol{\Psi} \,. \tag{102}$$

For the product of the inverse Hessian with the gradient we have:

$$\begin{aligned} \boldsymbol{H}_{\boldsymbol{\Psi}}^{-1} \, \mathrm{vec} \left( \boldsymbol{\Psi}^{-1} \; - \; \boldsymbol{\Psi}^{-1} \boldsymbol{E} \boldsymbol{\Psi}^{-1} \right) \; &= \; \mathrm{vec} \left( \boldsymbol{\Psi} \, \left( \boldsymbol{\Psi}^{-1} \; - \; \boldsymbol{\Psi}^{-1} \boldsymbol{E} \boldsymbol{\Psi}^{-1} \right) \boldsymbol{\Psi} \right) \\ &= \; \mathrm{vec} \left( \boldsymbol{\Psi} \; - \; \boldsymbol{E} \right) \,. \end{aligned} \tag{103}$$

If we apply a Newton update, then the update direction for $\boldsymbol{\Psi}$ in the M-step is

$$\Delta \boldsymbol{\Psi} \; = \; \boldsymbol{E} \; - \; \boldsymbol{\Psi} \,. \tag{104}$$

This is the exact EM update if the step-size $\eta$ is 1. □

## $\boldsymbol{\Psi}^{-1}$ as parameter.

**Theorem 7** (Newton Update for Inverse Noise Covariance). *The M-step objective $\mathcal{E}$ is convex in $\boldsymbol{\Psi}^{-1}$. The Newton update direction for $\boldsymbol{\Psi}^{-1}$ as parameter in the M-step is*

$$\Delta \boldsymbol{\Psi}^{-1} \; = \; \boldsymbol{\Psi}^{-1} \; - \; \boldsymbol{\Psi}^{-1} \, \boldsymbol{E} \, \boldsymbol{\Psi}^{-1} \,. \tag{105}$$

*A first order approximation of this Newton direction for $\boldsymbol{\Psi}$ in the M-step is*

$$\Delta \boldsymbol{\Psi} \; = \; \boldsymbol{E} \; - \; \boldsymbol{\Psi} \,. \tag{106}$$

*An update with $\Delta \boldsymbol{\Psi}$ ($\eta = 1$) leads to the minimum of the M-step objective $\mathcal{E}$.*

*Proof.* The M-step objective is the *expected reconstruction error $\mathcal{E}$*, which is according to Eq. (55)

$$\begin{aligned} \mathcal{E} \; = \; - \, \frac{1}{n} \sum_{i=1}^{n} \int_{\mathbb{R}^l} Q(\boldsymbol{h}_i) \, \log \left( p(\boldsymbol{v}_i \mid \boldsymbol{h}_i) \right) \, d\boldsymbol{h}_i \; &= \; \frac{1}{2} \Big( m \, \log \left( 2\pi \right) \; + \; \log |\boldsymbol{\Psi}| \\ &+ \; \mathrm{Tr} \left( \boldsymbol{\Psi}^{-1} \boldsymbol{C} \right) \; - \; 2 \, \mathrm{Tr} \left( \boldsymbol{\Psi}^{-1} \boldsymbol{W} \boldsymbol{U}^T \right) \; + \; \mathrm{Tr} \left( \boldsymbol{W}^T \boldsymbol{\Psi}^{-1} \boldsymbol{W} \boldsymbol{S} \right) \Big), \end{aligned} \tag{107}$$

where Tr gives the trace of a matrix.

Since

$$2 \, \nabla_{\boldsymbol{\Psi}^{-1}} \mathcal{E} \; = \; - \, \boldsymbol{\Psi} \; + \; \boldsymbol{E} \tag{108}$$

is

$$\boldsymbol{\Psi} \; = \; \boldsymbol{E} \tag{109}$$

the minimum of $\mathcal{E}$ with respect to $\boldsymbol{\Psi}^{-1}$. Therefore an update with $\Delta\boldsymbol{\Psi} \, = \, \boldsymbol{E} \, - \, \boldsymbol{\Psi}$ leads to the minimum.

The Hessian $\boldsymbol{H}_{\boldsymbol{\Psi}^{-1}}$ of $(2\mathcal{E})$ with respect to $\boldsymbol{\Psi}^{-1}$ as a vector is:

$$\boldsymbol{H}_{\boldsymbol{\Psi}^{-1}} \; = \; \frac{\partial \mathrm{vec} \, (2 \, \nabla_{\boldsymbol{\Psi}^{-1}} \mathcal{E})}{\partial \mathrm{vec}(\boldsymbol{\Psi}^{-1})^T} \; = \; \frac{\partial \mathrm{vec} \, (- \, \boldsymbol{\Psi} \, + \, \boldsymbol{E})}{\partial \mathrm{vec}(\boldsymbol{\Psi}^{-1})^T} \; = \; \boldsymbol{\Psi} \; \otimes \; \boldsymbol{\Psi} \, . \tag{110}$$

Since the Hessian is positive definite, the E-step objective $\mathcal{E}$ is convex in $\boldsymbol{\Psi}^{-1}$, which is the first statement of the theorem.

The inverse of $\boldsymbol{H}_{\boldsymbol{\Psi}^{-1}}$ is

$$\boldsymbol{H}_{\boldsymbol{\Psi}^{-1}}^{-1} \; = \; \boldsymbol{\Psi}^{-1} \; \otimes \; \boldsymbol{\Psi}^{-1} \, . \tag{111}$$

For the product of the inverse Hessian with the gradient we have:

$$\begin{aligned}
\boldsymbol{H}_{\boldsymbol{\Psi}^{-1}}^{-1} \, \mathrm{vec} \, (- \, \boldsymbol{\Psi} \, + \, \boldsymbol{E}) \; &= \; \mathrm{vec} \, \left( \boldsymbol{\Psi}^{-1} \, (- \, \boldsymbol{\Psi} \, + \, \boldsymbol{E}) \, \boldsymbol{\Psi}^{-1} \right) \\
&= \, \mathrm{vec} \, \left( - \, \boldsymbol{\Psi}^{-1} \, + \, \boldsymbol{\Psi}^{-1} \, \boldsymbol{E} \, \boldsymbol{\Psi}^{-1} \right) \, .
\end{aligned} \tag{112}$$

If we apply a Newton update, then the update direction for $\boldsymbol{\Psi}^{-1}$ in the M-step is

$$\Delta\boldsymbol{\Psi}^{-1} \; = \; \boldsymbol{\Psi}^{-1} \; - \; \boldsymbol{\Psi}^{-1} \, \boldsymbol{E} \, \boldsymbol{\Psi}^{-1} \, . \tag{113}$$

We now can approximate the update for $\boldsymbol{\Psi}$ by the first terms of the Taylor expansion:

$$\boldsymbol{\Psi} \, + \, \Delta\boldsymbol{\Psi} \; = \; \left( \boldsymbol{\Psi}^{-1} \, + \, \Delta\boldsymbol{\Psi}^{-1} \right)^{-1} \; \approx \; \boldsymbol{\Psi} \, - \, \boldsymbol{\Psi} \, \Delta\boldsymbol{\Psi}^{-1} \, \boldsymbol{\Psi} \, . \tag{114}$$

We obtain for the update of $\boldsymbol{\Psi}$

$$\Delta\boldsymbol{\Psi} \; = \; - \, \boldsymbol{\Psi} \, \Delta\boldsymbol{\Psi}^{-1} \, \boldsymbol{\Psi} \; = \; \boldsymbol{E} \, - \, \boldsymbol{\Psi} \, . \tag{115}$$

This is the exact EM update if the step-size $\eta$ is 1. $\qquad\square$

The Newton update derived from $\boldsymbol{\Psi}^{-1}$ as parameter is the Newton update for $\boldsymbol{\Psi}$. Consequently, the Newton direction for both $\boldsymbol{\Psi}$ and $\boldsymbol{\Psi}^{-1}$ is in the M-step

$$\Delta\boldsymbol{\Psi} \; = \; \boldsymbol{E} \, - \, \boldsymbol{\Psi} \, . \tag{116}$$

# 9 Gradient-based E-Step

## 9.1 Motivation for Rectifying and Normalization Constraints

The representation of data vector $\boldsymbol{v}$ by the model is the variational mean vector $\boldsymbol{\mu}_q$. In order to obtain sparse codes we want to have non-negative $\boldsymbol{\mu}_q$. We enforce non-negative mean values by constraints and optimize by projected Newton methods and by gradient projection methods. Non-negative constraints correspond to rectifying in the neural network field. Therefore we aim to construct sparse codes in analogy to the rectified linear units used for neural networks.

We constrain the variational distributions to the family of normal distributions with non-negative mean components. Consequently we introduce non-negative or **rectifying constraints**:

$$\boldsymbol{\mu} \; \geq \; \boldsymbol{0} \, , \tag{117}$$

where the inequality "$\geq$" holds component-wise.

However generative models with many coding units face a problem. They tend to *explain away small and rare signals by noise*. For many coding units, model selection algorithms prefer models with coding units which do not have variation and, therefore, are removed from the model. Other coding units hardly contribute to explain the observations. The likelihood is larger if small and rare signals are explained by noise, than the likelihood if coding units are use to explain such signals. Coding units without variance are kept on their default values, where they have maximal contribution to the likelihood. If they are used for coding, they deviate from their maximal values for each sample. In accumulation these deviations decrease the likelihood more than it is increased by explaining small or rare signals. For our RFN models the problem can become severe, since we aim at models with up to several tens of thousands of coding units. To avoid the explaining away problem, we enforce the selected models to use all their coding units on an equal level. We do that by keeping the variation of each noise-free coding unit across the training set at one. Consequently, we introduce a **normalization constraint** for each coding unit $1 \leq j \leq l$:

$$\frac{1}{n} \sum_{i=1}^{n} \mu_{ij}^2 \;=\; 1 \;. \tag{118}$$

This constraint means that the noise-free part of each coding unit has variance one across samples.

We will derive methods to increase the objective in the E-step both for only rectifying constraints and for rectifying and normalization constraints. These methods ensure to reduce the objective in the E-step to guarantee convergence via the GAM theory. The resulting model from the GAM procedure is at a local maximum of the objective given the model family and the family of variational distributions. *The solution minimizes the KL-distance between the family of full variational distributions and full model family*. "Full" means that both the observed and the hidden variables are taken into account.

## 9.2 The Full E-step Objective

The E-step maximizes $\mathcal{F}$ with respect to the variational distribution $Q$, therefore the E-step minimizes the Kullback-Leibler divergence (KL-divergence) Kullback and Leibler [1951] $D_{\mathrm{KL}}(Q(\boldsymbol{h}) \,\|\, p(\boldsymbol{h} \mid \boldsymbol{v}))$. The KL-divergence between $Q$ and $p$ is

$$D_{\mathrm{KL}}(Q \,\|\, p) \;=\; \int Q(\boldsymbol{h}) \, \log \frac{Q(\boldsymbol{h})}{p(\boldsymbol{h} \mid \boldsymbol{v})} \, d\boldsymbol{h} \;. \tag{119}$$

*Rectifying constraints* introduce non-negative constraints. The minimization with respect to $Q(\boldsymbol{h}_i)$ gives the constraint minimization problem:

$$\min_{Q(\boldsymbol{h}_i)} \;\; \frac{1}{n} \sum_{i=1}^{n} D_{\mathrm{KL}}(Q(\boldsymbol{h}_i) \,\|\, p(\boldsymbol{h}_i \mid \boldsymbol{v}_i)) \tag{120}$$

$$\text{s.t.} \quad \forall_i : \; \boldsymbol{\mu}_i \;\geq\; \boldsymbol{0} \;,$$

where $\boldsymbol{\mu}_i$ is the mean vector of $Q(\boldsymbol{h}_i)$.

*Rectifying and normalizing constraints* introduce non-negative constraints and equality constraints. The minimization with respect to $Q(\boldsymbol{h}_i)$ gives the constraint minimization problem:

$$\min_{Q(\boldsymbol{h}_i)} \;\; \frac{1}{n} \sum_{i=1}^{n} D_{\mathrm{KL}}(Q(\boldsymbol{h}_i) \,\|\, p(\boldsymbol{h}_i \mid \boldsymbol{v}_i)) \tag{121}$$

$$\text{s.t.} \quad \forall_i : \; \boldsymbol{\mu}_i \;\geq\; \boldsymbol{0} \;,$$

$$\forall_j : \; \frac{1}{n} \sum_{i=1}^{n} \mu_{ij}^2 \;=\; 1 \;,$$

where $\boldsymbol{\mu}_i$ is the mean vector of $Q(\boldsymbol{h}_i)$.

First we consider the families from which the model and from which the variational distributions stem. The posterior of the model with Gaussian prior $p(\boldsymbol{h})$ is Gaussian (see Section 5):

$$p(\boldsymbol{h} \mid \boldsymbol{v}) \;\sim\; (2\pi)^{-\frac{l}{2}} \; |\boldsymbol{\Sigma}_p|^{-\frac{1}{2}} \; \exp\left( -\frac{1}{2} \, (\boldsymbol{h} - \boldsymbol{\mu}_p)^T \, \boldsymbol{\Sigma}_p^{-1} \, (\boldsymbol{h} - \boldsymbol{\mu}_p) \right) \;. \tag{122}$$

To be as close as possible to the posterior distribution, we restrict $Q$ to be from a Gaussian family:

$$Q(\boldsymbol{h}) \sim (2\pi)^{-\frac{l}{2}} |\boldsymbol{\Sigma}_q|^{-\frac{1}{2}} \exp\left(-\frac{1}{2}(\boldsymbol{h} - \boldsymbol{\mu}_q)^T \boldsymbol{\Sigma}_q^{-1} (\boldsymbol{h} - \boldsymbol{\mu}_q)\right) . \tag{123}$$

For Gaussians, the Kullback-Leibler divergence between $Q$ and $p$ is

$$D_{\mathrm{KL}}(Q \parallel p) = \tag{124}$$

$$\frac{1}{2}\left\{\mathrm{Tr}\left(\boldsymbol{\Sigma}_p^{-1} \boldsymbol{\Sigma}_q\right) + (\boldsymbol{\mu}_p - \boldsymbol{\mu}_q)^T \boldsymbol{\Sigma}_p^{-1} (\boldsymbol{\mu}_p - \boldsymbol{\mu}_q) - l - \ln\frac{|\boldsymbol{\Sigma}_q|}{|\boldsymbol{\Sigma}_p|}\right\} .$$

This Kullback-Leibler divergence is convex in the mean vector $\boldsymbol{\mu}_q$ and the covariance matrix $\boldsymbol{\Sigma}_q$ of $Q$, simultaneously Dredze et al. [2008, 2012].

We now minimize Eq. (124) with respect to $Q$. For the moment we do not care about the constraints introduced by non-negativity and by normalization. Eq. (124) has a quadratic form in $\boldsymbol{\mu}_q$, where $\boldsymbol{\Sigma}_q$ does not enter, and terms in $\boldsymbol{\Sigma}_q$, where $\boldsymbol{\mu}_q$ does not enter. Therefore we can separately minimize for $\boldsymbol{\Sigma}_q$ and for $\boldsymbol{\mu}_q$.

For the minimization with respect to $\boldsymbol{\Sigma}_q$, we require

$$\frac{\partial}{\partial \boldsymbol{\Sigma}_q}\mathrm{Tr}\left(\boldsymbol{\Sigma}_p^{-1} \boldsymbol{\Sigma}_q\right) = \boldsymbol{\Sigma}_p^{-T} \tag{125}$$

and

$$\frac{\partial}{\partial \boldsymbol{\Sigma}_q}\ln|\boldsymbol{\Sigma}_q| = \boldsymbol{\Sigma}_q^{-T} . \tag{126}$$

For optimality the derivative of the objective $D_{\mathrm{KL}}(Q \parallel p)$ with respect to $\boldsymbol{\Sigma}_q$ must be zero:

$$\frac{\partial}{\partial \boldsymbol{\Sigma}_q}D_{\mathrm{KL}}(Q \parallel p) = \frac{1}{2}\boldsymbol{\Sigma}_p^{-T} - \frac{1}{2}\boldsymbol{\Sigma}_q^{-T} = \boldsymbol{0} . \tag{127}$$

This gives

$$\boldsymbol{\Sigma} = \boldsymbol{\Sigma}_q = \boldsymbol{\Sigma}_p . \tag{128}$$

We often drop the index $q$ since for $1 \leq i \leq n$ all covariance matrices $\boldsymbol{\Sigma}_q$ are equal to $\boldsymbol{\Sigma}_p$.

The mean vector $\boldsymbol{\mu}_q$ of $Q$ is the solution of the minimization problem:

$$\min_{\boldsymbol{\mu}} \frac{1}{2}(\boldsymbol{\mu}_p - \boldsymbol{\mu})^T \boldsymbol{\Sigma}_p^{-1} (\boldsymbol{\mu}_p - \boldsymbol{\mu}) \tag{129}$$

which is equivalent to

$$\min_{\boldsymbol{\mu}} \frac{1}{2}\boldsymbol{\mu}^T \boldsymbol{\Sigma}_p^{-1}\boldsymbol{\mu} - \boldsymbol{\mu}_p^T \boldsymbol{\Sigma}_p^{-1}\boldsymbol{\mu} . \tag{130}$$

The derivative and the Hessian of this objective is:

$$\frac{\partial}{\partial \boldsymbol{\mu}}D_{\mathrm{KL}}(Q \parallel p) = \boldsymbol{\Sigma}_p^{-1}(\boldsymbol{\mu} - \boldsymbol{\mu}_p) , \tag{131}$$

$$\frac{\partial^2}{\partial^2 \boldsymbol{\mu}}D_{\mathrm{KL}}(Q \parallel p) = \boldsymbol{\Sigma}_p^{-1} . \tag{132}$$

## 9.3 E-step for Mean with Rectifying Constraints

### 9.3.1 The E-Step Minimization Problem

Rectifying is realized by non-negative constraints. The mean vector $\boldsymbol{\mu}_q$ of $Q$ is the solution of the minimization problem:

$$\min_{\boldsymbol{\mu}} \frac{1}{2}(\boldsymbol{\mu} - \boldsymbol{\mu}_p)^T \boldsymbol{\Sigma}_p^{-1} (\boldsymbol{\mu} - \boldsymbol{\mu}_p) \tag{133}$$

$$\text{s.t.} \quad \boldsymbol{\mu} \geq \boldsymbol{0} .$$

This is a convex quadratic minimization problem with non-negativity constraints (convex feasible set).

If $\boldsymbol{\lambda}$ is the Lagrange multiplier for the constraints, then the dual is

$$\min_{\boldsymbol{\lambda}} \quad \frac{1}{2}\, \boldsymbol{\lambda}^T \boldsymbol{\Sigma}_p \boldsymbol{\lambda} \; + \; \boldsymbol{\mu}_p^T \boldsymbol{\lambda} \tag{134}$$
$$\text{s.t.} \quad \boldsymbol{\lambda} \; \geq \; \mathbf{0} \,.$$

The Karush-Kuhn-Tucker conditions require for the optimal solution for each component $1 \leq j \leq l$:

$$\lambda_j\, \mu_j \; = \; 0 \,. \tag{135}$$

Further the derivative of the Lagrangian with respect to $\boldsymbol{\mu}$ gives

$$\boldsymbol{\Sigma}_p^{-1}\boldsymbol{\mu} \; - \; \boldsymbol{\Sigma}_p^{-1}\boldsymbol{\mu}_p \; - \; \boldsymbol{\lambda} \; = \; \mathbf{0} \tag{136}$$

which can be written as

$$\boldsymbol{\mu} \; - \; \boldsymbol{\mu}_p \; - \; \boldsymbol{\Sigma}_p\, \boldsymbol{\lambda} \; = \; \mathbf{0} \,. \tag{137}$$

This minimization problem cannot be solved directly. Therefore we perform a gradient projection or projected Newton step to decrease the objective.

### 9.3.2 The Projection onto the Feasible Set

To decrease the objective, we perform a gradient projection or a projected Newton step. We will base our algorithms on *Euclidean least distance projections*. If projected onto convex sets, these projections do not increase distances. The Euclidean projection onto the feasible set is denoted by P, that is, the map that takes $\boldsymbol{\mu}_p$ to its nearest point $\boldsymbol{\mu}$ (in the $L^2$-norm) in the feasible set.

For rectifying constraints, the projection P (Euclidean least distance projection) of $\boldsymbol{\mu}_p$ onto the convex feasible set is given by the solution of the convex optimization problem:

$$\min_{\boldsymbol{\mu}} \quad \frac{1}{2}\, (\boldsymbol{\mu} \, - \, \boldsymbol{\mu}_p)^T\, (\boldsymbol{\mu} \, - \, \boldsymbol{\mu}_p) \tag{138}$$
$$\text{s.t.} \quad \boldsymbol{\mu} \; \geq \; \mathbf{0} \,.$$

The following Theorem 8 shows that update Eq. (139) is the projection P defined by optimization problem Eq. (138).

**Theorem 8** (Projection: Rectifying)**.** *The solution to optimization problem Eq. (138), which defines the Euclidean least distance projection, is*

$$\mu_j \; = \; [\mathrm{P}(\boldsymbol{\mu}_p)]_j \; = \; \begin{cases} 0 & \text{for} \quad (\mu_p)_j \; \leq \; 0 \\ (\mu_p)_j & \text{for} \quad (\mu_p)_j \; > \; 0 \end{cases} \tag{139}$$

*Proof.* For the projection we have the minimization problem:

$$\min_{\boldsymbol{\mu}} \quad \frac{1}{2}\, (\boldsymbol{\mu} \, - \, \boldsymbol{\mu}_p)^T\, (\boldsymbol{\mu} \, - \, \boldsymbol{\mu}_p) \tag{140}$$
$$\text{s.t.} \quad \boldsymbol{\mu} \; \geq \; \mathbf{0} \,.$$

The Lagrangian $L$ with multiplier $\boldsymbol{\lambda} \geq \mathbf{0}$ is

$$L \; = \; \frac{1}{2}\, (\boldsymbol{\mu} \, - \, \boldsymbol{\mu}_p)^T\, (\boldsymbol{\mu} \, - \, \boldsymbol{\mu}_p) \; - \; \boldsymbol{\lambda}^T\, \boldsymbol{\mu} \,. \tag{141}$$

The derivative with respect to $\boldsymbol{\mu}$ is

$$\frac{\partial L}{\partial \boldsymbol{\mu}} \; = \; \boldsymbol{\mu} \, - \, \boldsymbol{\mu}_p \, - \, \boldsymbol{\lambda} \; = \; \mathbf{0} \,. \tag{142}$$

The Karush-Kuhn-Tucker (KKT) conditions require for the optimal solution that for each constraint $j$:

$$\lambda_j\, \mu_j \; = \; 0 \,. \tag{143}$$

If $0 < (\mu_p)_j$ then Eq. (142) requires $0 < \mu_j$ because the Lagrangian $\lambda_j$ is larger than or equal to zero: $0 \leq \lambda_j$. From the KKT conditions Eq. (143) follows that $\lambda_j = 0$ and, therefore, $0 < \mu_j = (\mu_p)_j$. If $(\mu_p)_j < 0$ then $0 < \mu_j - (\mu_p)_j$, because the constraints of the primal problem require $0 \leq \mu_j$. From Eq. (142) follows that $0 < \lambda_j$. From the KKT conditions Eq. (143) follows that $(\mu_p)_j = 0$ and $0 < \lambda_j = -(\mu_p)_j$. If $(\mu_p)_j = 0$, then Eq. (142) and the KKT conditions Eq. (143) lead to $(\mu_p)_j = \mu_j = \lambda_j = 0$.

Therefore the solution of problem Eq. (138) is

$$\mu_j = \begin{cases} (\mu_p)_j & \text{for} \quad (\mu_p)_j > 0 \;\text{and}\; \lambda_j = 0 \\ 0 & \text{for} \quad (\mu_p)_j \leq 0 \;\text{and}\; \lambda_j = -(\mu_p)_j \end{cases} \quad . \tag{144}$$

This finishes the proof. $\qquad\qquad\qquad\qquad\qquad\qquad\qquad\qquad\qquad\qquad\qquad\qquad\square$

### 9.4 E-step for Mean with Rectifying and Normalizing Constraints

#### 9.4.1 The E-Step Minimization Problem

If we also consider normalizing constraints, then we have to minimize all KL-divergences simultaneously. The normalizing constraints connect the single optimization problems for each sample $v_i$. For the E-step, we obtain the minimization problem:

$$\min_{\boldsymbol{\mu}_i} \;\; \frac{1}{n} \sum_{i=1}^{n} (\boldsymbol{\mu}_i - (\boldsymbol{\mu}_p)_i)^T \, \boldsymbol{\Sigma}_p^{-1} \, (\boldsymbol{\mu}_i - (\boldsymbol{\mu}_p)_i) \tag{145}$$

$$\text{s.t.} \;\; \forall_i : \boldsymbol{\mu}_i \geq \mathbf{0} \;\;,\;\; \forall_j : \frac{1}{n} \sum_{i=1}^{n} \mu_{ij}^2 = 1 \;.$$

The "$\geq$"-sign is meant component-wise. The $l$ equality constraints lead to non-convex feasible sets. The solution to this optimization problem are the means vectors $\boldsymbol{\mu}_i$ of $Q(\boldsymbol{h}_i)$.

**Generalized Reduced Gradient.** The equality constraints can be solved for one variable which is then inserted into the objective. The equality constraint gives for each $1 \leq j \leq l$:

$$\mu_{1j}^2 = n - \sum_{i=2}^{n} \mu_{ij}^2 \quad \text{or} \quad \mu_{1j} = \sqrt{n - \sum_{i=2}^{n} \mu_{ij}^2} \;. \tag{146}$$

These equations can be inserted into the objective and, thereby, we remove the variables $\mu_{1j}$. We have to ensure that the $\mu_{1j}$ exist by

$$\sum_{i=2}^{n} \mu_{ij}^2 \leq n \;. \tag{147}$$

These constraints define a convex set feasible set. To solve the each equality constraints for a variable and insert it into the objective is called *generalized reduced gradient* method Abadie and Carpentier [1969]. For solving the reduced problem, we can use methods for constraint optimization were we now ensure a convex feasible set. These methods solve the original problem Eq. (145). We only require an improvement of the objective with a feasible value. For the reduced problem, we perform one step of a *gradient projection method*.

**Gradient Projection Methods.** Also for the original problem Eq. (145), *gradient projection methods* can be used. The gradient projection method has been generalized by Rosen to *non-linear constraints* Rosen [1961] and was later improved by Haug and Arora [1979]. The gradient projection algorithm of Rosen works for *non-convex feasible sets*. The idea is to linearize the nonlinear constraints and solve the problem. Subsequently a restoration move brings the solution back to the constraint boundaries.

### 9.4.2 The Projection onto the Feasible Set

To decrease the objective, we perform a gradient projection, a projected Newton step, or a step of the generalized reduced method. We will base our algorithms on *Euclidean least distance projections*. If projected onto convex sets, these projections do not increase distances. The Euclidean projection onto the feasible set is denoted by P, that is, the map that simultaneously takes $\{(\boldsymbol{\mu}_p)_i\}$ to the nearest points $\{\boldsymbol{\mu}_i\}$ (in the $L^2$-norm) in the feasible set.

For rectifying and normalizing constraints the projection (Euclidean least distance projection) of $\{(\boldsymbol{\mu}_p)_i\}$ onto the **non-convex** feasible set leads to the optimization problem

$$\min_{\boldsymbol{\mu}_i} \quad \frac{1}{n} \sum_{i=1}^n (\boldsymbol{\mu}_i - (\boldsymbol{\mu}_p)_i)^T (\boldsymbol{\mu}_i - (\boldsymbol{\mu}_p)_i) \tag{148}$$

$$\text{s.t.} \quad \forall_i : \boldsymbol{\mu}_i \geq \mathbf{0} \,,$$

$$\forall_j : \frac{1}{n} \sum_{i=1}^n \mu_{ij}^2 = 1 \,.$$

By using $(\boldsymbol{\mu}_i - (\boldsymbol{\mu}_p)_i)^T (\boldsymbol{\mu}_i - (\boldsymbol{\mu}_p)_i) = \boldsymbol{\mu}_i^T \boldsymbol{\mu}_i - 2\boldsymbol{\mu}_i^T (\boldsymbol{\mu}_p)_i + (\boldsymbol{\mu}_p)_i^T (\boldsymbol{\mu}_p)_i$, we see that the objective contains the sum $\sum_{ij} \mu_{ij}^2$. The constraints enforce this sum to be constant. Therefore inserting the equality constraints into the objective, optimization problem Eq. (148) is equivalent to

$$\min_{\boldsymbol{\mu}_i} \quad - \frac{1}{n} \sum_{i=1}^n \boldsymbol{\mu}_i^T (\boldsymbol{\mu}_p)_i \tag{149}$$

$$\text{s.t.} \quad \forall_i : \boldsymbol{\mu}_i \geq \mathbf{0} \,,$$

$$\forall_j : \frac{1}{n} \sum_{i=1}^n \mu_{ij}^2 = 1 \,.$$

The following Theorem 9 shows that updates Eq. (150) and Eq. (151) form the projection defined by optimization problem Eq. (148).

**Theorem 9** (Projection: Rectifying and Normalizing). *If at least one $(\mu_p)_{ij}$ is positive for $1 \leq j \leq l$, then the solution to optimization problem Eq. (148), which defines the Euclidean least distance projection, is*

$$\hat{\mu}_{ij} = \begin{cases} 0 & \text{for} \quad (\mu_p)_{ij} \leq 0 \\ (\mu_p)_{ij} & \text{for} \quad (\mu_p)_{ij} > 0 \end{cases} \tag{150}$$

$$\mu_{ij} = [\mathrm{P}((\boldsymbol{\mu}_p)_i)]_j = \frac{\hat{\mu}_{ij}}{\sqrt{\frac{1}{n} \sum_{i=1}^n \hat{\mu}_{ij}^2}} \,.$$

*If all $(\mu_p)_{ij}$ are non-positive for $1 \leq j \leq l$, then the optimization problem Eq. (148) has the solution*

$$\mu_{ij} = \begin{cases} \sqrt{n} & \text{for} \quad j = \arg\max_{\hat{j}} \{(\mu_p)_{i\hat{j}}\} \\ 0 & \text{otherwise} \end{cases} \,. \tag{151}$$

*Proof.* In the following we show that updates Eq. (150) and Eq. (150) are the projection onto the feasible set. For the projection of $\{(\boldsymbol{\mu}_p)_i\}$ onto the feasible set, we have the minimization problem:

$$\min_{\boldsymbol{\mu}_i} \quad \frac{1}{n} \sum_{i=1}^n (\boldsymbol{\mu}_i - (\boldsymbol{\mu}_p)_i)^T (\boldsymbol{\mu}_i - (\boldsymbol{\mu}_p)_i) \tag{152}$$

$$\text{s.t.} \quad \forall_i : \boldsymbol{\mu}_i \geq \mathbf{0} \,,$$

$$\forall_j : \frac{1}{n} \sum_{i=1}^n \mu_{ij}^2 = 1 \,.$$

The feasible set is non-convex because of the quadratic equality constraint. The Lagrangian with multiplier $\boldsymbol{\lambda} \geq \mathbf{0}$ is

$$L \,=\, \frac{1}{n} \sum_{i=1}^{n} (\boldsymbol{\mu}_i \,-\, (\boldsymbol{\mu}_p)_i)^T \,(\boldsymbol{\mu}_i \,-\, (\boldsymbol{\mu}_p)_i) \,-\, \sum_{i=1}^{n} \boldsymbol{\lambda}_i^T \,\boldsymbol{\mu}_i \tag{153}$$
$$+\, \sum_j \tau_j \left( \frac{1}{n} \sum_{i=1}^{n} \mu_{ij}^2 \,-\, 1 \right) \,.$$

The Karush-Kuhn-Tucker (KKT) conditions require for the optimal solution:

$$\lambda_{ij} \,\mu_{ij} \,=\, 0 \quad \text{and} \quad \tau_j \left( \frac{1}{n} \sum_{i=1}^{n} \mu_{ij}^2 \,-\, 1 \right) \,=\, 0 \,. \tag{154}$$

The derivative of $L$ with respect to $\mu_{ij}$ is

$$\frac{\partial L}{\partial \mu_{ij}} \,=\, \frac{2}{n} \,(\mu_{ij} \,-\, (\mu_p)_{ij}) \,-\, \lambda_{ij} \,+\, \frac{2}{n} \,\tau_j \,\mu_{ij} \,=\, 0 \,. \tag{155}$$

We multiply this equation by $\mu_{ij}$ and obtain:

$$\frac{2}{n} \,(\mu_{ij}^2 \,-\, (\mu_p)_{ij} \,\mu_{ij}) \,-\, \lambda_{ij} \,\mu_{ij} \,+\, \frac{2}{n} \,\tau_j \,\mu_{ij}^2 \,=\, 0 \,. \tag{156}$$

The KKT conditions give $\lambda_{ij}\mu_{ij} = 0$, therefore this term can be removed from the equation. Next we sum over $i$:

$$\frac{2}{n} \sum_{i=1}^{n} \left( \mu_{ij}^2 \,-\, (\mu_p)_{ij} \,\mu_{ij} \right) \,+\, \frac{2}{n} \sum_{i=1}^{n} \tau_j \,\mu_{ij}^2 \,=\, 0 \,. \tag{157}$$

Using the equality constraint $1/n \sum_{i=1}^{n} \mu_{ij}^2 = 1$ and dividing by 2 and gives:

$$1 \,-\, \frac{1}{n} \sum_{i=1}^{n} (\mu_p)_{ij} \,\mu_{ij} \,+\, \tau_j \,=\, 0 \,. \tag{158}$$

Solving for $\tau_j$ leads to:

$$\tau_j \,=\, \frac{1}{n} \sum_{i=1}^{n} (\mu_p)_{ij} \,\mu_{ij} \,-\, 1 \,. \tag{159}$$

We insert $\tau_j$ into Eq. (155)

$$-\, (\mu_p)_{ij} \,-\, \frac{n}{2} \lambda_{ij} \,+\, \left( \frac{1}{n} \sum_{s=1}^{n} (\mu_p)_{sj} \,\mu_{sj} \right) \,\mu_{ij} \,=\, 0 \,. \tag{160}$$

We immediately see, that if $\mu_{ij} = 0$ then $(\mu_p)_{ij} = -\frac{n}{2}\lambda_{ij} < 0$. Therefore we can assume $\mu_{ij} > 0$. Multiplying Eq. (160) with $\mu_{ij}$ and using the KKT conditions gives

$$-\, (\mu_p)_{ij} \,\mu_{ij} \,+\, \left( \frac{1}{n} \sum_{s=1}^{n} (\mu_p)_{sj} \,\mu_{sj} \right) \,\mu_{ij}^2 \,=\, 0 \,. \tag{161}$$

Therefore $(\mu_p)_{ij}\mu_{ij}$ and $\frac{1}{n}\sum_{s=1}^{n}(\mu_p)_{sj}\mu_{sj}$ have the same sign or $\mu_{ij} = 0$. Since $0 \leq \mu_{ij}$, we deduce that $(\mu_p)_{ij}$ and $\frac{1}{n}\sum_{s=1}^{n}(\mu_p)_{sj}\mu_{sj}$ have the same sign or $\mu_{ij} = 0$. Since the sum is independent of $i$, all $(\mu_p)_{ij}$ with $\mu_{ij} > 0$ have the same sign for $1 \leq i \leq n$. Solving Eq. (160) for $\mu_{ij}$ gives

$$\mu_{ij} \,=\, \frac{(\mu_p)_{ij} \,+\, \frac{n}{2}\lambda_{ij}}{\frac{1}{n} \sum_{s=1}^{n} (\mu_p)_{sj} \,\mu_{sj}} \,. \tag{162}$$

**I.** If all $(\mu_p)_{ij}$ are non-positive for $1 \le j \le l$, then the sum $\frac{1}{n} \sum_{s=1}^{n} (\mu_p)_{sj} \mu_{sj}$ is negative. From the first order derivative of the Lagrangian in Eq. (155), we can compute the second order derivative

$$\frac{\partial^2 L}{\partial \mu_{ij} \partial \mu_{ij}} = \frac{2}{n} + \frac{2}{n} \tau_j = 2 \sum_{i=1}^{n} (\mu_p)_{ij} \, \mu_{ij} < 0 \,. \tag{163}$$

We inserted the expression of Eq. (159) for $\tau_j$. Since all mixed second order derivatives are zero, the (projected) Hessian of the Lagrangian is diagonal with negative entries. Therefore it is strict negative definite. Thus, the second order necessary conditions cannot be fulfilled. The minimum is a border point of the constraints.

For each $j$ for which all $(\mu_p)_{ij}$ are non-positive for $1 \le j \le l$, optimization problem Eq. (149) defines a plane that has a normal vector in the positive orthant (hyperoctant). For such a $j$ the corresponding equality constraint defines a hypersphere. Minimization means that the plane containing the solution is parallel to the original plane and should be as close to the origin as possible. If we move the plane parallel from the origin into the positive orthant, then the first intersection with the hypersphere is

$$\mu_{ij} = \begin{cases} \sqrt{n} & \text{for} \quad j = \arg\max_{\hat{j}} \{(\mu_p)_{i\hat{j}}\} \\ 0 & \text{otherwise} \end{cases} \,. \tag{164}$$

This is the solution for $\mu_{ij}$ with $1 \le j \le l$ to our minimization problem.

**II.** If one $(\mu_p)_{ij}$ is positive, then from Eq. (160) with this $(\mu_p)_{ij}$ follows that $\frac{1}{n} \sum_{s=1}^{n} (\mu_p)_{sj} \mu_{sj}$ is positive, otherwise Eq. (160) has only negative terms on the left hand side. In particular, the second order necessary conditions are always fulfilled as Eq. (163) is positive. For $(\mu_p)_{ij} < 0$ it follows from Eq. (160) that $\lambda_{ij} > 0$ and from the KKT conditions that $\mu_{ij} = 0$. For $(\mu_p)_{ij} > 0$ it follows from Eq. (160) that $\mu_{ij} > 0$ and from the KKT conditions that $\lambda_{ij} = 0$. Therefore we define:

$$\hat{\mu}_{ij} = \begin{cases} 0 & \text{for} \quad (\mu_p)_{ij} \le 0 \\ (\mu_p)_{ij} & \text{for} \quad (\mu_p)_{ij} > 0 \end{cases} \,, \tag{165}$$

We write the solution as

$$\mu_{ij} = \frac{\hat{\mu}_{ij}}{\frac{1}{n} \sum_{s=1}^{n} (\mu_p)_{sj} \, \mu_{sj}} = \alpha_j \, \hat{\mu}_{ij} \,. \tag{166}$$

We now use the equality constraint:

$$\frac{1}{n} \sum_{i=1}^{n} \mu_{ij}^2 = \alpha_j^2 \frac{1}{n} \sum_{i=1}^{n} \hat{\mu}_{ij}^2 = 1 \,. \tag{167}$$

Solving for $\alpha_j$ gives:

$$\alpha_j = \frac{1}{\sqrt{\frac{1}{n} \sum_{i=1}^{n} \hat{\mu}_{ij}^2}} \,. \tag{168}$$

Therefore the solution is

$$\mu_{ij} = \frac{\hat{\mu}_{ij}}{\sqrt{\frac{1}{n} \sum_{i=1}^{n} \hat{\mu}_{ij}^2}} \,. \tag{169}$$

This finishes the proof.

$\square$

## 9.5 Gradient and Scaled Gradient Projection and Projected Newton

### 9.5.1 Gradient Projection Algorithm

The *projected gradient descent* or *gradient projection algorithm* Bertsekas [1976], Kelley [1999] performs first a gradient step and then projects the result to the *feasible set*. The projection onto the

feasible set is denoted by P, that is, the map that takes $\boldsymbol{\mu}$ into the nearest point (in the $L^2$-norm) in the feasible set to $\boldsymbol{\mu}$. The feasible set must be convex, however later we will introduce gradient projection methods for non-convex feasible sets.

The gradient projection method is in our case

$$\boldsymbol{\mu}_{k+1} = P\left(\boldsymbol{\mu}_k + \lambda\,\boldsymbol{\Sigma}_p^{-1}(\boldsymbol{\mu}_p - \boldsymbol{\mu}_k)\right) . \tag{170}$$

The Lipschitz constant for the gradient is $\|\boldsymbol{\Sigma}_p^{-1}\|_s = e_{\max}(\boldsymbol{\Sigma}_p^{-1})$, the largest eigenvalue of $\boldsymbol{\Sigma}_p^{-1}$. The following statement is Theorem 5.4.5 in Kelley [1999].

**Theorem 10** (Theorem 5.4.5 in Kelley [1999])**.** *The* sufficient decrease *condition*

$$D_{\mathrm{KL}}(Q(\boldsymbol{\mu}_{k+1}) \parallel p) - D_{\mathrm{KL}}(Q(\boldsymbol{\mu}_k) \parallel p) \leq \frac{-\alpha}{\lambda}\|\boldsymbol{\mu}_k - \boldsymbol{\mu}_{k+1}\|^2 \tag{171}$$

*(e.g. with $\alpha = 10^{-4}$) holds for all $\lambda$ such that*

$$0 < \lambda \leq \frac{2\,(1-\alpha)}{e_{\max}(\boldsymbol{\Sigma}_p^{-1})} . \tag{172}$$

*Proof.* See Kelley [1999]. $\square$

*Theorem 10 guarantees that we can increase the objective by gradient projection in the E-step, except the case where we already reached the maximum.*

For a fast upper bound on the maximal eigenvalue we use

$$e_{\max}(\boldsymbol{\Sigma}_p^{-1}) \leq \mathrm{Tr}(\boldsymbol{\Sigma}_p^{-1}) \tag{173}$$

and

$$e_{\max}(\boldsymbol{\Sigma}_p^{-1}) \leq \|\boldsymbol{W}\|_s^2\,\|\boldsymbol{\Psi}^{-1}\|_s - 1 , \tag{174}$$

where the latter follows from

$$\boldsymbol{\Sigma}_p^{-1} = \boldsymbol{I} + \boldsymbol{W}^T\boldsymbol{\Psi}^{-1}\boldsymbol{W} . \tag{175}$$

Improved methods for finding an appropriate $\lambda$ by line search methods have been proposed Birgin et al. [2000], Serafini et al. [2005]. We use a search with $\lambda = \beta^t$ with $t = 0, 1, 2, \ldots$ and $\beta = 2^{-1}$ or $\beta = 10^{-1}$.

A special version of the gradient projection method is the *generalized reduced method* Abadie and Carpentier [1969]. This method is able to solve our optimization problem with equality constraints. The gradient projection method has been generalized by Rosen to non-linear constraints Rosen [1961]. The gradient projection algorithm of Rosen can also be used for a region which is not convex. The idea is to linearize the nonlinear constraints and solve the problem. Subsequently a restoration move brings the solution back to the constraint boundaries. Rosen's gradient projection method was improved by Haug and Arora [1979]. *These methods guarantee that we can increase the objective in the E-step for non-convex feasible sets, except the case where we already reached the maximum.* These algorithms for non-convex feasible sets will only give a local maximum. Also the GAM algorithm will only find a local maximum.

### 9.5.2 Scaled Gradient Projection and Projected Newton Method

Both the *scaled gradient projection algorithm* and the *projected Newton method* were proposed in Bertsekas [1982]. We follow Kelley [1999].

The idea is to use a Newton update instead of the a gradient update:

$$\boldsymbol{\mu}_{k+1} = P\left(\boldsymbol{\mu}_k + \lambda\,\boldsymbol{H}^{-1}\,\boldsymbol{\Sigma}_p^{-1}(\boldsymbol{\mu}_p - \boldsymbol{\mu}_k)\right) . \tag{176}$$

$\boldsymbol{H}^{-1}$ can be an arbitrary strict positive definite matrix. If we set $\boldsymbol{H}^{-1} = \boldsymbol{\Sigma}_p$, then we have a Newton update of the *projected Newton method* Bertsekas [1982]. For $\lambda = 1$ we obtain

$$\boldsymbol{\mu}_{k+1} = P\left(\boldsymbol{\mu}_p\right) . \tag{177}$$

otherwise

$$\boldsymbol{\mu}_{k+1} = \mathrm{P}\left((1 - \lambda)\boldsymbol{\mu}_k + \lambda\boldsymbol{\mu}_p\right) . \tag{178}$$

The search direction for the unconstrained problem can be rotated by $\boldsymbol{H}^{-1}$ to be orthogonal to the direction of decrease in the inactive directions for the constrained problem.

To escape this possible problem, an $\epsilon$-active set is introduced which contains all $j$ with $\mu_j \leq \epsilon$. All columns and rows of the Hessian having an index in the $\epsilon$-active set are fixed to $\boldsymbol{e}_j$. After sorting the indices of the $\epsilon$-active set together, they form a block which is the sub-identity matrix. $\boldsymbol{H}$ is set to the Hessian $\boldsymbol{\Sigma}_p$ where the $\epsilon$-active set columns and rows are replaced by unit vectors.

The following Theorem 11 is Lemma 5.5.1 in Kelley [1999]. *Theorem 11 states that the objective decreases using the reduced Hessian in the projected Newton method for convex feasible sets.*

**Theorem 11** (Lemma 5.5.1 in Kelley [1999]). *The* sufficient decrease *condition*

$$D_{\mathrm{KL}}(Q(\boldsymbol{\mu}_{k+1}) \parallel p) - D_{\mathrm{KL}}(Q(\boldsymbol{\mu}_k) \parallel p) \leq -\alpha\left(\boldsymbol{\mu}_k - \boldsymbol{\mu}_p\right)^T \boldsymbol{\Sigma}_p^{-1}(\boldsymbol{\mu}_k - \boldsymbol{\mu}_{k+1}) \tag{179}$$

*holds for all $\lambda$ smaller than a bound depending on $\boldsymbol{H}$ and $\epsilon$.*

*Proof.* See Kelley [1999]. □

In practical applications, a proper $\lambda$ is found by line search. The *projected Newton method* uses $\lambda = 1$ to set $\epsilon$ Bertsekas [1982]:

$$\epsilon = \|\boldsymbol{\mu}_k - \mathrm{P}\left(\boldsymbol{\mu}_p\right)\| . \tag{180}$$

### 9.5.3 Combined Method

Following Kim et al. [2006], Serafini et al. [2005] we use the following very general update rule, which includes the gradient projection algorithm, the scaled gradient projection algorithm, and the projected Newton method.

We use following update for the E-step:

$$\boldsymbol{d}_{k+1} = \mathrm{P}\left(\boldsymbol{\mu}_k + \lambda\,\boldsymbol{H}^{-1}\,\boldsymbol{\Sigma}_p^{-1}(\boldsymbol{\mu}_p - \boldsymbol{\mu}_k)\right) , \tag{181}$$
$$\boldsymbol{\mu}_{k+1} = \mathrm{P}\left(\boldsymbol{\mu}_k + \gamma\left(\boldsymbol{d}_{k+1} - \boldsymbol{\mu}_k\right)\right) .$$

We have to project twice since the equality constraint produces a manifold in the parameter space.

We iterate this update until we see a decrease of the objective in the E-step:

$$D_{\mathrm{KL}}(Q_{k+1} \parallel p) - D_{\mathrm{KL}}(Q_k \parallel p) < 0 . \tag{182}$$

For the constraints we have only to optimize the mean vector $\boldsymbol{\mu}$ to ensure

$$D_{\mathrm{KL}}(Q(\boldsymbol{\mu}_{k+1}) \parallel p) - D_{\mathrm{KL}}(Q(\boldsymbol{\mu}_k) \parallel p) < 0 . \tag{183}$$

Even

$$D_{\mathrm{KL}}(Q(\boldsymbol{\mu}_{k+1}) \parallel p) = D_{\mathrm{KL}}(Q(\boldsymbol{\mu}_k) \parallel p) \tag{184}$$

can be sufficient if minimizing $\boldsymbol{\Sigma}_{k+1} = \boldsymbol{\Sigma}_p$ ensures

$$D_{\mathrm{KL}}(Q_{k+1} \parallel p) < D_{\mathrm{KL}}(Q_k \parallel p) . \tag{185}$$

We use following schedule:

1. • $\boldsymbol{H}^{-1} = \boldsymbol{\Sigma}_p$
   • $\lambda = 1$
   • $\gamma = 1$
   That is

$$\boldsymbol{\mu}_{k+1} = \mathrm{P}\left(\boldsymbol{\mu}_p\right) . \tag{186}$$

2.   • $\boldsymbol{H}^{-1} = \boldsymbol{\Sigma}_p$
     • $\lambda = 1$
     • $\gamma \in (0, 1]$
   That is

$$\boldsymbol{\mu}_{k+1} = \mathrm{P}\left((1 - \gamma)\,\boldsymbol{\mu}_k + \gamma\,\mathrm{P}\left(\boldsymbol{\mu}_p\right)\right) . \tag{187}$$

3.   • $\boldsymbol{H}^{-1} = \boldsymbol{\Sigma}_p$
     • $\lambda \in (0, 1]$
     • $\gamma = 1$
   That is

$$\boldsymbol{\mu}_{k+1} = \mathrm{P}\left((1 - \lambda)\boldsymbol{\mu}_k + \lambda\boldsymbol{\mu}_p\right) . \tag{188}$$

4.   • $\boldsymbol{H}^{-1} = \boldsymbol{\Sigma}_p$
     • $\lambda \in (0, 1]$
     • $\gamma =\in (0, 1]$
   That is

$$\boldsymbol{\mu}_{k+1} = \mathrm{P}\left((1 - \gamma)\,\boldsymbol{\mu}_k + \gamma\,\mathrm{P}\left((1 - \lambda)\boldsymbol{\mu}_k + \lambda\boldsymbol{\mu}_p\right)\right) . \tag{189}$$

5.   • $\boldsymbol{H}^{-1} = \mathrm{R}(\boldsymbol{\Sigma}_p)$
     • $\lambda \in (0, 1]$
     • $\gamma =\in (0, 1]$
   $\mathrm{R}(\boldsymbol{\Sigma}_p)$ denotes the reduced matrix (Hessian or a positive definite) according to the projected Newton method or the scaled gradient projection algorithm. For convex feasible sets we can guarantee at this level already an increase of the objective at the E-step.

6.   • $\boldsymbol{H}^{-1} = \boldsymbol{I}$
     • $\lambda \in (0, 1]$
     • $\gamma =\in (0, 1]$
   This is the gradient projection algorithm. In particular we include the generalized reduced method and Rosen's gradient projection method. At this step we guarantee an increase of the objective at the E-step even for non-convex feasible sets because we also use complex methods for constraint optimization.

Step 5. ensures an improvement if only using rectifying constraints according to the theory of projected Newton methods Kelley [1999]. Step 6. ensures an improvement if using both rectifying constraints and normalizing constraints, because we use known methods for constraint optimization. To set $\boldsymbol{\mu}_{k+1} = \boldsymbol{\mu}_k$ is sufficient to increase the objective at the E-step if $\boldsymbol{\Sigma}_{k+1} = \boldsymbol{\Sigma}_p$ decreases the KL divergence. However we will not always set $\boldsymbol{\mu}_{k+1} = \boldsymbol{\mu}_k$ to avoid accumulation points outside the solution set.

## 10   Alternative Gaussian Prior

We assume $\boldsymbol{h}$ is Gaussian with covariance $\boldsymbol{M}$ and mean $\boldsymbol{\xi}$

$$\boldsymbol{h} \sim \mathcal{N}\left(\boldsymbol{\xi}, \boldsymbol{M}\right) . \tag{190}$$

We derive the posterior for this prior.

The likelihood is Gaussian since a affine transformation of a Gaussian random variable is again a Gaussian random variable and the convolution of two Gaussians is Gaussian, too. Thus, $\boldsymbol{v} = \boldsymbol{W}\boldsymbol{h} + \boldsymbol{\epsilon}$ is Gaussian if $\boldsymbol{h}$ and $\boldsymbol{\epsilon}$ are both Gaussian. For the prior moments we have

$$\mathrm{E}(\boldsymbol{h}) = \boldsymbol{\xi} , \tag{191}$$

$$\mathrm{E}(\boldsymbol{h}\boldsymbol{h}^T) = \boldsymbol{M} + \boldsymbol{\xi}\,\boldsymbol{\xi}^T , \tag{192}$$

$$\mathrm{var}(\boldsymbol{h}) = \boldsymbol{M} \tag{193}$$

and for the likelihood of $\boldsymbol{v}$ we obtain the moments

$$\mathrm{E}(\boldsymbol{v}) \;=\; \boldsymbol{W}\boldsymbol{\xi}\,, \tag{194}$$

$$\mathrm{E}(\boldsymbol{v}\boldsymbol{v}^T) \;=\; \boldsymbol{W}\,\mathrm{E}(\boldsymbol{h}\boldsymbol{h}^T)\,\boldsymbol{W}^T \;+\; \boldsymbol{\Psi} \tag{195}$$

$$=\; \boldsymbol{W}\,\boldsymbol{M}\,\boldsymbol{W}^T \;+\; \boldsymbol{\Psi} \;+\; \boldsymbol{W}\,\boldsymbol{\xi}\,\boldsymbol{\xi}^T\,\boldsymbol{W}^T\,,$$

$$\mathrm{var}(\boldsymbol{v}) \;=\; \boldsymbol{W}\,\boldsymbol{M}\,\boldsymbol{W}^T \;+\; \boldsymbol{\Psi}\,. \tag{196}$$

We need some algebraic identities to derive the posterior. The Woodbury matrix identity gives

$$\boldsymbol{M}\;-\;\boldsymbol{M}\,\boldsymbol{W}^T\left(\boldsymbol{W}\,\boldsymbol{M}\,\boldsymbol{W}^T\;+\;\boldsymbol{\Psi}\right)^{-1}\boldsymbol{W}\,\boldsymbol{M}\;=\;\left(\boldsymbol{M}^{-1}\;+\;\boldsymbol{W}^T\boldsymbol{\Psi}^{-1}\boldsymbol{W}\right)^{-1}\,. \tag{197}$$

Multiplying this equation from the left hand side with $\boldsymbol{\Psi}^{-1}\boldsymbol{W}$ gives

$$\boldsymbol{\Psi}^{-1}\,\boldsymbol{W}\,\left(\boldsymbol{M}^{-1}\;+\;\boldsymbol{W}^T\boldsymbol{\Psi}^{-1}\boldsymbol{W}\right)^{-1} \tag{198}$$

$$=\;\boldsymbol{\Psi}^{-1}\,\boldsymbol{W}\,\boldsymbol{M}\;-\;\boldsymbol{\Psi}^{-1}\,\boldsymbol{W}\,\boldsymbol{M}\,\boldsymbol{W}^T\left(\boldsymbol{W}\,\boldsymbol{M}\,\boldsymbol{W}^T\;+\;\boldsymbol{\Psi}\right)^{-1}\boldsymbol{W}\,\boldsymbol{M}$$

$$=\;\boldsymbol{\Psi}^{-1}\left(\boldsymbol{W}\,\boldsymbol{M}\,\boldsymbol{W}^T\;+\;\boldsymbol{\Psi}\right)\left(\boldsymbol{W}\,\boldsymbol{M}\,\boldsymbol{W}^T\;+\;\boldsymbol{\Psi}\right)^{-1}\boldsymbol{W}\,\boldsymbol{M}\;-$$

$$\boldsymbol{\Psi}^{-1}\,\boldsymbol{W}\,\boldsymbol{M}\,\boldsymbol{W}^T\left(\boldsymbol{W}\,\boldsymbol{M}\,\boldsymbol{W}^T\;+\;\boldsymbol{\Psi}\right)^{-1}\boldsymbol{W}\,\boldsymbol{M}$$

$$=\;\left(\boldsymbol{\Psi}^{-1}\left(\boldsymbol{W}\,\boldsymbol{M}\,\boldsymbol{W}^T\;+\;\boldsymbol{\Psi}\right)\;-\;\boldsymbol{\Psi}^{-1}\,\boldsymbol{W}\,\boldsymbol{M}\,\boldsymbol{W}^T\right)\left(\boldsymbol{W}\,\boldsymbol{M}\,\boldsymbol{W}^T\;+\;\boldsymbol{\Psi}\right)^{-1}\boldsymbol{W}\,\boldsymbol{M}$$

$$=\;\left(\boldsymbol{W}\,\boldsymbol{M}\,\boldsymbol{W}^T\;+\;\boldsymbol{\Psi}\right)^{-1}\boldsymbol{W}\,\boldsymbol{M}\,.$$

It follows that

$$\boldsymbol{M}\,\boldsymbol{W}^T\left(\boldsymbol{W}\,\boldsymbol{W}^T\;+\;\boldsymbol{\Psi}\right)^{-1}\boldsymbol{a}\;=\;\left(\boldsymbol{M}^{-1}\;+\;\boldsymbol{W}^T\boldsymbol{\Psi}^{-1}\boldsymbol{W}\right)^{-1}\boldsymbol{W}^T\boldsymbol{\Psi}^{-1}\,\boldsymbol{a}\,. \tag{199}$$

The posterior $p(\boldsymbol{h}\mid\boldsymbol{v})$ is derived from Gaussian conditioning because both the likelihood $p(\boldsymbol{v})$ and the prior $p(\boldsymbol{h})$ are Gaussian distributed. The conditional distribution $p(\boldsymbol{a}\mid\boldsymbol{b})$ of two random variables $\boldsymbol{a}$ and $\boldsymbol{b}$ that both follow a Gaussian distribution is a Gaussian:

$$\boldsymbol{a}\;\sim\;\mathcal{N}\left(\boldsymbol{\mu}_a,\boldsymbol{\Sigma}_{aa}\right)\,, \tag{200}$$

$$\boldsymbol{b}\;\sim\;\mathcal{N}\left(\boldsymbol{\mu}_b,\boldsymbol{\Sigma}_{bb}\right)\,, \tag{201}$$

$$\boldsymbol{\Sigma}_{ba}\;=\;\mathrm{Cov}(\boldsymbol{b},\boldsymbol{a})\,, \tag{202}$$

$$\boldsymbol{\Sigma}_{ab}\;=\;\mathrm{Cov}(\boldsymbol{a},\boldsymbol{b})\,, \tag{203}$$

$$\boldsymbol{a}\mid\boldsymbol{b}\;\sim\;\mathcal{N}\left(\boldsymbol{\mu}_a\;+\;\boldsymbol{\Sigma}_{ab}\boldsymbol{\Sigma}_{bb}^{-1}\left(\boldsymbol{b}\;-\;\boldsymbol{\mu}_b\right)\,,\;\boldsymbol{\Sigma}_{aa}\;-\;\boldsymbol{\Sigma}_{ab}\boldsymbol{\Sigma}_{bb}^{-1}\boldsymbol{\Sigma}_{ba}\right)\,. \tag{204}$$

Therefore we need the second moments between $\boldsymbol{v}$ and $\boldsymbol{h}$:

$$\mathrm{E}(\boldsymbol{v}\boldsymbol{h}^T)\;=\;\mathrm{E}(\boldsymbol{W}\boldsymbol{h}\boldsymbol{h}^T)\;+\;\mathrm{E}(\boldsymbol{\epsilon}\boldsymbol{h}^T)\;=\;\boldsymbol{W}\left(\boldsymbol{M}\;+\;\boldsymbol{\xi}\,\boldsymbol{\xi}^T\right)\,. \tag{205}$$

The covariances between $\boldsymbol{v}$ and $\boldsymbol{h}$ are

$$\mathrm{Cov}(\boldsymbol{v},\boldsymbol{h})\;=\;\mathrm{E}(\boldsymbol{v}\boldsymbol{h}^T)\;-\;\mathrm{E}(\boldsymbol{v})\mathrm{E}(\boldsymbol{h}^T) \tag{206}$$

$$=\;\boldsymbol{W}\,\boldsymbol{M}\;+\;\boldsymbol{W}\boldsymbol{\xi}\,\boldsymbol{\xi}^T\;-\;\boldsymbol{W}\boldsymbol{\xi}\,\boldsymbol{\xi}^T\;=\;\boldsymbol{W}\,\boldsymbol{M}\,,$$

$$\mathrm{Cov}(\boldsymbol{h},\boldsymbol{v})\;=\;\mathrm{E}(\boldsymbol{h}\boldsymbol{v}^T)\;-\;\mathrm{E}(\boldsymbol{h})\mathrm{E}(\boldsymbol{v}^T)\;=\;\boldsymbol{M}\,\boldsymbol{W}^T\,. \tag{207}$$

Thus, the mean of $p(\boldsymbol{h}\mid\boldsymbol{v})$ is

$$\boldsymbol{\mu}_{\boldsymbol{h}|\boldsymbol{v}}\;=\;\boldsymbol{\xi}\;+\;\boldsymbol{M}\,\boldsymbol{W}^T\left(\boldsymbol{W}\,\boldsymbol{M}\,\boldsymbol{W}^T\;+\;\boldsymbol{\Psi}\right)^{-1}(\boldsymbol{v}\;-\;\boldsymbol{W}\boldsymbol{\xi}) \tag{208}$$

$$=\;\boldsymbol{\xi}\;+\;\left(\boldsymbol{M}^{-1}\;+\;\boldsymbol{W}^T\boldsymbol{\Psi}^{-1}\boldsymbol{W}\right)^{-1}\boldsymbol{W}^T\boldsymbol{\Psi}^{-1}\,(\boldsymbol{v}\;+\;\boldsymbol{W}\boldsymbol{\xi})$$

$$=\;\left(\boldsymbol{M}^{-1}\;+\;\boldsymbol{W}^T\boldsymbol{\Psi}^{-1}\boldsymbol{W}\right)^{-1}\left(\boldsymbol{M}^{-1}\;+\;\boldsymbol{W}^T\boldsymbol{\Psi}^{-1}\boldsymbol{W}\right)\boldsymbol{\xi}$$

$$+\;\left(\boldsymbol{M}^{-1}\;+\;\boldsymbol{W}^T\boldsymbol{\Psi}^{-1}\boldsymbol{W}\right)^{-1}\boldsymbol{W}^T\boldsymbol{\Psi}^{-1}\,(\boldsymbol{v}\;-\;\boldsymbol{W}\boldsymbol{\xi})$$

$$=\;\left(\boldsymbol{M}^{-1}\;+\;\boldsymbol{W}^T\boldsymbol{\Psi}^{-1}\boldsymbol{W}\right)^{-1}$$

$$\left(\boldsymbol{M}^{-1}\boldsymbol{\xi}\;+\;\boldsymbol{W}^T\boldsymbol{\Psi}^{-1}\boldsymbol{W}\,\boldsymbol{\xi}\;+\;\boldsymbol{W}^T\boldsymbol{\Psi}^{-1}\,\boldsymbol{v}\;-\;\boldsymbol{W}^T\boldsymbol{\Psi}^{-1}\boldsymbol{W}\boldsymbol{\xi}\right)$$

$$=\;\left(\boldsymbol{M}^{-1}\;+\;\boldsymbol{W}^T\boldsymbol{\Psi}^{-1}\boldsymbol{W}\right)^{-1}\left(\boldsymbol{W}^T\boldsymbol{\Psi}^{-1}\,\boldsymbol{v}\;+\;\boldsymbol{M}^{-1}\,\boldsymbol{\xi}\right)\,.$$

The covariance matrix of $p(\boldsymbol{h} \mid \boldsymbol{v})$ is

$$
\begin{aligned}
\boldsymbol{\Sigma}_{\boldsymbol{h}|\boldsymbol{v}} &= \boldsymbol{M} - \boldsymbol{M} \boldsymbol{W}^T \left( \boldsymbol{W} \boldsymbol{M} \boldsymbol{W}^T + \boldsymbol{\Psi} \right)^{-1} \boldsymbol{W} \boldsymbol{M} \qquad (209) \\
&= \left( \boldsymbol{M}^{-1} + \boldsymbol{W}^T \boldsymbol{\Psi}^{-1} \boldsymbol{W} \right)^{-1} .
\end{aligned}
$$

In particular, the variable $\boldsymbol{\xi}$ may be used to enforce more sparseness by setting its components to negative values. Since the covariance matrix $\boldsymbol{\Sigma}_{\boldsymbol{h}|\boldsymbol{v}}$ is positive semi-definite, we ensure that

$$
\boldsymbol{\xi}^T \left( \boldsymbol{M}^{-1} + \boldsymbol{W}^T \boldsymbol{\Psi}^{-1} \boldsymbol{W} \right)^{-1} \boldsymbol{\xi} \geq 0 . \qquad (210)
$$

If $\boldsymbol{\xi} = -\rho \mathbf{1}$ ($\mathbf{1}$ is the vector with all components being one), then the largest absolute components of $\boldsymbol{\Sigma}_{\boldsymbol{h}|\boldsymbol{v}} \boldsymbol{\xi}$ must be negative. Thus, $\boldsymbol{\xi} = -\rho \mathbf{1}$ leads to sparser solutions.

## 11 Hyperparameters Selected for Method Assessment

The performance of rectified factor networks (RFNs) as unsupervised methods for data representation was compared with:
(1) **RFN**: rectified factor networks,
(2) **RFNn**: RFNs without normalization,
(3) **DAE**: denoising autoencoders with rectified linear units,
(4) **RBM**: restricted Boltzmann machines with Gaussian visible units and hidden binary units,
(5) **FAsp**: factor analysis with Jeffrey's prior ($p(z) \propto 1/z$) on the hidden units which is sparser than a Laplace prior,
(6) **FAlap**: factor analysis with Laplace prior on the hidden units,
(7) **ICA**: independent component analysis by FastICA Hyvärinen and Oja [1999],
(8) **SFA**: sparse factor analysis with a Laplace prior on the parameters,
(9) **FA**: standard factor analysis,
(10) **PCA**: principal component analysis.
The number of components are fixed to 50, 100, or 150 for each method. The used hyperparameters are listed in Tab. 1.

Table 1: Hyperparameters of all methods that were used to assess the performance of rectified factor networks (RFNs) as unsupervised methods for data representation.

| Method | Used hyperparameters |
| --- | --- |
| RFN | {learning rate=0.1, iterations=1000} |
| RFNn | {learning rate=0.1, iterations=1000} |
| DAE | {corruption level=0.2, learning rate=1e-04, iterations=1000} |
| RBM | {learning rate=0.01, iterations=1000} |
| FAsp | {iterations=500} |
| FAlap | {iterations=500} |
| SFA | {Laplace weight decay factor=5e-05, iterations=500} |

## 12 Data Set I

The number of components are fixed to 50, 100 or 150.

We generated nine different benchmark data sets (D1 to D9), where each data set consists of 100 instances for averaging the results. Each instance consists of 100 samples and 100 features resulting in a 100×100 data matrix. Into these data matrices, structures are implanted as biclusters **?**. A bicluster is a pattern consisting of a particular number of features which is found in a particular number of samples. The size of the bicluster is given by the number of features that form the pattern and by the number of samples in which the pattern is found. The data sets had different noise levels and different bicluster sizes. We considered large and small bicluster sizes, where large biclusters have 20–30 samples and 20–30 features, while small biclusters have 3–8 samples and 3–8 features. The signal strength (scaling factor) of a pattern in a sample was randomly chosen according to the Gaussian $\mathcal{N}(1, 1)$. Finally, to each data matrix background noise was added, where the noise is distributed according to a zero-mean Gaussian with standard deviation 1, 5, or 10. The data sets are described in Tab. 2. The remaining components of the spanning outer product vectors were drawn by $\mathcal{N}(0, 0.01)$.

Table 2: Overview over the datasets. Shown is the background noise ("noise"), the number of large biclusters ($n_1$), and the number of small biclusters ($n_2$).

|       | D1 | D2 | D3 | D4 | D5 | D6 | D7 | D8 | D9 |
|-------|----|----|----|----|----|----|----|----|----|
| noise | 1  | 5  | 10 | 1  | 5  | 10 | 1  | 5  | 10 |
| $n_1$ | 10 | 10 | 10 | 15 | 15 | 15 | 5  | 5  | 5  |
| $n_2$ | 10 | 10 | 10 | 5  | 5  | 5  | 15 | 15 | 15 |

Table 3: Comparison for 50 factors / hidden units extracted by RFN, RFN without normalization (RFNn), denoising autoencoder (DAE), restricted Boltzmann machines (RBM), factor analysis with a very sparse prior (FAsp), factor analysis with a Laplace prior (FAlap), independent component analysis (ICA), sparse factor analysis (SFA), factor analysis (FA), and principal component analysis (PCA) on nine data sets. Criteria are: sparseness of the coding units (SP), reconstruction error (ER), and the difference between the empirical and the model covariance matrix (CO). The lower right column block gives the average SP (%), ER and CO. Results reported here, are the mean together with the standard deviation of 100 instances. The maximal value in the table and the maximal standard deviation was set to 999 and to 99, respectively.

| | D1 | | | D2 | | | D3 | | | D4 | | | D5 | | |
|---|---|---|---|---|---|---|---|---|---|---|---|---|---|---|---|
| | SP | ER | CO | SP | ER | CO | SP | ER | CO | SP | ER | CO | SP | ER | CO |
| RFN | $74\pm0$ | $58\pm1$ | $5\pm0$ | $75\pm0$ | $233\pm3$ | $66\pm1$ | $75\pm0$ | $456\pm5$ | $253\pm6$ | $74\pm0$ | $63\pm1$ | $6\pm1$ | $75\pm0$ | $236\pm3$ | $68\pm2$ |
| RFNn | $73\pm0$ | $85\pm3$ | $13\pm2$ | $75\pm0$ | $272\pm3$ | $85\pm2$ | $75\pm0$ | $531\pm6$ | $321\pm7$ | $72\pm0$ | $95\pm4$ | $17\pm2$ | $74\pm0$ | $276\pm4$ | $89\pm3$ |
| DAE | $65\pm0$ | $65\pm2$ | — | $66\pm0$ | $233\pm2$ | — | $66\pm0$ | $456\pm4$ | — | $65\pm1$ | $71\pm2$ | — | $66\pm0$ | $237\pm2$ | — |
| RBM | $25\pm2$ | $86\pm3$ | — | $11\pm1$ | $287\pm3$ | — | $10\pm1$ | $558\pm5$ | — | $25\pm2$ | $94\pm3$ | — | $11\pm1$ | $292\pm3$ | — |
| FAsp | $39\pm1$ | $232\pm31$ | $654\pm99$ | $40\pm1$ | $999\pm41$ | $999\pm99$ | $41\pm1$ | $999\pm99$ | $999\pm99$ | $38\pm1$ | $318\pm33$ | $999\pm99$ | $40\pm1$ | $999\pm48$ | $999\pm99$ |
| FAlap | $4\pm0$ | $53\pm2$ | $144\pm36$ | $4\pm0$ | $224\pm5$ | $185\pm5$ | $5\pm0$ | $439\pm9$ | $692\pm16$ | $4\pm0$ | $55\pm2$ | $180\pm39$ | $4\pm0$ | $226\pm5$ | $192\pm6$ |
| ICA | $2\pm0$ | $34\pm0$ | — | $2\pm0$ | $164\pm2$ | — | $2\pm0$ | $324\pm4$ | — | $2\pm0$ | $35\pm0$ | — | $2\pm0$ | $166\pm2$ | — |
| SFA | $1\pm0$ | $42\pm1$ | $11\pm2$ | $1\pm0$ | $206\pm4$ | $56\pm2$ | $1\pm0$ | $406\pm9$ | $215\pm7$ | $1\pm0$ | $42\pm1$ | $13\pm2$ | $1\pm0$ | $208\pm4$ | $58\pm2$ |
| FA | $1\pm0$ | $42\pm1$ | $6\pm1$ | $1\pm0$ | $206\pm4$ | $54\pm2$ | $1\pm0$ | $407\pm8$ | $210\pm6$ | $1\pm0$ | $42\pm1$ | $8\pm1$ | $1\pm0$ | $208\pm4$ | $56\pm2$ |
| PCA | $1\pm0$ | $34\pm0$ | — | $0\pm0$ | $164\pm2$ | — | $0\pm0$ | $324\pm4$ | — | $1\pm0$ | $35\pm0$ | — | $0\pm0$ | $166\pm2$ | — |

| | D6 | | | D7 | | | D8 | | | D9 | | | average | | |
|---|---|---|---|---|---|---|---|---|---|---|---|---|---|---|---|
| | SP | ER | CO | SP | ER | CO | SP | ER | CO | SP | ER | CO | SP | ER | CO |
| RFN | $75\pm0$ | $458\pm5$ | $256\pm6$ | $75\pm0$ | $53\pm1$ | $4\pm1$ | $75\pm0$ | $230\pm3$ | $64\pm1$ | $75\pm0$ | $454\pm5$ | $251\pm5$ | $75\pm0$ | $249\pm3$ | $108\pm3$ |
| RFNn | $75\pm0$ | $532\pm6$ | $323\pm7$ | $73\pm0$ | $73\pm3$ | $10\pm2$ | $75\pm0$ | $268\pm3$ | $82\pm2$ | $75\pm0$ | $528\pm6$ | $317\pm7$ | $74\pm0$ | $295\pm4$ | $140\pm4$ |
| DAE | $66\pm0$ | $458\pm4$ | — | $65\pm0$ | $58\pm1$ | — | $66\pm0$ | $230\pm2$ | — | $66\pm0$ | $453\pm5$ | — | $66\pm0$ | $251\pm3$ | — |
| RBM | $10\pm1$ | $561\pm5$ | — | $23\pm2$ | $76\pm2$ | — | $11\pm1$ | $282\pm3$ | — | $10\pm1$ | $555\pm5$ | — | $15\pm1$ | $310\pm4$ | — |
| FAsp | $40\pm2$ | $999\pm99$ | $999\pm99$ | $39\pm1$ | $152\pm26$ | $345\pm99$ | $40\pm1$ | $999\pm31$ | $999\pm99$ | $41\pm1$ | $999\pm99$ | $999\pm99$ | $40\pm1$ | $999\pm63$ | $999\pm99$ |
| FAlap | $5\pm0$ | $443\pm9$ | $701\pm15$ | $4\pm0$ | $50\pm2$ | $110\pm37$ | $4\pm0$ | $221\pm5$ | $177\pm4$ | $5\pm0$ | $439\pm10$ | $686\pm15$ | $4\pm0$ | $239\pm6$ | $341\pm19$ |
| ICA | $2\pm0$ | $325\pm4$ | — | $2\pm0$ | $34\pm0$ | — | $2\pm0$ | $163\pm2$ | — | $2\pm0$ | $322\pm4$ | — | $2\pm0$ | $174\pm2$ | — |
| SFA | $1\pm0$ | $408\pm9$ | $217\pm7$ | $1\pm0$ | $42\pm1$ | $8\pm2$ | $1\pm0$ | $204\pm4$ | $54\pm2$ | $1\pm0$ | $405\pm9$ | $213\pm7$ | $1\pm0$ | $218\pm5$ | $94\pm3$ |
| FA | $1\pm0$ | $409\pm9$ | $212\pm7$ | $1\pm0$ | $42\pm1$ | $4\pm1$ | $1\pm0$ | $205\pm4$ | $53\pm2$ | $1\pm0$ | $405\pm8$ | $208\pm6$ | $1\pm0$ | $218\pm4$ | $90\pm3$ |
| PCA | $0\pm0$ | $325\pm4$ | — | $1\pm0$ | $34\pm0$ | — | $0\pm0$ | $163\pm2$ | — | $0\pm0$ | $322\pm4$ | — | $0\pm0$ | $174\pm2$ | — |

Table 4: Comparison for 100 factors / hidden units extracted by RFN, RFN without normalization (RFNn), denoising autoencoder (DAE), restricted Boltzmann machines (RBM), factor analysis with a very sparse prior (FAsp), factor analysis with a Laplace prior (FAlap), independent component analysis (ICA), sparse factor analysis (SFA), factor analysis (FA), and principal component analysis (PCA) on nine data sets. Criteria are: sparseness of the coding units (SP), reconstruction error (ER), and the difference between the empirical and the model covariance matrix (CO). The lower right column block gives the average SP (%), ER and CO. Results reported here, are the mean together with the standard deviation of 100 instances. The maximal value in the table and the maximal standard deviation was set to 999 and to 99, respectively.

| | D1 | | | D2 | | | D3 | | | D4 | | | D5 | | |
|---|---|---|---|---|---|---|---|---|---|---|---|---|---|---|---|
| | SP | ER | CO | SP | ER | CO | SP | ER | CO | SP | ER | CO | SP | ER | CO |
| RFN | 79±1 | 23±3 | 2±0 | 82±1 | 63±9 | 16±3 | 82±1 | 120±17 | 61±15 | 78±1 | 27±3 | 2±1 | 82±1 | 62±7 | 16±3 |
| RFNn | 77±0 | 61±4 | 6±1 | 80±0 | 169±4 | 36±2 | 80±0 | 326±8 | 135±6 | 76±1 | 73±4 | 9±2 | 79±1 | 171±5 | 37±2 |
| DAE | 67±0 | 48±2 | — | 70±0 | 134±1 | — | 70±0 | 260±2 | — | 67±0 | 54±2 | — | 70±0 | 137±1 | — |
| RBM | 14±1 | 81±3 | — | 4±0 | 266±3 | — | 4±0 | 514±6 | — | 15±1 | 88±2 | — | 4±0 | 270±3 | — |
| FAsp | 72±0 | 233±32 | 499±99 | 62±0 | 999±43 | 999±99 | 56±0 | 999±99 | 999±99 | 71±0 | 320±34 | 878±99 | 62±0 | 999±49 | 999±99 |
| FAlap | 6±0 | 27±3 | 202±17 | 6±0 | 38±3 | 756±33 | 6±0 | 74±5 | 999±83 | 6±0 | 31±3 | 274±23 | 6±0 | 39±3 | 778±34 |
| ICA | 3±2 | 0±0 | — | 3±1 | 0±0 | — | 3±1 | 0±0 | — | 3±2 | 0±0 | — | 3±1 | 0±0 | — |
| SFA | 1±0 | 6±0 | 30±5 | 1±0 | 14±0 | 68±3 | 1±0 | 28±1 | 243±8 | 1±0 | 8±0 | 38±5 | 1±0 | 15±0 | 72±3 |
| FA | 1±0 | 6±0 | 18±3 | 1±0 | 14±0 | 50±2 | 1±0 | 28±1 | 182±7 | 1±0 | 8±0 | 24±4 | 1±0 | 15±0 | 52±2 |
| PCA | 4±0 | 0±0 | — | 2±0 | 0±0 | — | 1±0 | 0±0 | — | 4±0 | 0±0 | — | 2±0 | 0±0 | — |

| | D6 | | | D7 | | | D8 | | | D9 | | | average | | |
|---|---|---|---|---|---|---|---|---|---|---|---|---|---|---|---|
| | SP | ER | CO | SP | ER | CO | SP | ER | CO | SP | ER | CO | SP | ER | CO |
| RFN | 82±1 | 120±16 | 60±13 | 80±1 | 18±2 | 1±0 | 82±1 | 61±7 | 15±3 | 82±1 | 122±13 | 60±11 | 81±1 | 68±9 | 26±6 |
| RFNn | 80±0 | 329±7 | 137±6 | 78±0 | 49±3 | 4±1 | 80±0 | 165±4 | 34±1 | 80±0 | 325±7 | 134±6 | 79±0 | 185±5 | 59±3 |
| DAE | 70±0 | 261±2 | — | 68±0 | 39±2 | — | 70±0 | 132±1 | — | 70±0 | 259±2 | — | 69±0 | 147±2 | — |
| RBM | 4±0 | 517±6 | — | 12±1 | 71±2 | — | 4±0 | 261±3 | — | 4±0 | 512±5 | — | 7±1 | 287±4 | — |
| FAsp | 56±1 | 999±99 | 999±99 | 73±0 | 149±28 | 237±62 | 62±0 | 999±34 | 999±99 | 56±0 | 999±99 | 999±99 | 63±0 | 999±65 | 999±99 |
| FAlap | 6±0 | 74±6 | 999±91 | 6±0 | 22±3 | 134±14 | 6±0 | 37±2 | 733±28 | 6±0 | 73±6 | 999±84 | 6±0 | 46±4 | 985±45 |
| ICA | 3±1 | 0±0 | — | 3±2 | 0±0 | — | 3±1 | 0±0 | — | 3±1 | 0±0 | — | 3±1 | 0±0 | — |
| SFA | 1±0 | 28±1 | 247±8 | 1±0 | 5±0 | 21±5 | 1±0 | 14±0 | 64±2 | 1±0 | 27±1 | 240±7 | 1±0 | 16±1 | 114±5 |
| FA | 1±0 | 28±1 | 184±8 | 1±0 | 5±0 | 11±3 | 1±0 | 14±0 | 47±2 | 1±0 | 27±1 | 179±7 | 1±0 | 16±1 | 83±4 |
| PCA | 1±0 | 0±0 | — | 4±0 | 0±0 | — | 2±0 | 0±0 | — | 1±0 | 0±0 | — | 2±0 | 0±0 | — |

Table 5: Comparison for 150 factors / hidden units extracted by RFN, RFN without normalization (RFNn), denoising autoencoder (DAE), restricted Boltzmann machines (RBM), factor analysis with a very sparse prior (FAsp), factor analysis with a Laplace prior (FAlap), independent component analysis (ICA), sparse factor analysis (SFA), factor analysis (FA), and principal component analysis (PCA) on nine data sets. Criteria are: sparseness of the coding units (SP), reconstruction error (ER), and the difference between the empirical and the model covariance matrix (CO). The lower right column block gives the average SP (%), ER and CO. Results reported here, are the mean together with the standard deviation of 100 instances. The maximal value in the table and the maximal standard deviation was set to 999 and to 99, respectively.

| | D1 SP | D1 ER | D1 CO | D2 SP | D2 ER | D2 CO | D3 SP | D3 ER | D3 CO | D4 SP | D4 ER | D4 CO | D5 SP | D5 ER | D5 CO |
|---|---|---|---|---|---|---|---|---|---|---|---|---|---|---|---|
| RFN | 83±1 | 7±2 | 0±1 | 86±0 | 15±1 | 3±1 | 86±2 | 33±20 | 18±23 | 83±1 | 9±2 | 1±0 | 86±1 | 15±3 | 4±1 |
| RFNn | 79±0 | 48±3 | 4±1 | 81±0 | 129±3 | 21±1 | 81±0 | 250±7 | 80±4 | 78±0 | 60±4 | 6±1 | 81±0 | 131±3 | 22±1 |
| DAE | 68±0 | 44±2 | — | 72±0 | 118±1 | — | 72±0 | 229±2 | — | 68±0 | 50±2 | — | 72±0 | 120±2 | — |
| RBM | 10±1 | 81±3 | — | 3±0 | 265±3 | — | 3±0 | 514±6 | — | 10±1 | 88±2 | — | 3±0 | 270±4 | — |
| FAsp | 83±1 | 233±32 | 340±71 | 79±0 | 999±43 | 999±99 | 77±0 | 999±99 | 999±99 | 81±1 | 320±34 | 574±99 | 79±1 | 999±49 | 999±99 |
| FAlap | 4±0 | 27±3 | 295±25 | 4±0 | 38±3 | 791±41 | 3±0 | 74±5 | 999±91 | 4±0 | 31±3 | 394±31 | 4±0 | 39±3 | 817±39 |
| ICA | 3±2 | 0±0 | — | 3±1 | 0±0 | — | 3±1 | 0±0 | — | 3±2 | 0±0 | — | 3±1 | 0±0 | — |
| SFA | 1±0 | 6±0 | 49±7 | 1±0 | 14±0 | 173±4 | 1±0 | 28±1 | 632±10 | 1±0 | 8±0 | 61±7 | 1±0 | 15±0 | 181±5 |
| FA | 1±0 | 6±0 | 40±5 | 1±0 | 14±0 | 160±4 | 1±0 | 28±1 | 590±10 | 1±0 | 8±0 | 51±6 | 1±0 | 15±0 | 168±4 |
| PCA | 4±0 | 0±0 | — | 2±0 | 0±0 | — | 1±0 | 0±0 | — | 4±0 | 0±0 | — | 2±0 | 0±0 | — |

| | D6 SP | D6 ER | D6 CO | D7 SP | D7 ER | D7 CO | D8 SP | D8 ER | D8 CO | D9 SP | D9 ER | D9 CO | average SP | average ER | average CO |
|---|---|---|---|---|---|---|---|---|---|---|---|---|---|---|---|
| RFN | 86±1 | 30±13 | 15±16 | 84±2 | 5±3 | 0±1 | 86±0 | 14±1 | 3±1 | 86±1 | 30±8 | 15±9 | 85±1 | 17±6 | 7±6 |
| RFNn | 81±0 | 251±6 | 81±3 | 80±0 | 37±3 | 2±0 | 81±0 | 126±3 | 20±1 | 81±0 | 248±6 | 79±3 | 80±0 | 142±4 | 35±2 |
| DAE | 72±0 | 230±2 | — | 70±0 | 36±2 | — | 72±0 | 116±1 | — | 72±0 | 227±2 | — | 71±0 | 130±2 | — |
| RBM | 3±0 | 516±6 | — | 8±1 | 71±2 | — | 3±0 | 260±4 | — | 3±0 | 511±5 | — | 5±0 | 286±4 | — |
| FAsp | 77±0 | 999±99 | 999±99 | 84±0 | 149±28 | 168±55 | 80±0 | 999±34 | 999±99 | 77±1 | 999±99 | 999±99 | 80±0 | 999±65 | 999±99 |
| FAlap | 3±0 | 74±6 | 999±97 | 4±0 | 22±3 | 198±17 | 4±0 | 37±2 | 768±40 | 3±0 | 73±6 | 999±93 | 4±0 | 46±4 | 976±53 |
| ICA | 3±1 | 0±0 | — | 3±2 | 0±0 | — | 3±1 | 0±0 | — | 3±1 | 0±0 | — | 3±1 | 0±0 | — |
| SFA | 1±0 | 28±1 | 640±11 | 1±0 | 5±0 | 34±6 | 1±0 | 14±0 | 164±3 | 1±0 | 27±1 | 625±9 | 1±0 | 16±1 | 285±7 |
| FA | 1±0 | 28±1 | 596±10 | 1±0 | 5±0 | 27±5 | 1±0 | 14±0 | 153±3 | 1±0 | 27±1 | 583±9 | 1±0 | 16±1 | 263±6 |
| PCA | 1±0 | 0±0 | — | 4±0 | 0±0 | — | 2±0 | 0±0 | — | 1±0 | 0±0 | — | 2±0 | 0±0 | — |

## 13  Data Set II

This data sets was generate as described in Section 12, but instead of drawing the remaining components of the spanning outer product vectors from $\mathcal{N}(0, 0.01)$, they were now drawn from $\mathcal{N}(0, 0.5)$.

Table 6: Comparison for 50 factors / hidden units extracted by RFN, RFN without normalization (RFNn), denoising autoencoder (DAE), restricted Boltzmann machines (RBM), factor analysis with a very sparse prior (FAsp), factor analysis with a Laplace prior (FAlap), independent component analysis (ICA), sparse factor analysis (SFA), factor analysis (FA), and principal component analysis (PCA) on nine data sets. Criteria are: sparseness of the coding units (SP), reconstruction error (ER), and the difference between the empirical and the model covariance matrix (CO). The lower right column block gives the average SP (%), ER and CO. Results reported here, are the mean together with the standard deviation of 100 instances. The maximal value in the table and the maximal standard deviation was set to 999 and to 99, respectively.

| | D1 | | | D2 | | | D3 | | | D4 | | | D5 | | |
|---|---|---|---|---|---|---|---|---|---|---|---|---|---|---|---|
| | SP | ER | CO | SP | ER | CO | SP | ER | CO | SP | ER | CO | SP | ER | CO |
| RFN | 72±1 | 74±2 | 11±1 | 75±0 | 240±3 | 72±2 | 75±0 | 462±5 | 260±6 | 72±1 | 79±2 | 12±1 | 75±0 | 244±3 | 75±2 |
| RFNn | 68±1 | 122±5 | 32±4 | 74±0 | 285±4 | 97±3 | 74±0 | 537±7 | 331±8 | 65±1 | 144±6 | 48±6 | 74±0 | 290±4 | 102±4 |
| DAE | 61±0 | 82±2 | — | 66±0 | 243±2 | — | 66±0 | 461±4 | — | 60±0 | 88±2 | — | 66±0 | 247±3 | — |
| RBM | 22±1 | 106±3 | — | 11±1 | 301±3 | — | 10±1 | 566±6 | — | 22±1 | 113±3 | — | 11±1 | 308±4 | — |
| FAsp | 37±1 | 469±38 | 999±99 | 40±1 | 999±50 | 999±99 | 40±2 | 999±99 | 999±99 | 37±1 | 610±44 | 999±99 | 40±1 | 999±58 | 999±99 |
| FAlap | 4±0 | 50±1 | 392±66 | 4±0 | 228±5 | 135±13 | 5±0 | 443±9 | 406±18 | 4±0 | 51±1 | 477±63 | 4±0 | 230±6 | 147±18 |
| ICA | 2±0 | 35±0 | — | 2±0 | 168±2 | — | 2±0 | 327±4 | — | 2±0 | 35±0 | — | 2±0 | 170±2 | — |
| SFA | 1±0 | 42±1 | 26±3 | 1±0 | 210±5 | 61±2 | 1±0 | 409±8 | 220±6 | 1±0 | 41±1 | 32±4 | 1±0 | 211±5 | 63±2 |
| FA | 1±0 | 42±1 | 13±2 | 1±0 | 210±4 | 58±2 | 1±0 | 409±8 | 214±6 | 1±0 | 41±1 | 17±2 | 1±0 | 212±5 | 60±2 |
| PCA | 0±0 | 35±0 | — | 0±0 | 168±2 | — | 0±0 | 327±4 | — | 0±0 | 35±0 | — | 0±0 | 170±2 | — |

| | D6 | | | D7 | | | D8 | | | D9 | | | average | | |
|---|---|---|---|---|---|---|---|---|---|---|---|---|---|---|---|
| | SP | ER | CO | SP | ER | CO | SP | ER | CO | SP | ER | CO | SP | ER | CO |
| RFN | 75±0 | 464±5 | 264±6 | 73±0 | 68±2 | 9±1 | 75±0 | 237±3 | 69±1 | 75±0 | 459±5 | 257±6 | 74±0 | 259±3 | 114±3 |
| RFNn | 74±0 | 541±6 | 336±8 | 71±1 | 106±4 | 23±3 | 74±0 | 279±3 | 91±2 | 75±0 | 533±6 | 325±8 | 72±1 | 315±5 | 154±5 |
| DAE | 66±0 | 465±4 | — | 62±0 | 75±2 | — | 66±0 | 238±2 | — | 66±0 | 458±4 | — | 64±0 | 262±3 | — |
| RBM | 10±1 | 570±6 | — | 20±1 | 97±3 | — | 11±1 | 294±3 | — | 10±1 | 562±5 | — | 14±1 | 324±4 | — |
| FAsp | 41±1 | 999±99 | 999±99 | 38±1 | 335±32 | 999±99 | 41±1 | 999±40 | 999±99 | 41±1 | 999±99 | 999±99 | 39±1 | 999±69 | 999±99 |
| FAlap | 5±0 | 447±9 | 413±19 | 4±0 | 49±1 | 292±57 | 4±0 | 227±5 | 123±11 | 5±0 | 443±9 | 401±17 | 4±0 | 241±5 | 310±31 |
| ICA | 2±0 | 329±4 | — | 2±0 | 35±0 | — | 2±0 | 167±2 | — | 2±0 | 325±4 | — | 2±0 | 177±2 | — |
| SFA | 1±0 | 412±8 | 223±7 | 1±0 | 42±1 | 19±3 | 1±0 | 209±4 | 59±2 | 1±0 | 408±9 | 218±7 | 1±0 | 221±5 | 102±4 |
| FA | 1±0 | 412±8 | 217±7 | 1±0 | 42±1 | 10±1 | 1±0 | 209±4 | 57±2 | 1±0 | 409±9 | 213±7 | 1±0 | 221±5 | 95±3 |
| PCA | 0±0 | 329±4 | — | 0±0 | 35±0 | — | 0±0 | 167±2 | — | 0±0 | 325±4 | — | 0±0 | 177±2 | — |

Table 7: Comparison for 100 factors / hidden units extracted by RFN, RFN without normalization (RFNn), denoising autoencoder (DAE), restricted Boltzmann machines (RBM), factor analysis with a very sparse prior (FAsp), factor analysis with a Laplace prior (FAlap), independent component analysis (ICA), sparse factor analysis (SFA), factor analysis (FA), and principal component analysis (PCA) on nine data sets. Criteria are: sparseness of the coding units (SP), reconstruction error (ER), and the difference between the empirical and the model covariance matrix (CO). The lower right column block gives the average SP (%), ER and CO. Results reported here, are the mean together with the standard deviation of 100 instances. The maximal value in the table and the maximal standard deviation was set to 999 and to 99, respectively.

| | D1 | | | D2 | | | D3 | | | D4 | | | D5 | | |
|---|---|---|---|---|---|---|---|---|---|---|---|---|---|---|---|
| | SP | ER | CO | SP | ER | CO | SP | ER | CO | SP | ER | CO | SP | ER | CO |
| RFN | 76±1 | 34±3 | 4±1 | 82±1 | 67±8 | 18±3 | 82±1 | 124±16 | 63±12 | 75±1 | 38±3 | 5±1 | 82±1 | 69±10 | 19±5 |
| RFNn | 71±1 | 110±7 | 25±4 | 79±0 | 180±5 | 42±2 | 80±0 | 331±8 | 139±7 | 65±2 | 143±9 | 47±8 | 79±0 | 185±5 | 45±3 |
| DAE | 63±0 | 66±2 | — | 70±0 | 142±2 | — | 70±0 | 264±3 | — | 62±0 | 73±2 | — | 70±0 | 146±2 | — |
| RBM | 12±1 | 100±3 | — | 5±0 | 282±4 | — | 4±0 | 522±6 | — | 12±1 | 106±3 | — | 5±1 | 288±4 | — |
| FAsp | 71±0 | 474±38 | 999±99 | 62±0 | 999±53 | 999±99 | 56±0 | 999±99 | 999±99 | 70±0 | 616±44 | 999±99 | 62±0 | 999±60 | 999±99 |
| FAlap | 6±0 | 21±2 | 425±28 | 6±0 | 40±2 | 827±35 | 6±0 | 75±6 | 999±99 | 6±0 | 23±2 | 523±32 | 6±0 | 42±3 | 865±43 |
| ICA | 3±2 | 0±0 | — | 3±1 | 0±0 | — | 3±1 | 0±0 | — | 3±2 | 0±0 | — | 3±1 | 0±0 | — |
| SFA | 1±0 | 10±0 | 71±7 | 1±0 | 15±0 | 84±4 | 1±0 | 28±1 | 254±8 | 1±0 | 12±0 | 87±8 | 1±0 | 16±0 | 92±5 |
| FA | 1±0 | 10±0 | 48±5 | 1±0 | 15±0 | 59±3 | 1±0 | 28±1 | 189±7 | 1±0 | 12±1 | 61±6 | 1±0 | 16±0 | 64±3 |
| PCA | 4±0 | 0±0 | — | 2±0 | 0±0 | — | 1±0 | 0±0 | — | 3±0 | 0±0 | — | 2±0 | 0±0 | — |

| | D6 | | | D7 | | | D8 | | | D9 | | | average | | |
|---|---|---|---|---|---|---|---|---|---|---|---|---|---|---|---|
| | SP | ER | CO | SP | ER | CO | SP | ER | CO | SP | ER | CO | SP | ER | CO |
| RFN | 82±1 | 127±17 | 65±14 | 77±1 | 30±3 | 3±1 | 82±1 | 64±8 | 17±4 | 82±1 | 123±15 | 62±13 | 80±1 | 75±9 | 28±6 |
| RFNn | 80±0 | 334±8 | 141±7 | 74±1 | 86±4 | 14±2 | 79±0 | 174±4 | 39±2 | 80±0 | 329±7 | 137±6 | 76±1 | 208±6 | 70±5 |
| DAE | 70±0 | 266±2 | — | 64±0 | 57±2 | — | 70±0 | 138±1 | — | 70±0 | 262±2 | — | 68±0 | 157±2 | — |
| RBM | 4±0 | 527±6 | — | 11±1 | 92±2 | — | 4±0 | 274±4 | — | 4±0 | 518±6 | — | 7±1 | 301±4 | — |
| FAsp | 56±0 | 999±99 | 999±99 | 71±0 | 338±33 | 999±99 | 62±1 | 999±42 | 999±99 | 56±1 | 999±99 | 999±99 | 63±0 | 999±74 | 999±99 |
| FAlap | 6±0 | 75±6 | 999±89 | 6±0 | 18±2 | 337±24 | 6±0 | 40±3 | 793±37 | 6±0 | 74±6 | 999±89 | 6±0 | 45±3 | 999±53 |
| ICA | 3±1 | 0±0 | — | 3±1 | 0±0 | — | 3±1 | 0±0 | — | 3±1 | 0±0 | — | 3±1 | 0±0 | — |
| SFA | 1±0 | 28±1 | 260±9 | 1±0 | 8±0 | 52±7 | 1±0 | 15±0 | 76±3 | 1±0 | 28±1 | 248±7 | 1±0 | 18±1 | 136±6 |
| FA | 1±0 | 28±1 | 193±8 | 1±0 | 8±0 | 33±5 | 1±0 | 15±0 | 54±2 | 1±0 | 28±1 | 185±6 | 1±0 | 18±1 | 99±5 |
| PCA | 1±0 | 0±0 | — | 4±0 | 0±0 | — | 2±0 | 0±0 | — | 1±0 | 0±0 | — | 2±0 | 0±0 | — |

Table 8: Comparison for 150 factors / hidden units extracted by RFN, RFN without normalization (RFNn), denoising autoencoder (DAE), restricted Boltzmann machines (RBM), factor analysis with a very sparse prior (FAsp), factor analysis with a Laplace prior (FAlap), independent component analysis (ICA), sparse factor analysis (SFA), factor analysis (FA), and principal component analysis (PCA) on nine data sets. Criteria are: sparseness of the factors (SP) reported in %, reconstruction error (ER), and the difference between the empirical and the model covariance matrix (CO). The lower right column block gives the average SP (%), ER and CO. Results reported here, are the mean together with the standard deviation of 100 instances. The maximal value in the table and the maximal standard deviation was set to 999 and to 99, respectively.

|  | D1 | | | D2 | | | D3 | | | D4 | | | D5 | | |
|---|---|---|---|---|---|---|---|---|---|---|---|---|---|---|---|
|  | SP | ER | CO | SP | ER | CO | SP | ER | CO | SP | ER | CO | SP | ER | CO |
| RFN | $81\pm1$ | $12\pm2$ | $1\pm1$ | $86\pm0$ | $16\pm1$ | $4\pm1$ | $86\pm0$ | $29\pm4$ | $15\pm5$ | $80\pm1$ | $15\pm5$ | $2\pm2$ | $86\pm1$ | $17\pm5$ | $5\pm3$ |
| RFNn | $72\pm1$ | $100\pm8$ | $19\pm4$ | $80\pm0$ | $137\pm4$ | $24\pm1$ | $81\pm0$ | $254\pm6$ | $83\pm4$ | $66\pm0$ | $113\pm3$ | $52\pm5$ | $80\pm0$ | $141\pm4$ | $26\pm2$ |
| DAE | $64\pm0$ | $62\pm2$ | — | $71\pm0$ | $125\pm2$ | — | $72\pm0$ | $232\pm2$ | — | $63\pm0$ | $69\pm2$ | — | $71\pm0$ | $129\pm2$ | — |
| RBM | $8\pm0$ | $101\pm3$ | — | $4\pm0$ | $282\pm4$ | — | $3\pm0$ | $521\pm6$ | — | $8\pm0$ | $106\pm3$ | — | $4\pm0$ | $289\pm4$ | — |
| FAsp | $81\pm1$ | $474\pm38$ | $999\pm99$ | $79\pm0$ | $999\pm53$ | $999\pm99$ | $77\pm1$ | $999\pm99$ | $999\pm99$ | $80\pm1$ | $616\pm44$ | $999\pm99$ | $79\pm1$ | $999\pm60$ | $999\pm99$ |
| FAlap | $4\pm0$ | $21\pm2$ | $607\pm34$ | $4\pm0$ | $40\pm2$ | $879\pm40$ | $3\pm0$ | $75\pm6$ | $999\pm96$ | $4\pm0$ | $23\pm2$ | $749\pm42$ | $4\pm0$ | $42\pm3$ | $926\pm45$ |
| ICA | $3\pm2$ | $0\pm0$ | — | $3\pm1$ | $0\pm0$ | — | $3\pm1$ | $0\pm0$ | — | $3\pm2$ | $0\pm0$ | — | $3\pm1$ | $0\pm0$ | — |
| SFA | $1\pm0$ | $10\pm0$ | $103\pm9$ | $1\pm0$ | $15\pm0$ | $204\pm7$ | $1\pm0$ | $28\pm1$ | $656\pm12$ | $1\pm0$ | $12\pm0$ | $126\pm10$ | $1\pm0$ | $16\pm0$ | $220\pm8$ |
| FA | $1\pm0$ | $10\pm0$ | $87\pm8$ | $1\pm0$ | $15\pm0$ | $187\pm5$ | $1\pm0$ | $28\pm1$ | $611\pm11$ | $1\pm0$ | $12\pm1$ | $108\pm9$ | $1\pm0$ | $16\pm0$ | $200\pm6$ |
| PCA | $4\pm0$ | $0\pm0$ | — | $2\pm0$ | $0\pm0$ | — | $2\pm0$ | $0\pm0$ | — | $3\pm0$ | $0\pm0$ | — | $2\pm0$ | $0\pm0$ | — |

|  | D6 | | | D7 | | | D8 | | | D9 | | | average | | |
|---|---|---|---|---|---|---|---|---|---|---|---|---|---|---|---|
|  | SP | ER | CO | SP | ER | CO | SP | ER | CO | SP | ER | CO | SP | ER | CO |
| RFN | $86\pm1$ | $29\pm7$ | $15\pm6$ | $82\pm1$ | $10\pm3$ | $1\pm1$ | $86\pm1$ | $17\pm10$ | $5\pm9$ | $86\pm1$ | $31\pm19$ | $16\pm13$ | $84\pm1$ | $20\pm6$ | $7\pm4$ |
| RFNn | $81\pm0$ | $255\pm6$ | $84\pm3$ | $76\pm1$ | $74\pm5$ | $9\pm2$ | $81\pm0$ | $133\pm3$ | $23\pm1$ | $81\pm0$ | $250\pm7$ | $81\pm4$ | $77\pm0$ | $162\pm5$ | $45\pm3$ |
| DAE | $72\pm0$ | $234\pm2$ | — | $65\pm0$ | $53\pm2$ | — | $72\pm0$ | $122\pm1$ | — | $72\pm0$ | $230\pm2$ | — | $69\pm0$ | $140\pm2$ | — |
| RBM | $3\pm0$ | $525\pm6$ | — | $8\pm0$ | $93\pm3$ | — | $3\pm0$ | $273\pm4$ | — | $3\pm0$ | $517\pm6$ | — | $5\pm0$ | $301\pm4$ | — |
| FAsp | $77\pm1$ | $999\pm99$ | $999\pm99$ | $81\pm1$ | $338\pm33$ | $673\pm99$ | $79\pm0$ | $999\pm42$ | $999\pm99$ | $77\pm1$ | $999\pm99$ | $999\pm99$ | $79\pm1$ | $999\pm74$ | $999\pm99$ |
| FAlap | $3\pm0$ | $75\pm6$ | $999\pm94$ | $4\pm0$ | $18\pm2$ | $479\pm31$ | $4\pm0$ | $40\pm3$ | $831\pm43$ | $3\pm0$ | $74\pm6$ | $999\pm95$ | $4\pm0$ | $45\pm3$ | $999\pm58$ |
| ICA | $3\pm1$ | $0\pm0$ | — | $3\pm1$ | $0\pm0$ | — | $3\pm1$ | $0\pm0$ | — | $3\pm1$ | $0\pm0$ | — | $3\pm1$ | $0\pm0$ | — |
| SFA | $1\pm0$ | $28\pm1$ | $668\pm12$ | $1\pm0$ | $8\pm0$ | $78\pm8$ | $1\pm0$ | $15\pm0$ | $188\pm5$ | $1\pm0$ | $28\pm1$ | $644\pm9$ | $1\pm0$ | $18\pm1$ | $321\pm9$ |
| FA | $1\pm0$ | $28\pm1$ | $622\pm11$ | $1\pm0$ | $8\pm0$ | $64\pm7$ | $1\pm0$ | $15\pm0$ | $173\pm4$ | $1\pm0$ | $28\pm1$ | $599\pm9$ | $1\pm0$ | $18\pm1$ | $294\pm8$ |
| PCA | $1\pm0$ | $0\pm0$ | — | $4\pm0$ | $0\pm0$ | — | $2\pm0$ | $0\pm0$ | — | $1\pm0$ | $0\pm0$ | — | $2\pm0$ | $0\pm0$ | — |

## 14 RFN Pretraining for Convolution Nets

We assess the performance of RFN *first layer* pretraining on *CIFAR-10* and *CIFAR-100* for three deep convolutional network architectures: (i) the AlexNet Krizhevsky et al. [2012], (ii) Deeply Supervised Networks (DSN) Lee et al. [2014], and (iii) our 5-Convolution-Network-In-Network (5C-NIN).

Both CIFAR datasets contain 60k 32x32 RGB-color images, which were divided into 50k train and 10k test sets, split between 10 (CIFAR10) and 100 (CIFAR100) categories. Both datasets are preprocessed as described in Goodfellow et al. [2013] by global contrast normalization and ZCA whitening. Additionally, the datasets were augmented by padding the images with four zero pixels at all borders. For data augmentation, at the beginning of every epoch, images in the training set were distorted by random translation and random flipping in horizontal and vertical directions. For the AlexNet, we neither preprocessed nor augmented the datasets.

Inspired by Lin et al. [2013]'s Network In Network, we constructed a 5-Convolution-Network-In-Network (5C-NIN) architecture with five convolutional layers, each followed by a 2x2 max-pooling layer (stride 1) and a multilayer perceptron (MLP) convolutional layer. ReLUs were used for the convolutional layers and dropout for regularization. We followed Krizhevsky [2009] for weight initialization, learning rates, and learning policies. The networks were trained using mini-batches of size 100 and 128 for 5C-NIN and AlexNet, respectively.

For RFN pretraining, we randomly extracted 5x5 patches from the training data to construct 192 filters for DSN and 5C-NIN while 32 for AlexNet. These filters constitute the first convolutional layer of each network which is then trained using default setting. For assessing the improvement by RFNs, we repeated training with randomly initialized weights in the first layer. The results are presented in Tab. 9. For comparison, the lower panel of the table reports the performance of the currently top performing networks: Network In Network (NIN, Lin et al. [2013]), Maxout Networks (MN, Goodfellow et al. [2013]) and DeepCNiN Graham [2014]. *In all cases pretraining with RFNs decreases the test error rate.*

Table 9: The upper panel shows results of convolutional deep networks with first layer pretrained by RFN ("RFN") and with first layer randomly initialized ("org"). The first column gives the network architecture, namely, AlexNet, Deeply Supervised Networks (DSN), and our 5-Convolution-Network-In-Network (5C-NIN). The test error rates are reported (for CIFAR-100 DSN model was missing). Currently best performing networks Network In Network (NIN), Maxout Networks (MN), and DeepCNiN are reported in the lower panel. In all cases pretraining with RFNs decreased the test error rate.

| Dataset | CIFAR-10 | | CIFAR-100 | | |
|---|---|---|---|---|---|
| | org | RFN | org | RFN | augmented |
| AlexNet | 18.21 | 18.04 | 46.18 | 45.80 | |
| DSN | 7.97 | 7.74 | 34.57 | - | $\sqrt{}$ |
| 5C-NIN | 7.81 | 7.63 | 29.96 | 29.75 | $\sqrt{}$ |
| NIN | 8.81 | - | 35.68 | - | $\sqrt{}$ |
| MN | 9.38 | - | 38.57 | - | $\sqrt{}$ |
| DeepCNiN | 6.28 | - | 24.30 | - | $\sqrt{}$ |

## 15 Running Times for RFN's Projected Newton Step

In this section, we report the running times for RFN's projected Newton step and for solving a quadratic program using NumPy Python and CVXOPT (Python Software for Convex Optimization), respectively. Both benchmarks were profiled with the same hardware using only the CPU. Fig. 1 shows the run times for various problem sizes in [s] both approaches. The projected Newton step complexity per iteration is $O(nl)$, see Fig. 2. In contrast, a quadratic program solver typically requires for the $(nl)$ variables (the means of the hidden units for all samples) $O(n^4 l^4)$ steps to find the minimum Ben-Tal and Nemirovski [2001].

Figure 1: Running times for various problem sizes in [s] of RFN's projected Newton step and of quadratic program solver.

Figure 2: Running times for various problem sizes in [s] of RFN's projected Newton step.