[Reviews · NeurIPS 2015]

Submitted by Assigned_Reviewer_1

It would be good to see the running times for all the algorithms. Using _five_ different algorithms for computing a gradient step towards the constrained posterior seems excessive and makes the algorithm less attractive. How necessary is each one of these steps?
Summary: The paper proposes using sparsity-enforcing posterior regularization to make inference in factor analyzers a nonlinear process, which allows stacking such models to obtain hierarchical representations. This an interesting technique, though I wonder how scalable it is due to its batch nature. The results look reasonable, even if the algorithm seems considerably more expensive than the baselines it is compared to.

Submitted by Assigned_Reviewer_2

The authors present an intriguing take on factor analysis: namely, the use of Ganchev et al's posterior regularization to enforce non negativity constraints on the posterior. They present proofs of convergence and correctness, and present a scalable method for inference and learning in stacked constrained factor analysis models.

The method appears to be sound and is an interesting direction overall, a refreshing departure from much of the existing literature, and the first instance of which I am aware that posterior regularization has been highlighted in a deep learning/unsupervised feature learning context. While I did not review it in detail, the degree of thoroughness demonstrated by the supplementary material is truly impressive.

My main concern with this paper is one of being somewhat underwhelmed by the empirical evaluation, in light of the norms of the community. It makes the contribution difficult to judge from that vantage point.

In particular, one question I'm left with is whether there is a reason that these models, at least in the one layer case (and probably also in the multi-layer case) cannot be evaluated in terms of their test set likelihood. The quantitative evaluation of unsupervised methods on MNIST, for example, is usually phrased in terms of test set likelihood. Synthetic data results are presented for a set of chosen metrics meant to demonstrate the effectiveness of the model at satisfying the design goals of RFNs, but are difficult to interpret. The classification baselines employed also seem to be a bit dated. For example, Komer et al (2014) report a test set error of 11.7% on CONVEX using a combination of a polynomial SVM and PCA preprocessing. While that result is relatively hard to find, unsupervised learning on CIFAR, for example, has progressed quite a bit (albeit in some cases with domain-knowledge-heavy architectures).

- "Current unsupervised deep learning approaches like autoencoders or restricted Boltzmann machines (RBMs) do not model specific structures in the data." This is unclear. It seems that the filters of individual units can model "specific structures". The next sentence refers to "generative models" but RBMs and (deep belief networks built from them) _are_ generative models. More precision is required in this section about what is meant. - I'm curious as to the authors' response to the success of representations that eschew sparsity altogether and still manage to perform very well, such as maxout networks (Goodfellow et al, 2013). - More elaboration on the role of normalizing constraints would be helpful in exposition. - I assume the "projected Newton method" is the Newton step projected onto the constraint surface in terms of the closest (in terms of L2 distance) point obeying the constraint. It would be good to explicitly state this.

POST-REBUTTAL COMMENTS: - The argument against using likelihood is somewhat compelling though it seems like in a deep model one should be able to recover the lost ground caused by posterior regularization in a shallow model. - I'd suggest moving the CIFAR results front and center along with the drug design stuff, and perhaps relegating the table presented to the supplementary. CIFAR10/100 being a point of contact for many in the deep learning world I think highlighting that in the main text is helpful.

I've upgraded my score to a 7.
Summary: An intriguing take on factor analysis, applying non-negativity constraints on the posterior to obtain sparse representations. The method appears to be sound and there is detailed discussion of implementation details necessary for scaling up; quantitative empirical evaluation is somewhat lacking due to the datedness and uncommonness of certain benchmarks.

Submitted by Assigned_Reviewer_3

The paper proposes a novel generative unsupervised model aimed at obtaining representations with a variety of properties which are excepted as desirable for representations -- sparse, non-negative, high in dimentionality. The model builds upon factor analysis by enforcing in addition the desired properties by posterior regularization.

The model shows very good performance as autoencoder for reconstruction as well as a pre-training for classification networks. The model is well studied both analytically and empirically. The results are convincing of its merits.

A few comments: -- A better explanation is needed why one needs to try five different gradient descend methods in the E-step of the learning (lines 136 - 140) in the given order? This seems a bit arbitrary. -- It would be informative to do an ablative analysis of RFN by removing all constraints (normalization is being removed but no non-negativity for example).
Summary: The paper introduces a convincing, both analytically as well as empircally, method which deserves the attention of the NIPS audience.

Submitted by Assigned_Reviewer_4

- the authors try to make the case for non-negative transforms, like ReLUs, being important for successful models when they say "Representations learned by ReLUs are not only sparse but also non-negative" and "Correctness means that the RFN codes are non-negative, sparse, have a low reconstruction error, and explain the covariance structure of the data." In that case, how does one explain MSR's paper with PReLUs outperforming ReLUs when PReLUs are not non-negative. This is a weak argument, the only serious advantage is the non-saturating nature of the transform.

- The authors explicitly state that "In summary, our goal is to construct input representations that (1) are sparse, (2) are non-negative, (3) are non-linear, (4) use many code units, and (5) model structures in the input data" as mentioned above. Not sure what the importance of goal#2 is. Goal#4 is a questionable goal - if anything, good representation will capture the same structure in the data with less hidden units. Low reconstruction error would be a good goal - it's missing. - The authors state "Current unsupervised deep learning approaches like autoencoders or restricted Boltzmann machines (RBMs) do not model specific structures in the data. On the other hand, generative models explain structures in the data but their codes cannot be enforced to be sparse and non-negative." - are they trying to say that RBMs are not generative? or is it unclear writing? Hopefully the latter. - Table 1 is confusing. The authors state it demonstrates their method has lowest reconstruction error and yet PCA has lower reconstruction methods in the table? - Figure 2 supposedly shows the method is robust to background noise. However, the filters look significantly less robust than what I've seen in denoising autoencoders - Table 2 is misleading: it compares accuracy on computer vision task across deep learning models that leverage pretraining and the authors show that they are competitive with state of the art. Except they seem to have omitted schmidhuber's convolutional autoencoders (http://people.idsia.ch/~ciresan/data/icann2011.pdf) which beat their method on MNIST and really outperform on CIFAR 10 - the author's test test error is 41% , Schmidhuber gets 22% and that's from 2011, today the state of the art is 9%.

For the experiments, I'd have liked to see: - "RFNs vs. other unsupervised methods" should have included some explicit sparse coding methods (e.g., l1 regularized regression). They should have done comparisons of sparsity and reconstruction error on real-world examples, instead of just on the toy dataset.

- "RFN pretraining for deep nets" would have been stronger if the authors compared their methods to more other methods for pre-training. Instead they mostly do comparisons against other non-deep-net methods, which is not the point.

- It would also have been nice to see the results of classification tasks done directly on the output of the RFNs vs. on the output of other unsupervised methods.
Summary: The authors work out many details needed to efficiently apply posterior regularization to factor analysis. The results shown in this paper are promising applications to unsupervised learning and unpromising applications to pre-training strategies of deep learning models - classification accuracy on CIFAR10 is significantly worse than Convolutional Neural Networks pre-trained with convolutional auto-encoders and far worse than state of the art.

Submitted by Assigned_Reviewer_5

This paper addresses the problem of learning a good representation, unsupervised. It proposes a 'deep'(stacked) version of factor analysis model, with a posterior constraints. The optimization is done by EM with a projection in E step making sure constraints are always satisfied. It compares with several other popular unsupervised method and show a consistent improvement.

To the best of my knowledge, it's the first to introduce posterior regularization to a deep factor analysis model and solve it using a simple projected version of EM algorithm.

Section 2 is not clearly written which reads much more complicated than it should be.

The author claims huge speedup for their algorithm over regular Newton method but how is the number of steps computed at line 145? and it would be better to compare timing for different methods as well.

For first experiment, as the PCA gives 0 error on reconstruction with 100/150 code units, the dataset seems to be trivial. While comparing RFN and RBM, RFN is better in terms of reconstruction but RBM is better in terms of sparsity. It's hard to say RFN is much better and also it would be better to explain more about the numbers themselves, i.e. how much difference should be considered huge.

For second experiment on pre-training, it shows improvement of around 0.5%, but on MNIST, this difference doesn't mean much. And the proposed method doesn't do well on CIFAR. So it would be great to see experiments on more real and larger dataset.
Summary: This paper proposes a novel unsupervised learning model for learning sparse, non-linear, high-dimensional representation. Experimental results demonstrate the effectiveness of the model.

Author Feedback
Author rebuttal: We wish to thank the reviewers for their time, helpful comments, and useful hints. In our opinion, we obtained high quality reviews. We will follow all reviewers' suggestions on which we do not comment below.
In the limited space, it was difficult to present a novel method, its theoretical background, and all experiments. Therefore we moved material and experiments to the supplement.

A - The main critique of Reviewer1 is that the "quantitative empirical evaluation is somewhat lacking due to the datedness and uncommonness of certain benchmarks" and the main critique of Reviewer4 is that the "classification accuracy on CIFAR10 is significantly worse than Convolutional Neural Networks pre-trained with convolutional autoencoders and far worse than state of the art." and "today the state of the art is 9%."

Due to space constraints we did not show results for convolutional networks (CNNs) in the main manuscript. However, in the supplementary Section 14 "RFN Pretraining for Convolution Nets" we presented results using CNNs. On CIFAR10, RFNs led to a test error rate of 7.63% and to 29.75% on CIFAR100, in each case improving upon nets without RFN pretraining. At the time of the experiments, 7.63% and 29.75% were the lowest reported error rates on CIFAR10 and CIFAR100, respectively, until Graham's CNN architecture. They are still among the best known results. Our fault was that we did not properly refer to the supplement: we apologize for that and will correct that.

B - Some reviews focused on our experiments where we used RFNs for pretraining deep nets. We think that novel unsupervised methods should not only be judged by their success in pretraining supervised architectures. Our largest datasets (ref. [29]) were drug design data from big pharma, where RFNs led to new biological insights while PCA, factor analysis, correspondence factor analysis (CFA), spectral map analysis (SMA), ICA, various clustering approaches, etc. failed.

C - Due to our unclear writing we gave the impression that we do not consider RBMs as generative models (Reviewer1 and Reviewer4) - sorry. We wanted to say that our goal was to design an unsupervised method which codes for specific structures in the data irrespective of noise, using a generative model. In contrast, RBMs and autoencoders do encode any and all peculiarities in the data (including noise). We will clarify this in the new version, especially that RBMs are generative models.

D - (Reviewer3 and Reviewer4) "PCA has lowest reconstruction error in table 1". If PCA/ICA have at least as many code units as input dimensions, then their reconstruction error is zero. We compared sparse representation methods in the upper half of the table (separated by a line) while the lower half serves as reference. We wrote that RFNs have the lowest reconstruction error "of all methods that yielded sparse representations (SP>10%)". We will write this more clearly.

E - We will include running times to show the overwhelming improvement in computational complexity of RFNs (Reviewer3 and Reviewer6).

Reviewer1:
- "likelihood": The likelihood is not a good criterion because the posterior regularization method trades decreased likelihood for increased sparseness (see ref [12] and supplement p. 20). Methods that only maximize the likelihood are supposed to have higher likelihood. We will add a note to the paper.
- "maxout": In our view maxout is implicitly sparse, as units that do not contribute to maxima might as well be set to zero (same for maxpooling).
The reviewer's main critique is that the "quantitative empirical evaluation is somewhat lacking". Referring to item A, we kindly ask Reviewer1 to reconsider their decision and to upgrade their score.

Reviewer3:
- "RBMs sparser": RFNs are sparser than RBMs: the higher the value, the sparser the code (Tab.1).
- "real and larger dataset": See item A & B.

Reviewer4:
- PReLUs outperform ReLUs whenever the number of units is too small. For each PReLU net there is an equivalent ReLU net with twice as many units.
- Low reconstruction error is not an RFN goal: this would require encoding both noise and signal, but RFNs aim at encoding only noise-free structures.
- "included some explicit sparse coding methods (e.g., l1 regularized regression)": FAlap uses a Laplace prior on the hidden units, which is L1. FAsp uses an even sparser (Jeffrey's) prior.
The reviewer's main critique is that the "classification accuracy on CIFAR10 is significantly worse than Convolutional Neural Networks". Referring to item A, we kindly ask Reviewer4 to reconsider their decision and to upgrade their score.

Reviewer6:
The last algorithm guarantees convergence, while the other trade off the probability of a successful update step against speed. In almost all cases, the first/fastest algorithm suffices.